# Late Quaternary faulting in southern Matese (Italy): implications for earthquake potential and slip rate variability in the southern Apennines

Paolo Boncio[1,2], Eugenio Auciello[3], Vincenzo Amato[4], Pietro Aucelli[5], Paola Petrosino[6], Anna C. Tangari[7], Brian R. Jicha[8]

[1]Università degli Studi "G. d'Annunzio" Chieti - Pescara, Department of Engineering and Geology, Chieti, 66100, Italy
[2]CRUST – Interuniversity Center for 3D Seismotectonics with territorial Applications, Chieti, 66100, Italy
[3]Geoscience practitioner, Pesche (IS), Italy
[4]Università degli Studi del Molise, Department of Biosciences and Territory, Pesche (IS), 86090, Italy
[5]Università degli Studi di Napoli Parthenope, Department of Science and Technology, Neaples, 80133, Italy
[6]Università degli Studi di Napoli Federico II, Department of Earth, Environmental and Resources Sciences, Neaples, 80126, Italy
[7]Università degli Studi "G. d'Annunzio" Chieti - Pescara, Department DiSPUTer, Chieti, 66100, Italy
[8]University of Wisconsin-Madison, Department of Geoscience, Madison, Wisconsin 53706-1692, USA

*Correspondence to*: Paolo Boncio (paolo.boncio@unich.it)

**Abstract.** We studied in detail the Gioia Sannitica active normal fault (GF) along the Southern Matese Fault (SMF) system in the southern Apennines of Italy. The current activity of the fault system and its potential to produce strong earthquakes have been underestimated so far, and are now defined. Precise mapping of the GF fault trace on a 1:20,000 geological map and several point data on geometry, kinematics and slip rate of the faults forming the SMF system are made available in electronic format. The GF, and in general the entire fault system along the southern Matese mountain front, is made of slowly-slipping faults, with a long active history revealed by the large geologic offsets, mature geomorphology, and complex fault pattern and kinematics. Present activity has resulted in Late Quaternary fault scarps resurrecting the foot of the mountain front, and Holocene surface faulting. Resurrected mountain front indicates variation of slip rate through time. The slip rate varies along-strike, with maximum latest Pleistocene – Holocene slip rate of ~0.5 mm/yr. Activation of the 11.5 km-long GF can produce up to M 6.2 earthquakes. If activated together with the 18.5 km-long Ailano-Piedimonte Matese fault (APMF), the seismogenic potential would be M 6.8. The slip history of the two faults is compatible with a contemporaneous rupture. The observed Holocene displacements on the GF and APMF are compatible with activations during some poorly constrained historical earthquakes, such as the 1293 (M 5.8), 1349 (M 6.8; southern prolongation of the rupture on the Aquae Iuliae fault?) and 346 earthquakes. A fault rupture during the 847 poorly-constrained historical earthquake is also chronologically compatible with the dated displacements.

# 1 Introduction

Detailed field mapping of active faulting is essential for populating fault databases oriented at mitigating the seismic risk from ground shaking and fault displacement hazard (e.g., DISS Working Group, 2018; Styron and Pagani, 2020; Faure Walker et al., 2021; California U.S. Alquist-Priolo Earthquake Fault Zoning Act, https://www.conservation.ca.gov/cgs/alquist-priolo; New Zealand Active Faults Database, https://data.gns.cri.nz/af/; ITaly HAzards from CApable faults, ITHACA, http://sgi2.isprambiente.it/ithacaweb/). The implementation of accurate fault mapping is particularly important in areas where geodetic or seismologic evidence of active tectonics contrast with poor knowledge of active faulting from surface geology.

In the central-southern Apennines of Italy, presently stretching at rates of ~3 mm/yr in the SW-NE direction, two areas were highlighted as being characterized by significant deficit of seismic moment release for the last 500 years (D'Agostino, 2014; Fig. 1). Shortly after the characterization, one of the areas was struck by the 2016 central Italy normal faulting earthquakes on the Mt. Vettore – Mt. Bove normal fault (maximum Mw = 6.5; Chiaraluce et al., 2017; Civico et al., 2018). The second area extends for ~80 km between the central and southern Apennines, and includes a large part of the Matese Mts. In the Mt. Vettore – Mt. Bove area, paleoseismologic studies have demonstrated that the fault responsible for the 2016 earthquake ruptured repeatedly in prehistoric times, with average recurrence interval 1.8±0.3 kyr, and the penultimate earthquake occurred well before the historical catalogue (Cinti et al., 2019; Galli et al. 2019). Thus, the 2016 earthquakes demonstrate that the geology of active faults is equally as critical as historical seismicity for estimating the true seismic potential in areas characterized by low strain rates (velocities of a few mm/yr) and long return periods of strong earthquakes (≥ M 6.5), such as the Italian Apennines. This has implications in seismic hazard assessments (e.g., Valentini et al., 2019).

In the Matese Mts, the geology of active faults is poorly constrained, with only two known active normal faults: the SW-dipping Aquae Iuliae fault and the NE-dipping Northern Matese fault system (Di Bucci et al., 2005; Galli and Naso, 2009; Boncio et al., 2016; Ferrarini et al., 2017; Galli et al., 2017) (Fig. 1). To the SE, the activity of the Matese normal faults is less constrained. In a recent attempt to derive fault slip rates from geodetic data for the Matese area (Carafa et al., 2020), the results were not conclusive due to the sparseness of GNSS stations and the paucity of geologic constraints.

This paper focuses on the normal faults cropping out along the southern slopes of the Matese Mts., named Southern Matese Fault system (SMF, Ailano – Piedimonte Matese and Gioia Sannitica faults; Fig. 2). The main goal is to determine if the SMF must be considered active, and with which seismogenic potential and slip rate. Attention is paid to evidence of slip rate variations. Discovering variations in the rate of activity is important for identifying the most appropriate time window for computing the average long-term slip rate and its variability, which in turn are crucial parameters for assessing earthquake recurrence and the associated seismic hazard (e.g., Cowie et al., 2012). There is evidence that slip rates vary in time for the active normal fault systems of the Italian Apennines (e.g., Benedetti et al., 2013; Cowie et al., 2017). Evidence of possible temporal slip rate variation along the SMF is suggested by paleoseismological data on the Aquae Iuliae fault (Galli and Naso, 2009), where the paleoseimology-derived throw rates (1.5-1.9 mm/yr) are 3-to-4 times higher than the long-term throw rates estimated from offset of Middle Pleistocene terraces or from fault scarp morphology (post-230 ka to post-Last Glacial

Maximum throw rates of 0.2-0.5 mm/yr; Boncio et al., 2016; Cinque et al., 2000; Galli and Naso, 2009). These differences open a question as to whether they are due to uncertainties in constraining the ages and offsets of geological and geomorphological markers, or are evidence of slip rate variations.

A tectonic geomorphology investigation of the mountain front along SMF system (Ascione et al., 2018; Valente et al., 2019) highlights that in the last ~600 ka, the E-W-striking faults in the central part of the system are characterized by higher slip rate compared to the NW-SE-striking faults located at the western and eastern sides of the system, which are considered as less active faults. In particular, a higher mountain front maturity, interpreted as a consequence of low, waning fault activity, has been highlighted for the NW-SE-striking San Potito mountain front along the Gioia Sannitica fault, which is the fault studied in this work. These evidences of low tectonic activity along the NW-SE San Potito mountain front described by Valente et al. (2019) appear to contrast with the presence of meters-scale fault scarps along the fault trace revealed by high resolution topography that will be described in this paper, and which might indicate recent surface faulting. These apparently contrasting evidences might be an indication of recent fault reactivation, or acceleration of a former low-slipping fault, which would imply slip rate variation through time.

We seek to answer the above questions by performing an earthquake geology study aimed at mapping the fault traces in detail, collecting field evidence of recent activity, particularly Late Pleistocene – Holocene activity, looking for evidence of earthquake-related surface faulting episodes, and combining all the collected data in a consistent seismotectonc frame for estimating the likely earthquake potential. We start studying in detail the geology of the Gioia Sannitica normal fault, for the motivations introduced above, and because this is the less constrained fault of the system. The results are described in Section 4, and a detailed, 1:20,000-scale geologic map of the Gioia Sannitica normal fault is attached as supplementary material (Plate S1). Fault scarp heights are recognized, carefully selected to avoid scarps of non-tectonic origin, measured using high-resolution topography, and used to derive fault slip rates. In sub-section 4.2, we describe evidence of Holocene surface faulting, discovered both on the Gioia Sannitica and Ailano – Piedimonte Matese faults. For the first time, we show clear evidence of late Quaternary and Holocene faulting, thanks to detailed field analyses of fault zones in Quaternary sediments and radiometric dating ($^{14}$C and $^{40}$Ar/$^{39}$Ar) of faulted sediments. In Section 5 we integrate our new data with data deriving from previous geological works on the Ailano – Piedimonte Matese fault (Boncio et al., 2016). Our results are then discussed in the light of the study by Valente et al. (2019). Finally, the new and pre-existing data are discussed together in terms of present activity, overall seismogenic potential, and slip rate variability of the SMF system.

## 2 Geologic and seismotectonic setting

### 2.1 General geologic setting

The Matese Mts. form a 20 km-wide, 50 km-long massif of carbonate rocks elongated in the NW-SE direction, in the northern part of the southern Apennines of Italy. The massif has a maximum elevation of about 2000 m a.s.l. and is delimited to the NE by the Bojano depression and to the W and SW by the valley of the Voltuno River (Venafro and Alife depressions). This

morphologic setting is largely due to Quaternary extensional tectonics and down-faulting along NE-dipping and SW-dipping normal faults that border the massif to the NNE and to the SSW, respectively (Aucelli et al., 2013; Amato et al., 2014; Ascione et al., 2018; Valente et al., 2019).

The Matese massif is made up of successions of Meso-Cenozoic carbonate rocks with sedimentary facies varying from
shallow-water carbonate platform in the south to by-pass margin and slope-to-basin transition facies in the north. The arrangement of the Meso-Cenozoic sedimentary facies was controlled by Jurassic normal faults, striking mostly E-W, that determined the progressive transition from a persistent structural high in central-southern Matese to deep basin conditions in northern Matese (Calabrò et al., 2003; Valente et al., 2019). The Cretaceous, Jurassic or Triassic carbonate successions are covered discontinuously by Upper Miocene hemipelagic and turbiditic siliciclastic deposits related to the Apennine orogeny
(D'Argenio et al., 1973; Di Bucci et al., 1999; Patacca and Scandone, 2007 with references).

Late Miocene - Pliocene Apennine compression deformed the Meso-Cenozoic units via NE-directed shortening, generating S-dipping reverse faults and NNE-to-N-verging folds and monoclines. The geometry of compressional structures was largely conditioned by the pre-existing E-W faults. The Mio-Pliocene compressional structures are post-dated by Quaternary high-angle normal faults. Normal faulting dissected the western side of the Matese Mt.s since Early Pleistocene, produced a massive
deposition of slope-derived breccias, and formed the Venafro and Alife depressions along the Volturno River valley (Ferranti et al., 1996; Brancaccio et al., 1997; Calabrò et al., 2003). The early stages of crustal extension (Early-Middle Pleistocene) were characterized by the formation of major SW-NE-trending structures such as the Garigliano graben, which hosts the Roccamonfina volcano west of the Matese ridge, and the Venafro basin on the north-western side of Matese. There is also evidence of Middle Pleistocene volcaniclastic sediments faulted by SW-NE-striking normal faults originated during this phase
of NW-SE-oriented extension (Amato et al.., 2014, 2017; Boncio et al., 2016). During this stage, inherited E-W-striking Mesozoic normal faults were reactivated with left-lateral normal-oblique kinematics (Boncio et al., 2016).

After the NW-SE-directed extension, normal faulting driven by SW-NE extension became dominant in the Middle Pleistocene, generating major NW-SE-striking faults and reactivating pre-existing normal and strike-slip faults across the entire Matese area (Ferranti et al., 1996; Calabrò et al., 2003; Di Bucci et al., 2005; Amato et al., 2014). Overall, five major depressions
originated due to normal faulting: the Venafro and Alife depressions to the W and SW, the Isernia and Bojano depressions to the N and NE and the Matese Lake depression in the core of the Matese massif (Fig. 1). The Quaternary extension was accompanied by accumulation of continental deposits within the depressions and along the mountain fronts, including Pleistocene slope-derived breccias, lacustrine and alluvial deposits and large Middle Pleistocene to Holocene alluvial fans (Brancaccio et al., 1997; Di Bucci et al., 2005; Amato et al., 2014; Valente et al., 2019). Middle Pleistocene sediments often
contain pyroclastic horizons, mostly deriving from the nearby Roccamonfina stratovolcano (0.55-0.15 Ma activity; Luhr and Giannetti, 1987; Rouchon et al., 2008) and from the Campanian Volcanic Zone (Rolandi et al., 2003), including the 39 ka-old Campanian Ignimbrite and the 15 ka-old Neapolitan Yellow Tuff (De Vivo et al., 2001; Deino et al., 2004; Giaccio et al., 2017).

## 2.2 Quaternary tectonics

Three main normal fault systems can be mapped in the Matese area (Fig. 1): 1) the NE-dipping Northern Matese Fault system; 2) the SW-dipping Aquae Iuliae and Gallo-Letino-Matese Lake Fault systems in central Matese; and 3) the SW-dipping SMF system. The Presenzano - Ailano fault system connects the SMF with the SW-dipping San Pietro Infine fault and allows extensional strain to be transferred to the west (Boncio et al., 2016).

Late Quaternary faulting and Holocene surface faulting associated with historic or pre-historic earthquakes are known for the

Northern Matese Fault system (Galli and Galadini, 2003; Di Bucci et al., 2005; Ferrarini et al., 2017; Galli et al., 2017) and for the Aquae Iuliae Fault (Galli and Naso, 2009; Boncio et al., 2016). For the Gallo-Letino-Matese Lake Fault system, evidence of post-compressional Quaternary faulting has been documented using structural geology and tectonic geomorphology analyses (Bousquet et al., 1993; Calabrò et al., 2003; Aucelli et al., 2013; Valente et al., 2019).

The SMF can be divided into the Ailano – Piedimonte Matese fault to the NW, and the Gioia Sannitica fault to the SE (Boncio

et al., 2016). The 18 km-long Ailano – Piedimonte Matese fault is in turn divided into the Raviscanina and Piedimonte Matese fault sections (Fig. 2). The Raviscanina section is 11.5 km long, strikes NW-SE and progressively bends to ~W-E in the southern part (~1 km SE of the Sant'Angelo d'Alife village). The Piedimonte Matese fault section is 7 km long and strikes from W-E to WSW-ENE. The eastern part of the Piedimonte Matese fault section, striking ~ W-E, has strong geomorphic evidence of Quaternary activity, while the western part, striking WSW-ENE, is less evident due to cover deposits and larger

mountain front sinuosity.

Along the Raviscanina section of the Ailano – Piedimonte Matese fault, a post-350 ka throw rate of 0.27-0.30 mm/yr has been documented by Boncio et al. (2016). The strongest geomorphic evidence of Late Quaternary faulting is in the southern part of the section, close to the bend, where Late Quaternary throw rates of ≥0.15 mm/yr have been estimated. Boncio et al. (2016) suggest that the entire SMF may be presently active and possibly responsible for the 346 earthquake.

Valente et al. (2019), integrating a previous study by Ascione et al. (2018), performed a tectonic geomorphology and Quaternary stratigraphy analysis of the SMF mountain front, including the system of alluvial fans along the mountain front and the sedimentary basin in the hanging wall (Alife basin). The authors divide the SMF into the Raviscanina-Piedimonte Matese front (RPf in their Fig. 1) and San Potito Sannitico front (SPf in their Fig. 1). The Raviscanina-Piedimonte Matese front does not coincide with the Ailano - Piedimonte Matese fault previously defined in Boncio et al. (2016), as they do not

consider the northern, NW-SE-striking fault strand near Ailano. The San Potito Sannitico front corresponds to the northern half of the Gioia Sannitica fault.

In their study, Ascione et al. (2018) and Valente et al. (2019) show that the geomorphology of the SMF front varies along-strike. The shape of the mountain front changes from convex in the Raviscanina area, to rectilinear in the Piedimonte Matese area, to concave in the San Potito area. The calculated mountain front sinuosity (from 1.53 to 1.64) seems to indicate a relatively

slow mountain front activity (Bull, 1987), with values that are lower in the Raviscanina - Piedimonte Matese front (1.53) compared to the San Potito Sannitico front (1.64). The morphometry and the degree of entrenching of the alluvial fans is

consistent with a higher tectonic subsidence in the Raviscanina - Piedimonte Matese front compared to the San Potito Sannitico front. In summary, the authors suggest that there is an increase in maturity of the mountain front moving from NW (Raviscanina) to SE, due to an evolution of the SMF since ~600 ka ago that determined a differentiation of the activity rate of the central part of the system compared to the faults at the eastern and western sides of the system. In particular, the Raviscanina - Piedimonte Matese front, that in their final model is thought to be a main fault with average E-W strike, is considered to have the highest activity, while the NW-SE San Potito Sannitico front is considered to have lower, waning activity rate. For the Raviscanina - Piedimonte Matese mountain front, they estimated a post-Middle Pleistocene throw rate of > 0.2-0.3 mm/yr.

## 2.3 Seismicity

Several strong earthquakes struck the Matese region during the last two millennia (346, 847, 1293 M 5.8, 1349 M 6.8, 1456 M 7.2, 1688 M 7.1, 1805 M 6.7; Guidoboni et al., 2019; Rovida et al., 2020; Fig. 1). The NE-dipping Northern Matese Faults were responsible for the 1805 earthquake in the Bojano area (Espostito et al., 1987; Porfido et al., 2002; Cucci et al., 1996; Galli and Galadini, 2003; Serva et al., 2007), and possibly for the first shock of the 1456 earthquake sequence (Galli and Galadini, 2003), even though for the latter event a different interpretation has been proposed (Fracassi and Valensise, 2007). The location of the ancient 847 earthquake is highly uncertain. In the CFTI5Med catalogue (Guidoboni et al., 2019), it is located north of the Matese massif, near Isernia. According to Bottari et al. (2020), the macroseismic area could be larger, and the source located significantly SW of the CFTI5Med epicentre, possibly on the Aquae Iuliae fault, as suggested by Galli and Naso (2009), or on the SMF system.

The 1349 earthquake has been associated to the SW-dipping Aquae Iuliae fault on the basis of paleoseismologic investigations (Galli and Naso, 2009), while the sources responsible for the 1293 and 1688 earthquakes in southern Matese are still unknown (1293) or poorly constrained (1688; Di Bucci et al., 2006; Serva et al., 2007). A strong ancient earthquake is known to have seriously damaged the area E, W and SW of the Matese massif in 346 (Galadini and Galli, 2004), but the causative fault is unknown. Only hypothetical associations to the Aquae Iuliae fault (Galli and Naso, 2009) or to the SMF system (Boncio et al., 2016; Valente et al., 2019) have been proposed.

Since 1980, there have not been strong earthquakes associated with a fault in the southern Matese. The largest recorded earthquake occurred in 2013 (Mw 5.2) within the core of the south-eastern Matese massif, at a depth of 10-20 km (Ferranti et al., 2015). The focal mechanism indicates normal faulting on NW-SE-striking normal faults, nearly parallel to the normal faults mapped at the surface, but its association to the down-dip prolongation of the outcropping faults is not clear. A small-magnitude event occurred in 2016 (Mw 4.1), ~10 km NE of the Northern Matese Fault system, at depths shallower than ~10 km. The 2013 and 2016 earthquakes share a common SW-NE-oriented direction of extension, which is consistent with directions obtained from structural geology of Late Quaternary normal faults (Ferranti et al., 2015; Boncio et al., 2016; Ferrarini et al., 2017) and GPS data (Ferranti et al., 2014; D'Agostino, 2014; Carafa et al., 2020).

# 3 Materials and methods

## 3.1 Field mapping and fault scarp measurements

Geological mapping was based on a traditional field survey supplemented with a digital survey on a GPS-integrated digital mapping suite (Field Move Software Suite, Petroleum Experts). The topographic maps in digital format were made available by the Campania Regional authority (Carta Tecnica Regionale, CTR: Italian name and acronym of the 1:5,000-scale topographic map of the Campania Region). The fault traces and Quaternary units were mapped on a 1 m-resolution Digital Elevation Model (DEM) from airborne Light Detection and Ranging (LiDAR) made available by the Italian Ministry of Environment. The LiDAR DEM was used to extract topographic profiles across fault scarps for measuring throw values. Particular attention was paid to the identification of appropriate sites for scarp analysis, which were selected in order to avoid anthropogenic modifications or erosional processes that might determine non-tectonic exhumation of the fault. Throw and slip rates across the measured fault scarps were estimated using the age constraints of faulted sediments/morphologies from the site or from nearby, correlated sites.

Two sites were analysed using a paleoseismologic approach. They are two outcrops of fault zones exposed by a road cut across the Gioia Sannitica fault splays and a small hand-dug trench (~3 m-long, ~1 m-deep) located in the vicinity of an outcropping limestone fault plane of the Raviscanina fault section of the Ailano – Piedimonte Matese fault.

## 3.2 Sample dating

Three tephra layers interbedded within faulted alluvial fan and colluvial sediments were sampled for lithological analysis and $^{40}Ar/^{39}Ar$ dating. Samples were mostly extracted from deeply argillified slightly indurated to lithified deposits. In SEM laboratory of DiSTAR - University of Naples "Federico II", they were repeatedly washed in deionized water in order to remove the clay fraction, then treated with at least four,10 minutes-long ultrasonic washing with renewal of water. The clasts were dried and sieved at 1-phi intervals, finally a lithological component analysis was carried out on the 1 and 2 phi fractions under a binocular microscope. Unfortunately, no well-preserved glass fragment survived washing pre-treatment and hence chemical composition of glass, useful for tephrostratigraphic analysis, could not be achieved. The coarsest and best preserved sanidine phenocrysts were extracted from the three samples for $^{40}Ar/^{39}Ar$ dating. Single crystal fusions were performed at the University of Wisconsin-Madison. Isotopic analyses were conducted using a Noblesse multi-collector mass spectrometer (Jicha et al., 2016). The weighted mean ages listed in Tab. 1 are calculated relative to the 1.1864 Ma Alder Creek sanidine standard. Complete $^{40}Ar/^{39}Ar$ analytical data are provided in Tab. S1 of the supplementary material.

The radiocarbon dating was performed on charred material or bulk organic sediments using Accelerator Mass Spectrometry (AMS) technique in the laboratories of Beta Analytic (www.radiocarbon.com/) and CEDAD (Italy, http://www.cedad.unisalento.it/en/) (Tab. 2). For the buried paleosol faulted by the Gioia Sannitica fault (samples S228_3 and SPOT-1 in Tab. 2), given its importance for constraining the Holocene activity of the fault, and in order to verify the occurrence of possible contamination by young carbon, for one sample (SPOT-1) we dated: the alkali soluble organic fraction, the bulk

organic fraction, and the alkali insoluble fraction in the Beta Analytic laboratory (details of the method at https://www.radiocarbon.com/ams-dating-sediments.htm). The complete reports of the radiocarbon dating analyses are in Annex S1 of the supplementary material.

### 3.3 Paleosol analysis

The paleosol faulted by the Gioia Sannitica fault, which developed on an altered tuff, was analysed at the University of Chieti for macro/micro-morphology and geochemistry. Physical and chemical analyses (soil texture, pH ($H_2O$), organic matter content, and cation exchange capacity (CEC)) were performed on air-dried, and sieved (< 2 mm) fraction of the bulk sample of each soil horizon (van Reeuwijk, 2002; Burt, 2004). Micro-morphological observations were made on thin sections (10 cm x 5 cm x 30 µm) from undisturbed soil samples. Selective extraction techniques were used to determine different forms of Al,

Fe and Si such as acid ammonium oxalate extractable, Alo, Feo and Sio (Schwertmann, 1964), sodium pyrophosphate extractable, Alp (Bascomb,1968) and dithionite-citrate-bicarbonate extractable iron pool, Fed (Mehra and Jackson,1960). Their amount was measured using atomic absorption spectroscopy (AAS) on the fine earth fraction (< 2 mm). The total iron content, Fet, was analysed using a Rigaku Supermini X-ray fluorescence (XRF) spectroscopy. These data were used to calculate pedogenetic indices, shown in Tab. S2 of supplementary material, to estimate the andic properties of the paleosol

(ICOMAND, 1988; Parfitt and Wilson, 1985; Tangari et al., 2018) and the degree of soil maturity (e.g., Arduino et al., 1984; Scarciglia et al., 2018).

## 4 Results

### 4.1 Geology of the Gioia Sannitica normal fault from field mapping

The Gioia Sannitica normal fault (GF) has been mapped in detail along the piedmont of the southern Matese ridge (Fig. 2). A

detailed geologic map with cross sections is available as supplementary material in Plate S1. Plate S1 contains the synthetic stratigraphic logs of numerous shallow-depth drill holes used to constrain the Quaternary stratigraphy. The fault can be divided into two fault sections, on the basis of fault geometry, Quaternary geology and geomorphology of the hanging wall and footwall blocks: the San Potito section to the north and the Castello di Gioia section to the south.

### 4.1.1 Geomorphology and stratigraphy of the southern Matese piedmont along the GF

The GF originates a mountain front along the foot of the south-western slopes of the Matese carbonate ridge, striking on average N130° in the northern part (San Potito fault section) and N155° in the southern part (Castello di Gioia fault section), with a ~2 km-wide embayment in between (M. Erbano embayment; Fig. 2). The range slopes moderately (25-40°) down to the mountain front, without sharp, steep faceted spurs. The slope profile is straight to concave (see Fig. 2 and geologic sections in Plate S1), suggesting a poorly active or low-uplift-rate range front, as already suggested by Valente et al. (2019). However,

low linear scarps at the mountain front rise from the gently dipping slopes, indicating recent rejuvenation due to normal faulting (Fig. 2b).

Different generations of entrenched and overimposed alluvial fans and slope deposits accumulated in the hanging wall of the GF. Alluvial fans cover the bedrock units in the hanging wall of the GF in the northern part (San Potito section), where the top surfaces of alluvial fans slope gently to the SW down to the adjacent Alife plain. Isolated outcrops of carbonate or siliciclastic

pre-Quaternary bedrock arise from the piedmont, suggesting that the alluvial fans prograded into a morphologically articulated substratum. The hanging wall of the Castello di Gioia section is in general less depressed than the San Potito counterpart. Continental deposits are less diffuse and confined closer to the fault trace.

The morpho-stratigraphic relations among the mapped Quaternary units are shown in Fig. 3. The unit ages are based on volcanic layers and buried paleosols dated in this work, integrated with chronologic determinations in the Ailano – Piedimonte

Matese continental succession from previous works (Boncio et al., 2016; Valente et al., 2019) and regional-scale correlations with the Quaternary basins around the Matese massif (Brancaccio et al., 1997; Amato et al., 2014, 2017). We have distinguished two generations of slope-derived deposits (units sd1 and sd2) and four generations of alluvial fan deposits (units U1, U2, U3 and U4), interfingering with coeval fluvial (al) or lacustrine (lac) deposits in the Volturno river plain.

Units sd1 and U1 are made up of carbonate slope-derived breccias and dense to poorly cemented alluvial fan gravels,

respectively. Slope breccias crop out only at the base of carbonate slopes, often faulted against the Triassic dolostones. From a sedimentologic point of view, these deposits can be correlated with the Early-Middle Pleistocene slope-derived breccia cropping out widely along the fault-controlled range front in the Volturno River valley, in the Venafro basin, in the Prata Sannita area (Brancaccio et al., 1997; Amato et al., 2017), and in the Letino - Matese Lake area (Aucelli et al., 2013) (Fig. 1). In particular, the oldest mapped deposits are the well-cemented angular breccia of unit sd1, which can be reasonably correlated

with the Lower Pleistocene slope breccias of the Laiano Synthem (Carannante et al., 2011), described by Amato et al. (2018) in the Calore River area, about 20 km SE of the GF.

In the upper part of the U1 alluvial fan gravels, we sampled two layers of distal tephra containing altered leucite-bearing pumice fragments and sanidine crystals which gave $^{40}Ar/^{39}Ar$ ages of 564.5 ± 2.1 ka (sample S-227; Tab. 1) and 508.5 ± 0.9 ka (sample S228-F2) (samples located in the San Potito site of Fig. 2b; see also section 4.2.1; see Tab. S1 of the supplementary

material for complete $^{40}Ar/^{39}Ar$ data). The 564.9±2.4 ka age of sample S-AIL12, found by Valente et al. (2019) in the Ailano area, along the southern Matese mountain front, and there attributed to the early stages of activity of the Roccamonfina volcano, well corresponds to the age of 564.5±2.1 found here for sample S227, allowing the correlation. Unit U1 is hence coeval, at least in part, with the Ailano lacustrine deposits (Valente et al., 2019) and can be correlated with the lacustrine-fluvial unit MU1 described in the Venafro basin, about 30 km NW of the GF, by Amato et al. (2017), who dated the upper part of the unit

to ~475 ka. Therefore, based on tephra ages and regional correlations, the succession formed by units sd1 and U1 can be considered to be part of the Early Pleistocene to Middle Pleistocene (older than ~450 ka), with sd1 and U1 that are possibly etheropic to each other in the lower part of the succession (Fig. 3).

Unit U2 is made up of heterometric carbonate alluvial fan gravels in a brown silty-sandy matrix with interlayers of decimetre-thick dark brown leucite-free tuffs and thicker pedogenic layers observed in the field and in numerous boreholes (Plate S1). In the light of their stratigraphic position (younger than U1 unit) and of the fact that Low K-Series leucite-free explosive products of the 35 km-far Roccamonfina volcano are younger than at least 385 ka (Brown Leucitic Tuff, Luhr and Giannetti, 1987), these leucite-free tuffs can be tentatively ascribed to the White Trachytic Tuff series (De Rita and Giordano, 1996). So, the age of this unit reasonably ranges from Middle to Late Pleistocene.

Units sd2 and U3 comprise slope-derived colluvial and alluvial fan gravel deposits, respectively. Colluvial deposits sd2 (Late Pleistocene – Holocene) are made up of poorly organized gravels in a brown sandy matrix with dark silty-sandy pedogenic layers. A buried paleosol cropping out ~1.4 km ENE of San Potito Sannitico within the colluvial unit has been dated at 43.3 – 42.2 ka (calibrated age Before Present, cal. BP; Tab. 2), thus indicating that a large part of the colluvial gravels accumulated during the glacial periods of Late Pleistocene. Unit sd2 includes thin layers of Holocene colluvial deposits accumulated mostly in the hanging wall and in proximity of the GF trace (see next sections). Alluvial fans of Unit U3 (Late Pleistocene) are formed by prevailing medium grain-sized carbonate gravel, matrix- to grain-supported, with textural and structural characters of debris and hyperconcentrated flows. The unit contains reworked tephra layers and paleosols developed on pyroclastic materials ascribed to the Late Pleistocene Neapolitan volcanoes (Amato et al., 2018; Leone, 2016). In general, they show a better organization than the deposits of older generations. In some cases, they are deposited within the entrenchment of older fans, forming terraced deposits. The U3 alluvial fans slope gently down to the Alife plain and the distal fans interfinger with the alluvial deposits of the Volturno River, containing the ~40 ka-old Campanian Ignimbrite (Valente et al., 2019).

Unit U4 is made up of Holocene small alluvial fans formed by heterometric, calcareous gravel and sandy gravel. In the distal part, the fan gravels have decreasing clast size, are supported by sandy matrix, and contain paleosols and archaeological remains.

### 4.1.2 Geometry, kinematics and fault scarp morphology

The GF extends from the Piedimonte Matese town to the south-western slopes of M. Monaco di Gioia, for a total length of ~11.5 km (Fig. 4). The northern San Potito fault section is ~5.5 km long. The southern Castello di Gioia fault section is about 3.5 km long. A ~2 km-long subdued fault scarp connects the San Potito and Castello di Gioia sections across the M. Erbano embayment. A sharp bend from NW-SE to ~W-E directions connects the GF with the Ailano – Piedimonte Matese fault. To the south, a ~1 km-wide gap separates the GF from a SW-dipping normal fault, with poor geomorphic evidence of recent activity, that delimits to the west the small M. Acero carbonate ridge (Fig. 2).

### San Potito fault section

The average strike of the San Potito section is N130°. At a detailed view, the fault trace is formed by longer strands striking NW-SE, separated by short strands where the fault bends to ~W-E. Striated fault planes at the base of carbonate scarps crop out discontinuously in the northern, central and southern part of the fault trace (Fig.s 5a, b and c). The slip vectors plunge to

SW and SSW, determining normal dip-slip to left-lateral normal-oblique kinematics; close to the southern part of the fault section (Fig. 4, south of Fig. 5c) the fault bends to E-W and the kinematics is normal with right-lateral component (WSW-plunging slip vectors) (see Tab. S3 of the supplementary material for data points of structural data).

The fault displaces Late Miocene siliciclastic deposits, containing a lithological unit of varicoloured clays (sensu Vitale and Ciarcia, 2018), against the Cretaceous or Late Triassic-Jurassic carbonate bedrock (Fig. 4 and Plate S1). The geological

displacement is likely very large, but the amount of vertical throw cannot be precisely constrained, as the Miocene siliciclastic rocks unconformably overlie Cretaceous or Jurassic carbonate rocks, indicating that the pre-Miocene bedrock was already faulted, and the morphology already articulated at the time of the Miocene siliciclastic sedimentation. Pre-Miocene faulting occurred on faults striking from SW-NE to ~W-E. It is also likely that NW-SE-striking normal faults, including the GF itself, were already formed before the Miocene siliciclastic sedimentation, as testified by the different pre-Miocene bedrock in the

hanging wall and footwall blocks of the GF fault (Cretaceous in the hanging wall, Late Triassic-Jurassic in the footwall; see geologic map and sections in Plate S1). Therefore, the total displacement was accumulated both before and after the Late Miocene. A throw of >225 m, postdating the base of the Late Miocene siliciclastic deposits, can be estimated in the southern part of the San Potito fault section (cross section 3 in Fig. 4). In the same place, Early (?) Pleistocene slope breccia (sd1, close to cross section 3 in Fig. 4) hangs 75-100 m upslope the fault trace, in the footwall, thus indicating that throw >75-100 m was

accumulated in the Quaternary. In the central part of the fault, the total geological throw is estimated to be > 1000 m (cross section C-C' in Plate S1; Fig. 4).

Sharp changes in slope between the carbonate bedrock and the continental cover, and fault scarps on both bedrock and continental deposits have been observed in numerous places along the fault trace. Particularly informative is the site located ENE of San Potito Sannitico (see Fig. 2a), where the fault forms a 5-to-7 m-high scarp, measured on the 1-m resolution LiDAR

DEM, on Upper Pleistocene slope colluvial deposits of Unit sd2 (Fig. 6). The fault plane on Triassic bedrock crops out in the western side of the map of Fig. 6a, and shows slickenlines plunging to SSW (Fig. 6b). Both the western and eastern sides of the fault scarp are disturbed by anthropogenic modifications (A in Fig. 6a), and have been discarded from scarp measurements. The central, undisturbed part of the scarp is located just along the projection of the measured bedrock fault plane and is formed entirely within Unit sd2. There is no lithology variation across the scarp, suggesting that differential erosion is not a plausible

mechanism for scarp formation. Moreover, a small valley crossing the scarp is only slightly incised within Unit sd2 downslope the scarp and much more incised within Unit sd2 upslope the scarp. All these features suggest that the scarp formed due to normal faulting. The slope deposits of Unit sd2, observed in a man-made cut down-slope the scarp, contain a buried paleosol dated to 43.3 – 42.2 ka (cal. BP; Tab. 2; Fig. 6c), indicating that the upper part of the succession accumulated during the Last Glacial Maximum (LGM), when large volumes of coarse-grained sediments piled up at the piedmont of the Apennine ridges

(e.g., Giraudi and Frezzotti, 1997). Considering that the dip of the LGM deposits is nearly coincident with the dip of the topographic surface (profile 7 in Fig. 6d), it is likely that the fault offset the topographic surface and the underlying stratigraphy after the LGM sedimentation (i.e., post-LGM fault scarp).

Figure 7 shows topographic profiles across the GF fault scarp extracted from the 1 m-resolution LiDAR DEM (profiles 1 to 10, location in Fig. 4). Table 3 summarizes the throws measured on topographic profiles across fault scarps. In particular, east of Piedimonte Matese, at the foot of the M. Olnito slope (Fig. 4), late Quaternary colluvial deposits of unit sd2 and a small valley (profile 1 in Fig. 7) are faulted and uplifted upslope (NNE) of a ~3 m-high fault scarp, suggesting recent faulting, possibly after the LGM for similarity with the area of Fig. 6. A ~36 m-high fault scarp is measured along profile 3, with gravels of Middle Pleistocene unit U1 in both hanging wall and footwall that indicates post-U1 (i.e., post-450 ka) faulting. The same fault trace forms a ~4 m-high fault scarp, across a ~15 m-wide zone, on the floor of a Holocene alluvial plain (profile 4) that can be the result of cumulated late Holocene surface faulting events. A long profile built along the river bed close to profile 4 shows a knickpoint across the fault trace ("knick zone" in Fig. S1 of the supplementary material). The knickpoint height (elevation difference between crest and toe) is 3-4 m. By considering the long profile slope and the fault dip, the throw is on the order of 2-3 m. This is less than the cumulated throw measured on profile 4. This is consistent with profile 4, as the knickpoint should have registered younger slip compared to the top of the Holocene alluvium, reinforcing the hypothesis of Holocene surface faulting. South of the fault scarp illustrated in Fig. 6, two synthetic splays form a ~14.5 m-high cumulative fault scarp (profile 9). The upper splay forms a ~12 m-high scarp (elevation difference between crest and toe), interpreted as due to ~8 m of vertical displacement of the topographic surface. The similarity of the slope angle in the hanging wall (on U1 deposits) and footwall (on carbonate bedrock) suggests that the ~8 m of throw accumulated after the sedimentation of U1. The lower splay offsets vertically of ~6.5 m the topographic surface on top of U1. Therefore, the cumulative scarp seems to have formed due to post-U1 (i.e., post-450 ka) faulting. Further to the south, the two splays merge into a single trace forming a 2-3 m-high scarp for which we have no age constraints (profile10); a 2.5 m post-LGM throw can be tentatively inferred due to its small size and for comparison with the scarps previously described.

**Castello di Gioia fault section**

The Castello di Gioia section strikes on average N155° at the foot of the Mt. Monaco di Gioia slope, with a simple, linear fault trace. Striated fault planes crop out in the central part of the fault trace (Fig.s 4 and 5d). The slip vectors plunge to WSW, determining normal kinematics with slight right-lateral component; in one site, both S- and SW-plunging slip vectors have been observed.

The fault displaces Late Miocene siliciclastic deposits against Late Triassic dolostones, with vertical geological separation exceeding 1000 m (Plate S1, section F-F'). It is likely that such a large total geological displacement was accumulated both before and after the Late Miocene, similar to the San Potito fault section. Near Castello di Gioia, the Early (?) Pleistocene slope breccia of Unit sd1 hangs ~ 80-100 m in the footwall of the fault, suggesting that > 80 m of throw accumulated in the Quaternary (Fig. 7, profile 12). However, the site crossed by profile 12 is disturbed by an unquantified amount of erosion, preventing reliable throw estimates.

In the M. Erbano embayment, NW-SE-striking Quaternary normal faults displacing the pre-Quaternary bedrock and the Early
(?) Pleistocene sd1 slope breccia has been mapped. The fault crossed by profile 11 (Fig. 7) is characterized by a ~18 m-high
fault scarp with sd1 breccia in both hanging wall and footwall, with no additional evidence of Late Quaternary activity.

The connection between the Castello di Gioia and San Potito fault sections is poorly constrained and is inferred here (geologic
map in Plate S1) on the basis of displacement of pre-Quaternary bedrock and escarpments on the Early (?) Pleistocene sd1
slope breccia between the terminations of the Castello di Gioia and San Potito fault traces.

Fault scarps on slopes covered by continental deposits of the units U1 and sd1 have been observed in the central part of the
Castello di Gioia section. The scarps have heights on the order of 4 to 7 m and interrupt the regular, gently-dipping slopes of
the M. Monaco di Gioia piedmont (profiles 13, 14 and 15 in Fig. 7). This indicates that the scarp formed after the sedimentation
of unit U1, and possibly in more recent times.

The Castello di Gioia fault section has been mapped up to the Faicchio area, where geologic and geomorphic evidences of
Late Pleisocene – Holocene faulting dissipate. Here the M. Monaco di Gioia mountain front, in the footwall of the Castello di
Gioia fault section, suddenly bends from NW-SE to WNW-ESE and, further to E, to WSW-ENE, for a total length of ~ 8 km
(Fig. 2). The mountain front is controlled by a S-dipping normal fault that appears to be capped by Early-Middle Pleistocene
deposits of the sd1 and U1 units. Further to the SE, a SW-dipping normal fault can be inferred along the SW slope of the M.
Acero Jurassic-Cretaceous carbonate ridge (Fig. 2; Plate S1). A ~1.2 km-wide step-over separates the M. Acero fault from the
Castello di Gioia fault section. We could not observe fault planes or scarps or other evidence of Late Pleistocene – Holocene
activity along the M. Acero fault trace.

## 4.2 Late Pleistocene – Holocene surface faulting

### 4.2.1 The San Potito site on the Gioia Sannitica Fault

In the central-southern part of the San Potito fault section, the fault is formed by two parallel splays (Section 2 in Fig. 4). The
eastern splay separates the continental deposits from the dolomitic bedrock, while the western splay is within the continental
deposits. Across the western splay, a ~40 m-long cut along a dirty road was accurately cleaned and logged for analysing two
fault zones, one located in the footwall of the splay (Fault zone 1, Fig. 8) and one located on the main trace of the splay (Fault
zone 2, Fig. 9; a photographic documentation of fault zone 2 is reported in Fig. S2 of the supplementary material).

Fault zone 1 (Fig. 8) is formed by two SW-dipping normal fault strands, F1 and F2. F2 separates poorly-organized, coarse,
white gravels in the footwall (unit 1, corresponding to unit U1 of the geologic map in Fig. 4 and Plate S1), from a succession
of stacked, matrix-rich colluvial gravel units in the hanging wall (units 4 to 7, belonging to unit sd2 of the geologic map in
Fig. 4 and Plate S1).

Unit 1 contains a 20 cm to 40 cm-thick layer of yellow (lower part) to yellowish grey (upper part) distal tephra (unit 2)
containing yellowish glass aggregates, poorly preserved leucite-bearing grey pumice fragments, rare loose sanidine crystals

that are dated to 564.5 ± 2.1 ka (sample S-227; Tab. 1; Tab. S1), brown and minor green clinopyroxene and few dark mica grains, and a lithic fraction formed by leucite-bearing lava fragments.

Colluvial units 4, 5 and 6 are faulted in the hanging wall of fault F2. The youngest faulted unit is unit 6, the fine-grained matrix of which has been dated 1,377-1,126 yrs BCE (Before Current Era, calendar calibrated age; sample S227_5-1, Tab. 2).

Fault zone 2 (Fig. 9) is formed by three main SW-dipping normal fault strands (F3, F4 and F5), a NE-dipping antithetic fault
(F6) and several secondary synthetic shear planes. Fault F3 separates organized, coarse, white gravels in the footwall (unit 1, corresponding to unit U1 of the geologic map in Fig. 4 and Plate S1), from a succession of colluvial gravels (unit 2) covered by a thick body of brown distal tephra (unit 3). The tephra contains whitish to grey glass aggregates and rare, very altered pumice fragments with leucite crystals, loose crystals of abundant sanidine and minor green and brown clinopyroxene grains, and a lithic fraction with limestone and rare leucite-bearing lava fragments. The sanidine crystals have been dated to 508.5 ±
0.9 ka (Tab. 1; Tab. S1). In the hanging wall of faults F3-F4 there is a succession of colluvial deposits belonging to unit sd2 of the geologic map (Fig. 4 and Plate S1).

The tephra unit is trapped within the main fault zone, between F3 and F5, and the original geometry cannot be reconstructed due to synthetic and antithetic faulting and erosion. The top of the tephra unit, in the hanging wall of fault F4, is altered by a paleosol buried by coarse, non-organized, white gravels (unit 4), likely deriving from erosion and colluviation of unit 1. The
buried paleosol has the appearance of a volcanic soil with andic properties. In detailed analysis (see Text S1 and Tab. S4 for a detailed description), the paleosol shows weak andic properties and poor pedogenetic evolution, suggesting a young age (Late Pleistocene – Holocene), and possible disturbance by erosional ad colluvial processes. Two paleosol samples, close to each other, have been collected for radiocarbon dating (samples S228_3 and SPOT-1 in Fig. 9b and Tab. 2). One sample (S228_3) was analysed by the CEDAD laboratory for dating the bulk organic fraction, which gave ages of 6,510 – 6,250 yrs BCE (Tab.
2). In order to check if possible contamination by young carbon can have affected the results, a second sample (SPOT-1) was collected and analysed by the Beta Analytic laboratory for dating the: bulk organic fraction, alkali soluble fraction, and alkali insoluble fraction (see section 3.2). The analysis gave no sufficient alkali soluble organics for dating; the dating of the bulk organic fraction gave ages ranging from 5,363 and 5,216 yrs BCE, and the dating of the alkali insoluble fraction gave ages ranging from 9,983 and 9,563 yrs BCE. There is a large variability in the results, possibly due to contamination with young
carbon for the youngest ages, disturbance due to colluviation, and duration of the pedogenic process. Nevertheless, all the results converge towards a Holocene age of the paleosol, possibly developed during the early-middle Holocene thermal optimum (e.g., Giraudi et al., 2011), in agreement with the results from paleosol analysis (Text S1 and Tab. S4).

The paleosol and the overlying unit 4 are faulted by F4 and F5. In the hanging wall of F5, a sequence of colluvial units crops out. The lack of organic material prevented any attempt of dating unit 4. Part of unit 4 is trapped between fault F5 and a
synthetic splay of F5, and is affected by shear fabric (unit 4b). Unit 5 differs from unit 4 due to average larger percentage of boulders, and for the presence of brown, sandy matrix. Radiocarbon dating of the bulk organic fraction within the matrix of unit 5 gave a very young age (post-1,950 Current Era (CE); Tab. 2, sample 228_7-1), which is younger than the overlying unit 7 (see below), suggesting that the result is unreliable due to likely contamination by young carbon.

Two wedge-shaped colluvial units (units 6 and 7) are present between units 4-5 and the modern soil. Unit 6 is formed by coarse, non-organized, whitish gravels, with textural and compositional features very similar to unit 4, suggesting that it formed by colluviation of material from unit 4. The bottom of unit 6 is an erosional surface which cuts units 5, 4 and 4b. Unit 6 seals a synthetic splay of F5, and is faulted by the main F5 strand (Fig. 9b and Fig. S2). The facies and wedge-shaped geometry of unit 6, thickening towards F5, suggest that the sedimentation was sourced by a fault scarp that exposed unit 4 in the footwall of F5, possibly after a surface faulting event (scarp-derived colluvial wedge). The colluvial wedge was then faulted by the main F5 strand. Unit 7 is a wedge-shaped, gravel unit rich in dark brown organic matrix with charcoals dated 1,445 – 1,625 yrs CE (Tab. 2, sample 228_1ter). The wedge thickens sharply in the hanging wall of F5 and seals a steep scarp formed by F5, with unit 4 in the footwall. There is no evidence of faulting of, or shear fabric within unit 7 (see also Fig. S2).

**Tectonic interpretation of fault zone 2 (Fig. 9)**

The tectonic interpretation of the outcrop in Fig. 9 is difficult due to the complex architecture of the fault zone. Nevertheless, the buried paleosol between units 3 and 4 provides insights on the minimum Holocene displacement. The overall properties and numerical ages of the faulted paleosol indicate weak pedogenesis over a short period of time. The soil overlies a much older parent material (>500 ka), suggesting that the faults displacing the paleosol were activated, or reactivated, after a long period without sediment accumulation, and after a period of pre-Holocene erosion. After the onset, or restart, of faulting, a significant hanging wall subsidence and sediment accumulation occurred, as testified by the accumulation of colluvial deposits of unit 4 and younger.

In detail, the paleosol is displaced by F4, F5 and the antithetic fault F6. F6 is an antithetic splay of F3 and displaces fault F4 and the paleosol of about 25 cm (see Fig. S3 of the supplementary material for the restoration of F4, F5 and F6). The paleosol is displaced vertically of >1.2 m by F4 and >1.5 m by F5 (total minimum throw >2.7 m) (Fig. S3b).

Considering the geometric and sedimentary features of units 6 and 7, and their relations with fault F5, two surface faulting events can be hypothesized (see Fig. S4 of the supplementary material for a restoration of the surface faulting events).

The most recent surface faulting event (MRE) should have formed a free face exposing unit 4 (Fig. S4, stage 5). Unit 7, which covers the steep fault scarp without evidence of shear (see Fig. S2), should have deposited shortly after the surface faulting event. Considering that the maximum thickness of unit 7 measured close to the fault is ~30 cm, a coseismic surface throw between 30 and 60 cm can be hypothesized (50 cm of vertical displacement in Fig. S4, stage 5).

The penultimate surface faulting event (PE), should have formed a free face on unit 4 that sourced unit 6 (Fig. S4, stages 1 to 3). After PE, the vertical displacement of which is hypothesized to be of the same size of MRE, it seems necessary to consider: a) a first period of fault-parallel erosion in the hanging wall of the reactivated F5 that cuts into units 5, 4 and 4b, probably accompanied by partial degradation of the exposed free face (Fig. S4, stage 3a); and b) a second period during which the colluvial accumulation from the partially degraded fault scarp forms unit 6 (Fig. S4, stage 3b).

A ~1 m-high degraded, and retreated scarp across the F3-F5 fault zone is visible on a detailed topographic profile of the ground surface (Fig. 9c), supporting the interpretation of surface faulting.

### 4.2.2 The Sant'Angelo d'Alife trench site on the Ailano – Piedimonte Matese Fault

A small trench was dug across the southern part of the Raviscanina fault section (see Fig. 2 for location) in order to build upon previous work (Boncio et al., 2016) and obtain additional age constraints on the APMF activity.

The trench crosses a fault zone formed by a main fault (F1) and a synthetic splay (F2) (Fig. 10). Fault F1 separates cataclastic dolostone in the footwall (unit 1) from high-strength, light grey-to-light brown tuff pervasively cut by fault-parallel veins of calcium-carbonate concretions (unit 2). Unit 2 is truncated upwards by an erosional surface and covered by reddish-brown, clayey colluvium containing sparse carbonate clasts (unit 4) faulted by F1 and F2. Radiocarbon dating of two samples from unit 4 gave Holocene ages, but with significantly different dates, varying from 770 - 660 yrs CE (sample C1_D-E; Fig. 10d)

to 905 – 805 yrs BCE (sample C8_D-W; Fig. 10b), indicating large age uncertainty, possibly due to contamination by young carbon in sample C1_D-E. Unit 4 is covered by a wedge-shaped colluvial unit formed by light brown silty-sand matrix with abundant cataclasite clasts deriving from the F1 fault rock (unit 5). The matrix of unit 5 has been dated 1,270 – 1,385 yrs CE (samples C3_D-E and C7_D-W; Tab. 2). Unit 5 is not faulted. Possibly, this unit formed after the fault rock was exposed to weathering and erosion due to a surface faulting event (scarp-derived colluvial wedge).

Fault F2 displaces vertically the bottom of unit 4 by ~30 cm. In the hanging wall of F2, unit 4 covers a 188.8 ± 3.0 ka old colluviated tephra unit (unit 3) made up of very altered whitish to yellowish pumice fragments, abundant loose sanidine crystals and rare green and brown clinopyroxene crystals, together with several opaque grains in the finer fraction (sample C9-DO; Tab. 1; Tab. S1). Unit 3 is colluviated, therefore it is younger than the obtained ages.

The total, minimum vertical displacement accumulated by unit 4 is estimated to be ~0.8 m, corresponding to the maximum

vertical thickness of unit 5 (~0.5 m; the space created by surface faulting and filled by the colluvial wedge) plus the 0.3 m of displacement on F2.

### 5 Discussion

### 5.1 Architecture and kinematics of the GF and the entire SMF system

The detailed study of the GF appears to provide a fundamental piece of evidence to understand the overall geometry, kinematics

and activity of the entire SMF system. The characterization of the 30 km-long SMF system is in turn crucial for drawing a complete tectonic picture of the Matese area, were several historical earthquakes are still without an associated seismogenic fault (Fig. 1). In order to understand the implications that our findings have within the regional context of the Matese tectonics, in Fig. 11 (a and b) the GF is mapped together with all the other known Matese normal faults (from Di Bucci et al., 2005; Galli and Naso, 2009; Boncio et al., 2016; Ferrarini et al., 2017; Galli et al., 2017; Valente et al., 2019). In particular, for the SMF

system reported in Fig. 11b, the results from our detailed field mapping are integrated with data of comparable detail published by Boncio et al. (2016) for the Ailano – Piedimongte Matese fault (their Fig. 9). Valente et al. (2019) mapped several short, certain or inferred, mostly from geomorphological analysis, secondary faults within the Alife basin, in the hanging wall of the

SMF. Not all the faults mapped by Valente et al. (2019) are reported in Fig. 11, as the following discussion will focus on the main fault. This does not exclude that other secondary faults can be present in the hanging wall of the main SMF.

Once integrated with previous data, the results of our fault mapping show that the geometry of the SMF system is complex (Fig. 11b), with NW-SE, E-W and WSW-ENE fault sections. The northern Raviscanina section strikes on average NW-SE, and bends to ESE and then to E-W in the southern portion, approaching the Piedimonte Matese section. Internally, the Raviscanina section bends locally to E-W, and intersects with short E-W to SW-NE-striking faults in the hanging wall. The Piedimonte Matese section strikes from E-W to WSW-ENE. The San Potito and Castello di Gioia sections of the Giosia

Sannitica fault strikes on average NW-SE. The largest part of the SMF system strikes NW-SE.

A summary of fault geometry (strike, dip) and kinematics (slip vector trend) is presented in Fig. 12 (numerical data in Tab. S3). All the fault sections are represented in the diagrams of Fig. 12, but unfortunately the sampling of fault slip data is not homogeneous along the system, depending on the different exposure conditions. Therefore, in the diagrams in Fig. 12 there can be some bias due to under-sampling of poorly-exposed faults. In order to avoid over-sampling, for well-exposed outcrops,

only one representative measurement, or an average, has been plotted. The most under-sampled fault section is the ~E-W Piedimonte Matese section, due to the poor exposure of the fault plane, which is covered by alluvial fan sediments for most of the section length. The prevailing measured strikes range between 115° and 140°. Variations to E-W strike (95-105°) are measured at a smaller scale within major NW-SE faults (Fig. 12a). Dip angles are mostly within the range typical of high-angle normal faults (60-80°, Fig. 12b).

The orientation of slip vectors is uneven along-strike the SMF system. The trend of slip vectors varies from 270° to 110°, with prevailing slip vectors plunging to SW (200-240°; 20 out of 52 data) and to SE (110-180°; 15 out of 52 data). SW-plunging slip vectors are consistent with the direction of active regional extension obtained from focal mechanisms and GPS velocity vectors (210-240°, Fig. 12c; Fig. 1). Interestingly, both SW-plunging and SE-plunging slip vectors are recorded on NW-SE, E-W and WSW-ENE faults. This pattern cannot be explained by simple models of normal fault growth, even considering

variations due to slip vector convergence towards the fault center during fault growth (Roberts, 1996; Roberts and Michetti, 2004). It seems more likely that this pattern is the result of a different tectonic model, possibly characterized by repeated reactivations of pre-existing faults. The geometrical pattern of interfering NW-SE and nearly E-W faults has been observed by previous authors to be common in the Matese area, and interpreted as the result of interaction between newly-formed Quaternary NW-SE faults and reactivated pre-existing E-W Mesozoic faults (e.g., Calabrò et al., 2003). Even in this context,

the presence of SE-plunging slip vectors is not easily explained. A possible explanation has been proposed by Boncio et al. (2016), also thanks to structural analysis and relative chronology of slip vectors (e.g., Fig. 16 in Boncio et al., 2016), and by Amato et al. (2017) based on tectono-stratigraphic arguments. According to those authors, SE-plunging slip vectors were recorded mostly during the NW-SE extensional phase that opened the Garigliano graben in the Early-Middle Pleistocene, when new NE-striking normal faults formed, and pre-existing Jurassic E-W normal faults, and pre-Miocene or pre-Quaternary NW-

SE normal faults have been reactivated with left-lateral normal-oblique kinematics. The mutual relationships among fault segments with different orientation and the overall geometry of the system probably evolved through time, depending on the

dominant extensional direction. The present overall architecture shown in Fig. 11b was likely achieved during the latest stage of extension, dominated by NE-directed stretching, operating since Middle Pleistocene and still active.

## 5.2 Fault activity and slip rates

Following the approach proposed by Faure Walker at al. (2021), in Fig. 12b the constraints on fault activity are preserved at the level of the single fault trace, which is the primary level of data collection in the field. The extent of a trace depends on the distance over which the criteria for determining the activity and location certainty remain the same. This is a useful way to avoid propagating uncertainties along the entire fault, to identify traces where the lack of constraints on very recent activity might indicate poor exposure, or less active portions of the fault system, or simply to identify traces where further investigations should be planned to reduce uncertainties. This is also a way to help seismic hazard modellers who use fault data to perform more complete uncertainty analyses. Fault activity classes are divided considering: a) evidence of displacement of dated Late Pleistocene – Holocene sediments, which is the strictest constraint for fault activity; b) post-LGM fault scarps; c) evidence of faulted Middle Pleistocene sediments; and d) evidence of Quaternary activity, intended as a generic control over the Quaternary evolution of the area, without further constraints. Post-LGM fault scarps are scarps, clearly related to the presence of a fault, that interrupt the regularity of the slope and that can be explained only by invoking a fault displacement of the topographic surface (i.e., the original topography can be reconstructed after fault restoration) occurred after the LGM (i.e., after 15±3 ka ago; Giraudi and Frezzotti, 1997). Post-LGM fault scarps have been widely used in the Apennines to derive fault activity and slip rates (e.g., Roberts and Michetti, 2004). Nevertheless, the interpretation of a post-LGM fault scarp as such is not always straightforward, often due to the lack of age constraints for the displaced topographic surface. In this work, the classification was done only after calibration with the site in Fig. 6, where the fault scarp is clearly post-LGM. All the fault traces classified as post-LGM share with Fig. 6 similar morpho-tectonic evidence, including the size of the scarp (i.e., scarps of the same order of height).

The displacement of the dated sedimentary layers and the vertical displacement reconstructed across fault scarps were used to derive slip rates, which have the same chronologic constraints of the host fault trace. The slip rates were obtained by measuring the throw rate, and then converting it into slip rate by using the fault dip measured in the site or taken from the closest outcrop. Therefore, the obtained value is the along-dip slip rate, which equals the net slip rate only for pure dip-slip faults. We feel that this approximation is acceptable considering that: there are uncertainties in the measurements; the slip vectors are not always measurable; and the measured rake angles at or close to the sites of slip rate measurement are close to dip-slip. The results are summarized in Tab. 3, and represented graphically in Fig. 11b. In order to have a complete picture of the entire SMF system, we included also a number of previously published punctual slip rate data (i.e., Boncio et al., 2016). Other slip or throw rate estimates have been proposed by Cinque et al. (2000, ~1 mm/yr; >0.5 mm/yr in a related paper by Galadini et al., 2001), and more recently by Valente et al. (2019; inferred displacement rate >0.2-0.3 mm/yr at the boundary of the Raviscanina - Piedimonte Matese mountain front). Unfortunately, we could not incorporate these data, as the precise criteria used in obtaining

them are not described, and the values cannot be assigned precisely to a point or to a specific fault trace. Assigning slip rate data to the precise points of measurement is important, as it is recognised that slip, and consequently slip rate, can vary significantly along-strike, and detailed along-strike slip-rate profiles are needed for using them for seismic hazard applications (e.g., Faure Walker et al., 2019).

Two fault traces are constrained by displacement of dated Late Pleistocene – Holocene sediments: the synthetic hanging wall

splay in the central portion of the San Potito section, and the southern portion of the Raviscanina section, both investigated in this work. The synthetic splay of the San Potito section is distant 110-130 m from the main fault trace, and likely joins with the principal fault plane at shallow depths (see section 2 in Fig. 4). Therefore, the activity of the splay provides insights on the activity of the main fault. The Holocene buried paleosol, displaced vertically >2.7 m by the fault strands F4 and F5 (Fig.9 and Fig. S3), provide constraints on the minimum slip rate. By considering the oldest and youngest paleosol ages obtained from

the different dating analyses (11,932 and 7,165 yrs cal. BP, respectively; Tab. 2), the minimum obtained slip rate ranges from 0.24 mm/yr to 0.4 mm/yr. The average minimum slip rate is 0.32±0.08 mm/yr.

In the Sant'Angelo d'Alife site, the minimum calculated slip rate is 0.37 mm/yr, obtained by dividing the >0.8 m throw of unit 4 by the largest time window between the age of the faulted unit 4 and the age of the unfaulted unit 5 (time window of ≤2,285 yrs).

The post-LGM slip rates were obtained from the throw measured by restoring the fault scarps (Fig. 7), divided by the time window of 15±3 ka (Giraudi and Frezzotti, 1997; Roberts and Michetti, 2004). The obtained values range from 0.15 and 0.49 mm/yr (Tab. 3). In one case (profile 4) a time window of 10 ka was considered, as the scarp offsets the top of the Holocene alluvium (al unit), providing minimum slip rates of 0.42 mm/yr.

The post-Middle Pleistocene slip rates are minimum values, obtained from the throw measured by restoring the fault scarps

(Fig. 7), divided by the age of the stratigraphic unit beneath the topographic surface. The fault scarps crossed by profiles 13, 14 and 15 along the Caastello di Gioia section (Fig. 7) are very similar, in terms of geometry and size of the scarp, to the post-LGM scarp of Fig. 6, suggesting a likely post-LGM age. Nevertheless, in the absence of clear constraints, we used the conservative time window of <450 ka (younger than Unit U1). The minimum post-Middle Pleistocene slip rates calculated over the entire SMF vary from 0.01 to 0.31 mm/yr.

The along-strike SMF slip rate profile is shown in Fig. 11c. From the profile, a pattern of increasing slip rate towards the Sant'Angelo d'Alife trace and towards the central San Potito trace comes out. However, the profile has several gaps, numerous data are only minimum values, and all these limitations make the interpretation not straightforward. There is a large gap along the entire Piedimonte Matese section. Hence, it is not possible to conclude if a maximum or a minimum of slip rate is expected within that section. This can have important implications in terms of fault segmentation. In fact, a maximum within the

Piedimonte Matese section would imply a full linkage of the faults (e.g., Cowie, 1998), and the potential for a single large rupture along the entire SMF system should be considered in case of earthquake reactivation. On the other hand, a minimum of slip rate would imply a less active fault section, and possibly a barrier to the propagation of the rupture during earthquake reactivation. We cannot exclude neither of the two hypotheses, even though the lack of young scarps similar to those observed

in the San Potito section allows us to prefer the hypothesis of a less active (i.e., minimum slip rate) Piedimonte Matese fault
section. Possibly, the slip rate decreases from the Sant'Angelo d'Alife site towards the eastern end of the Piedimonte Matese
section, and then starts to increase again towards the San Potito site.

## 5.3 Comparison with other tectonic models and implications for time variability of slip rate

An evolutionary model of the SMF system has been proposed by Valente et al. (2019) on the basis of their tectonic
geomorphology study, summarized in section 2.2. In their model, the evolution of the fault system during the last ~600 ka
would have determined a progressive increase in activity along a major E-W-striking fault in the central part of the system.
The main central E-W-striking fault is mapped from the western tip of the fault named Raviscanina fault scarp (RFS in Fig.
11b) to the eastern tip of the Piedimonte Matese fault section (see Fig. 13 in Valente et al., 2019). This active E-W-striking
normal fault is considered an inherited, reactivated fault, which interacted with newly-formed NW-SE-striking normal faults
during the NE-directed Quaternary extension. This process would have determined local extension to be oriented N-S. This
evolution would have determined the deactivation of the NW-SE-striking Ailano fault trace, and the progressive decrease of
activity along the San Potito fault section. The low activity of the San Potito section would be indicated by the overall higher
maturity of the mountain front and higher degree of entrenching of the alluvial fans. The onset of extension along N-S-striking
normal faults, supposed to be in the hanging wall of the main E-W fault, would have favoured the progressive deactivation of
the San Potito fault.

Our results on the Sant'Angelo d'Alife fault trace, at the southern termination of the Raviscanina fault section, characterized
by higher slip rates, agree with Valente et al. (2019). Actually, the Raviscanina section bends progressively from NW-SE to
N100° to E-W direction. This bend occurs close to the central part of the SMF system. On the other hand, our results on the
San Potito section, where we found clear evidence of Holocene faulting and the largest slip rates, contrast with Valente et al.
(2019). In the map of Fig. 11b, derived from detailed field mapping, we cannot recognize a main E-W-striking fault from the
western tip of the RFS and the eastern tip of the Piedimonte Matese section. Therefore, further comparisons with the Valente
et al.'s model are difficult, probably due to the different scales of fault mapping.

The discrepancy between our findings and the conclusions by Valente et al. (2009) on the activity of the San Potito section is
of particular interest. The apparent discrepancy can be due to both: a) the rejuvenation of the mountain front along the Gioia
Sannitica fault in recent times that produced small wavelength scarps detectable only with detailed fault mapping; and b) the
different scale used in Valente et al. (2019) compared to this study. The study by Valente et al. (2019), performed at the scale
of the entire mountain front, was able to identify the first order pattern, characterized by a mature mountain front along a low-
slip rate fault, but was not sufficiently detailed to identify recent fault scarps. If this interpretation is correct, this would imply
a recent reactivation, or acceleration of the fault. This also suggests that tectonic geomorphology at the scale of the mountain
front should be complemented with detailed, local scale fault mapping, in order to catch the details necessary to constrain
variations occurred over short time scales.

In support of the above interpretation, the bedrock and Quaternary geology of the area defined in our filed work indicates that the GF has a long tectonic history that started before the syn-orogenic sedimentation of Late Miocene flysch (different pre-Miocene bedrock in the hanging wall and footwall). The activity continued after the Mio-Pliocene compressional tectonic phase and persisted for the entire Quaternary. The mature geomorphology, indicated by straight-to-concave mountain slopes, poorly faceted front, and entrenched alluvial fans, might be the result of the combined long pre-Quaternary and Quaternary geomorphological evolution and average low-slip rate. Recent resurrection, or acceleration, of fault activity after a period of inactivity or of very low slip rate determined the formation of morphotectonic markers of young fault activity, mostly in the form of metric-scale post-LGM scarps. Faulting of the <12 ka-old paleosol that overlies the > 500 ka-old sediments along a synthetic splay in the San Potito site suggests that reactivation occurred in the Holocene, after a long period of low, or negligible subsidence and erosion in the hanging wall of the Gioia Sannitica fault.

This suggests that for seismic hazard applications, the most appropriate slip rates to be used are those calculated for the Holocene, or post-LGM time window.

## 5.4 Seismogenic potential

The identification of colluvial wedges sedimented in close relation with fault activity (e.g., Fig 9, Fig. S4) and the occurrence of post-LGM meters-scale scarps and smaller scarps on Holocene alluvium (Fig. 7) indicate that the GF is capable of producing surface faulting events during moderate-to-strong earthquakes.

Given the geologic evidence of fault activity, two questions arise for the GF and APMF: how large is the expected earthquake potential? Is there any historical earthquake that can be associated to these faults?

Concerning the earthquake potential, fault lengths and the derived magnitudes estimated from various empirical relationships available in the literature (Wells and Coppersmith, 1994; Pavlides and Caputo, 2004; Wesnousky, 2008; Galli et al., 2008) are summarized in Tab. 4. The obtained values assume that the entire mapped fault length will rupture during the earthquake. The lengths of GF and APMF taken separately are 11.5 km and 18.5 km, respectively. Due to the uncertain activity of the Piedimonte Matese section in very recent times (Late Pleistocene – Holocene), we have considered separately also the Raviscanina fault section (length 11.5 km). The average estimated magnitudes are M 6.2, M 6.2 and M 6.5 for the GF, Raviscanina fault section and entire APMF, respectively. Assuming a rupture of both the APMF and GF (total length of 30 km), the estimated average magnitude is M 6.8.

The data collected in the San Potito and Sant'Angelo d'Alife sites give insights on the possible association of historical earthquakes to the faults (Fig.s 11a and 13); some possible rupture scenarios compatible with dates of historical earthquakes are proposed in Tab. 4. The association of the 1688 large earthquake (M 7.1) can be excluded, as the last surface faulting event on both GF and APMF occurred before 1688, and because the macroseismic epicentre is located much to the south of the GF (Serva et al., 2007). The youngest wedge-shaped colluvial units in the two studied sites can be interpreted as scarp-derived colluvial wedges (CW in Fig. 13), accumulated in the hanging wall of the free face shortly after surface faulting events (e.g.,

McCalpin, 2009). The ages of the CWs are similar in both sites, suggesting that APMF and GF could have been activated during the same event, or during two events close in time. The event(s) is(are) compatible with the 1349 or 1293 earthquakes, or both. The strong 1349 event (M 6.8) has been associated to the Aquae Iuliae fault, located 4.5 km N of the APMF (Galli and Naso, 2009). An activation of the APMF and/or GF during the 1349 event cannot be excluded. The Aquae Iuliae fault is not a long fault (~20 km in Galli and Naso, 2009; 16.5 km in Boncio et al., 2016), and a longer rupture can better match the

estimated magnitude (rupture ~>30 km for M = 6.8). Nevertheless, the damage is distributed mostly to the NW of the Aquae Iuliae fault, and not to the SE as expected in case of an activation of the APMF and/or the GF (Fig. 11a). Therefore, an association of the 1349 earthquake to the APMF and/or GF is uncertain. An activation of the GF and/or APMF during the 1293 earthquake seems plausible. The earthquake is constrained by a few data points, with heavy damage documented 20 km north (Intensity IX) and 18 km south (Int. VIII-IX) of the estimated epicenter (southern Matese; Guidoboni et al., 2019). The

estimated magnitude from historical data is moderate (M 5.8), and an activation of the GF alone, located between the two damaged sites, or the APMF alone, seems plausible. If we assume a reactivation of both the GF and APMF, we should conclude that the macroseismic magnitude in the historical catalogue probably underestimates the true magnitude of the event. Considering that the event is very old, and poorly documented, we consider this hypothesis plausible.

A second, faulted wedge-shaped colluvial unit is present in the San Potito site, beneath the most recent CW (unit 6 in Fig. 9).

Unit 6 is free of organic matrix, suggesting deposition during a cold period. Assuming that the sediments of unit 6 originated from a fault scarp formed before 1293, the only known earthquake sufficiently large to likely produce surface faulting is the 346 earthquake (Galadini and Galli; 2004; Guidoboni et al., 2019). The cold period during which unit 6 was formed could correspond with the cold period of the High Middle Ages (e.g., Giraudi, 2005; see grey areas in Fig. 13). The colluvial unit sourced by the 346 scarp would have been faulted by the 1293 event, with a corresponding inter-event time of ~950 yrs.

If the macroseismic area of the 847 earthquake is larger than that reported in the CFTI5Med catalogue, and if the epicenter is located WSW of the Matese massif, as proposed by Bottari et al. (2020), this earthquake could be considered a possible source of surface faulting in the studied area. We think that the association of the 847 event to the SMF is weak, due to the paucity of macroseismic data. Nevertheless, its age is compatible with the ages of faulting at both the San Potito and Sant'Angelo d'Alife sites. Therefore, the possible association of the 847 event to the SMF suggested by Bottari et al. (2020) cannot be excluded.

It is evident that the earthquake history of the SMF is still fragmentary. If asked to indicate a preferred interpretation, on the basis of all the collected data our preferred interpretation would be a rupture of the SMF during the 1293 event, perhaps with close in time separate ruptures on GF and APMF, or a partial rupture in 1293 (GF) and 1349 (APMF), and a rupture of the entire SMF during the 346 event. The different possible associations proposed in Tab. 4 can help seismic hazard modellers in testing different rupture scenarios. Additional paleoseismological studies could help to reduce the uncertainties.

The seismogenic potential estimated in this work, and the likely association to poorly known, ancient historical earthquakes have obvious implications for seismic hazard of this region. In fact, seismic hazard assessments for practical applications (e.g., building design, civil protection; MPS04, Stucchi et al., 2011) are currently based on historical earthquakes of the last

millennium, with no possibility of including the effects of older events or of faults able to source long-recurrence earthquakes. Therefore, the seismic hazard of the southern Matese area, and west of it, is probably underestimated.


## 6 Conclusions

The 11.5 km-long Gioia Sannitica normal fault (GF), at the foot of the southern Matese mountain front, is an active fault showing evidence of surface faulting during the Late Pleistocene - Holocene (post-LGM fault scarps, faulted colluvial deposits and paleosols). The GF and the 18 km-long Ailano-Piedimonte Matese fault (APMF) form the 30 km-long Southern Matese

Fault system. The average slip rate varies along-strike, as expected, with largest latest Pleistocene – Holocene slip rates ~0.5 mm/yr.

The mature mountain front morphology can be explained by low uplift rates, combined with long geologic history of the fault. The onset of fault activity can be dated back to the early Quaternary, and possibly earlier. The combined presence of small post-LGM fault scarps at the foot of the mature mountain front can be explained by recent resurrection, or acceleration, of

fault activity after a period of inactivity or of very low slip rate. This implies slip rate variability through time. Multiple slip episodes have been constrained between 9,983 BCE and 1,635 CE on the GF, and between 905 BCE and 1,390 CE on the APMF. The two sites are compatible with surface faulting events that could have ruptured the entire Southern Matese Fault system. Activation of the GF and APMF separately is also compatible with the observed displacements. The estimated seismogenic potential for each individual fault are M 6.1 for GF or Raviscanina fault section alone and M 6.5 for the APMF

alone. In case of rupturing of the entire Southern Matese Fault system, the seismogenic potential would be up to M 6.8. The observed displacements on the GF and APMF are compatible with activations during some poorly-constrained historical earthquakes, such as the 1293 (M 5.8), 1349 (M 6.8; southern prolongation of the rupture on the Aquae Iuliae fault?) and 346 earthquakes. A fault rupture during the 847 poorly-constrained earthquake is also compatible with the dated displacements.

## Data availability

All datasets presented in this study are included in the article and in the supplementary material.

## Author contributions

PB contributed to conceptualization, funding acquisition, investigation (field mapping), data curation, data analysis, supervision and validation. EA contributed to conceptualization, investigation (field mapping), data curation and data analysis. VA and PA contributed to investigation (field mapping), data curation and data analysis (mostly Quaternary geology and

geomorphology). PP analysed the tephra layers. ACT analysed the San Potito paleosol. BRJ performed $^{40}$Ar/$^{39}$Ar experiments for sample dating. PB and EA wrote the original draft, all the authors contributed to review & editing of the manuscript.

## Competing interests

The authors declare that they have no conflict of interest.

## Acknowledgements

This work was funded by the Departments DiSPUTer and INGEO, "G. d'Annunzio" University of Chieti, "ex 60%" research funds to P. Boncio. Part of this wok was realized during the PhD project of E. Auciello.

We thank F. Scarciglia (Department of Biology, Ecology and Earth Sciences (DiBEST), University of Calabria) for suggestions on the micromorphological analysis of the San Potito paleosol.

We thank the Editor F.J. Pazzaglia and the Reviewers F. Pavano, E. Valente and F.M. Michetti for the detailed review and for
useful comments.

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

**Table 1**. Summary of $^{40}$Ar/$^{39}$Ar experiments

| Sample (Lon, Lat) | Material | N | | | MSWD | Weighted mean Age (ka) ± 2σ |
|---|---|---|---|---|---|---|
| S228-F2 (14.41820, 41.33636) | sanidine | 12 | of | 19 | 0.94 | 508.5 ± 0.9 |
| S277 (14.41842, 41.33659) | sanidine | 6 | of | 16 | 0.80 | 564.5 ± 2.1 |
| C9-DO (14.29065, 41.35314) | sanidine | 12 | of | 14 | 1.14 | 188.8 ± 3.0 |

Ages calculated relative to 1.1864 Ma Alder Creek sanidine (Jicha et al., 2016) using decay constants of Min et al. (2000)

Uncertainties shown at 95% confidence level

**Table 2**. Summary of radiocarbon dating analyses

| Site / geology | Sample (Lon, Lat) | Material | Method – Lab. | Conventional Radiocarbon Age (BP) | Calendar Calibrated Results (95% prob.) |
|---|---|---|---|---|---|
| San Potito / Buried paleosol | S249-A (14.4106, 41.3403) | Charred material | AMS - Beta Analytic | 38800 +/- 380 BP | 41352 - 40260 cal BC (43301 - 42209 cal BP) |
| San Potito / colluvium | S227_5-1 (14.41842, 41.33660) | Organic sediment | AMS - Beta Analytic | 3000 +/- 30 BP | (89.4%) 1304 - 1126 cal BC (3253 - 3075 cal BP)<br>(6.0%) 1377 - 1348 cal BC (3326 - 3297 cal BP) |
| San Potito / colluvium | S228_1ter (14.41815, 41.33637) | Charred material | AMS - Beta Analytic | 370 +/- 30 BP | Cal AD 1445 to 1530 (Cal BP 505 to 420)<br>Cal AD 1545 to 1635 (Cal BP 405 to 315) |
| San Potito / colluvium | S228_7-1 (14.41813, 41.33635) | Organic sediment (bulk organic fraction) | AMS-Beta Analytic | 104.71 +/- 0.39 pMC | (95.4%) post AD 1950 |
| San Potito / buried paleosol | S228_3 (14.41816, 41.33638) | Organic sediment (bulk organic fraction) | AMS-CEDAD | 7561 ± 60 BP | (95.4%) 6510 – 6250 cal BC (8459 - 8199 cal BP) |
| | SPOT-1 (14.41816, 41.33638) | Alkali soluble fraction | AMS-Beta Analytic | | No sufficient alkali soluble organics for dating |
| | | Bulk organic fraction | | 6320 +/- 30 BP | (50%) 5278 - 5216 cal BC (7227 - 7165 cal BP)<br>(45.4%) 5363 - 5283 cal BC (7312 - 7232 cal BP) |
| | | Alkali insoluble fraction | | 10140 +/- 30 BP | (82.2%) 9936 - 9738 cal BC (11885 - 11687 cal BP)<br>(9.6%) 9724 - 9669 cal BC (11673 - 11618 cal BP)<br>(3.2%) 9983 - 9954 cal BC (11932 - 11903 cal BP)<br>(0.5%) 9570 - 9563 cal BC (11519 - 11512 cal BP) |
| S. A. d'Alife / colluvium | C1_Donia_E (14.29065, 41.35315) | Organic sediment (bulk organic fraction) | AMS-Beta Analytic | 1290 +/- 30 BP | Cal AD 660 to 770 (Cal BP 1290 to 1180) |
| S. A. d'Alife / colluvium | C3_Donia_E (14.29065, 41.35316) | Organic sediment (bulk organic fraction) | AMS-Beta Analytic | 690 +/- 30 BP | Cal AD 1270 to 1305 (Cal BP 680 to 645) and Cal AD 1365 to 1385 (Cal BP 585 to 565) |
| S. A. d'Alife / colluvium | C7_Donia_W (14.29065, 41.35315) | Organic sediment (bulk organic fraction) | AMS - Beta Analytic | 670 +/- 30 BP | Cal AD 1275 to 1315 (Cal BP 675 to 635) and Cal AD 1355 to 1390 (Cal BP 595 to 560) |
| S. A. d'Alife / colluvium | C8_Donia_W (14.29065, 41.35315) | Organic sediment (bulk organic fraction) | AMS - Beta Analytic | 2700 +/- 30 BP | 905 - 805 Cal BC (2855 - 2755 Cal BP) |

**Table 3.** Measured throw and slip rates along the Gioia Sannitica (GF) and Ailano-Piedimonte Matese (APMF) normal faults.

| Code | Long | Lat | Fault | F. section | Site/Profile n. | throw (m) | Dip* (°) | method | Window (ka) | slip rate# (mm/yr) | Source |
|------|------|-----|-------|------------|-----------------|-----------|----------|--------|-------------|--------------------|--------|
| GF1 | 14.3896 | 41.3526 | GF | San Potito | Pr. 1 | 3 | 67 | topog. prof. | 15±3 | 0.21±0.04 | this work |
| GF2 | 14.3973 | 41.3491 | GF | San Potito | Pr. 2 | 6.5 | 74 | topog. prof. | 15±3 | 0.47±0.09 | this work |
| GF3 | 14.3992 | 41.3478 | GF | San Potito | Pr. 3 | 36 | 74 | topog. prof. | <450 (post U1) | >0.08 | this work |
| GF4 | 14.3997 | 41.3470 | GF | San Potito | Pr. 4 | 4 | 73 | topog. prof. | <10 (Holoc.) | >0.42 | this work |
| GF5 | 14.4034 | 41.3436 | GF | San Potito | Pr. 5 | 7 | 80 | topog. prof. | 15±3 | 0.49±0.1 | this work |
| GF7 | 14.4105 | 41.3406 | GF | San Potito | Pr. 7 | 5 | 73 | topog. prof. | 15±3 | 0.36±0.07 | this work |
| GF228 | 14.4182 | 41.3364 | GF | San Potito | S.Potito site | >2.7 | 71 | dated sed. | ≤11.932-7.165 | 0.32±0.08 | this work |
| GF9 | 14.4212 | 41.3338 | GF | San Potito | Pr. 9 | 14.5 | 60 | topog. prof. | <450 (post U1) | >0.04 | this work |
| GF10 | 14.4312 | 41.3254 | GF | San Potito | Pr. 10 | 2.5 | 68 | topog. prof. | 15±3 | 0.19±0.04 | this work |
| GF11 | 14.4544 | 41.3188 | GF | M. Erbano embayment | Pr. 11 | 18 | 41 | topog. prof. | <780 (post sd1) | >0.04 | this work |
| GF13 | 14.4544 | 41.3022 | GF | Castello di Gioia | Pr. 13 | 6.5 | 66 | topog. prof. | <450 (post U1) | >0.02 | this work |
| GF14 | 14.4572 | 41.2992 | GF | Castello di Gioia | Pr. 14 | 7 | 75 | topog. prof. | <450 (post U1) | >0.02 | this work |
| GF15 | 14.4598 | 41.2961 | GF | Castello di Gioia | Pr. 15 | 4 | 66 | topog. prof. | <450 (post U1) | >0.01 | this work |
| APMF 1 | 14.2907 | 41.3532 | APMF | Raviscanina | Section D | 50-60 | 76 | geologic section | <350 | ≥0.16 | B16 |
| APMF 2 | 14.2271 | 41.3793 | APMF | Raviscanina | Ps4 | 2±0.1 | 72 | topog. prof. | 15±3 | 0.15±0.03 | B16 |
| APMF 3 | 14.2846 | 41.3537 | APMF | Raviscanina | Ps5 | 2.9±0.6 | 73 | topog. prof. | 15±3 | 0.21±0.04 | B16 |
| APMF 4 | 14.2864 | 41.3534 | APMF | Raviscanina | S.Angelo d'Alife site | ≥0.8 | 73 | dated sedim. | ≤2.285 | >0.37 | this work |
| APMF 5 | 14.2982 | 41.3534 | APMF | Raviscanina | Section F | 65-105 | 73 | geologic section | 230-350 | 0.31±0.1 | B16 |

* Fault dip measured at the site, or taken from the closest fault outcrop; # along-dip slip rate; B16 = Boncio et al. (2016)

**Table 4.** Magnitudes obtained from surface fault length and possible rupture scenarios for historical earthquakes

| Fault | Length (L) (km) | W&C94 NF | P&C04 | W08 ALL | Gal08 | average | possible rupture scenarios (historical earthquakes) | |
|---|---|---|---|---|---|---|---|---|
| | | | | **Magnitude** | | | | |
| Gioia Sannitica (GF) | 11.5 | 6.3 | 6.1 | 6.4 | 6.1 | 6.2 | 1293 | 1293 (macroseismic M underestimated?) |
| Ailano - Piedimonte Matese (APMF) | 18.5 | 6.5 | 6.4 | 6.6 | 6.4 | 6.5 | 1349 (south prolong. AIF rupture?) | |
| Raviscanina section of APMF | 11.5 | 6.3 | 6.1 | 6.4 | 6.1 | 6.2 | | |
| APMF + GF | 30 | 6.8 | 6.7 | 6.8 | 6.7 | 6.8 | 346 (847?) | 346 (847?) |

W&C94 = Wells and Coppersmith, 1994 (Normal Faulting); P&C04 = Pavlides and Caputo, 2004; W08 ALL = Wesnousky, 2008 (ALL kinematics); Gal08 = Galli et al., 2008. AIF = Aquae Iuliae fault.

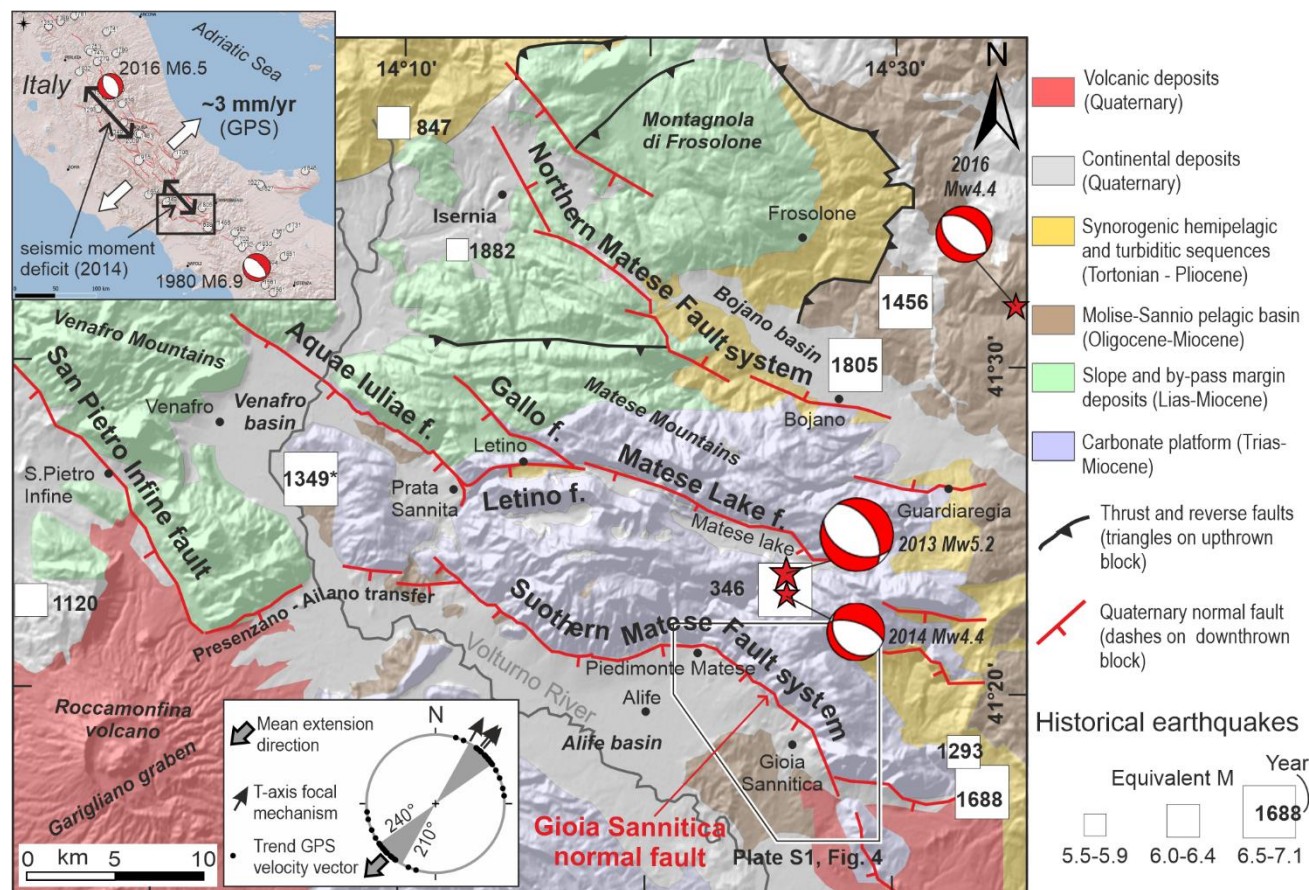

**Figure 1: Simplified tectonic map of the Matese area (southern Apennines) with traces of Quaternary normal faults, epicentres of largest historical earthquakes (from CPTI15, Rovida et al., 2020; from CFTI5Med for events before 1000, Guidoboni et al., 2019; * = epicentre from Galli and Naso, 2009) and locations of the Southern Matese Fault system and Gioia Sannitica normal fault studied in this work. The mean extension direction in the inset is derived from orientations of GPS velocity vectors and T-axes of focal mechanisms (modified from Boncio et al., 2016). Focal mechanisms are from RCMT Catalogue (http://rcmt2.bo.ingv.it/). In location map: double arrows parallel to the Apennines indicate areas of seismic moment deficit (M≥6.5), compared to tectonic (geodetic) strain accumulated in the last 500 years, according to D'Agostino (2014); focal mechanisms refer to M≥6.5 instrumental earthquakes; circles are historical earthquakes with M≥6.0 (CPTI15).**

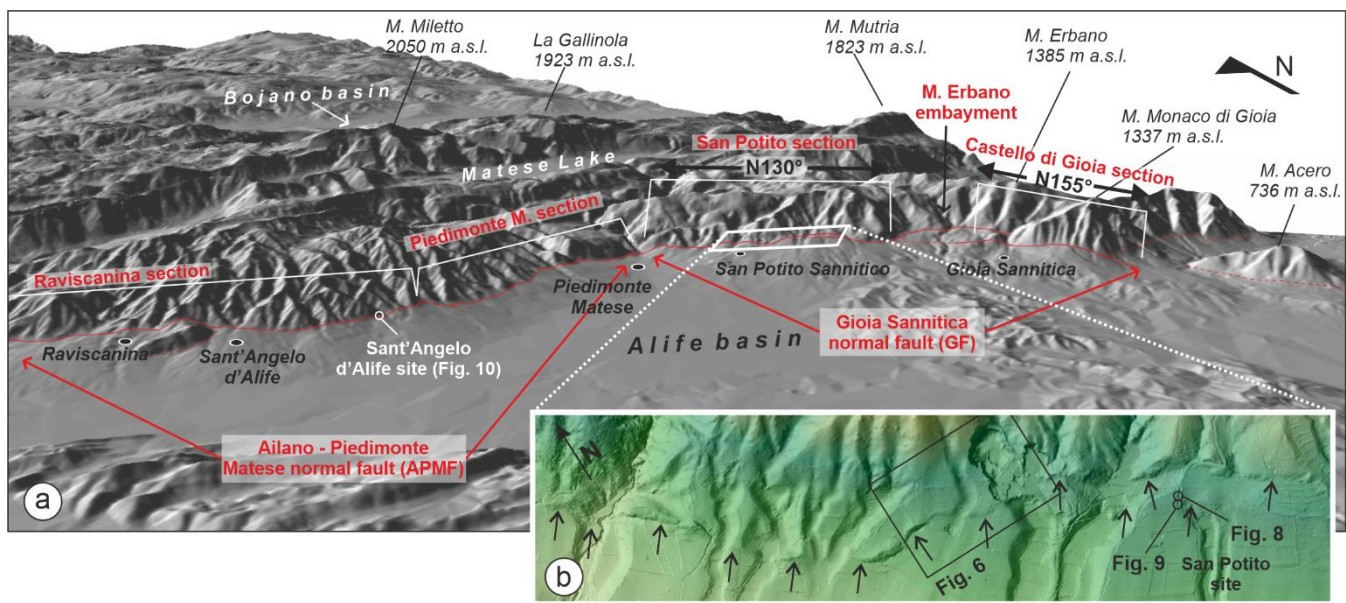

**Figure 2: a) Oblique view of the southern Matese mountain front from a 5-m resolution Digital Elevation Model (DEM) with traces of the Ailano-Piedimonte Matese (APMF) and Gioia Sannitica (GF) normal faults, and b) detail of the GF fault trace, pointed by black arrows, on a 1-m resolution DEM from airborne Light Detection and Ranging (LiDAR).**

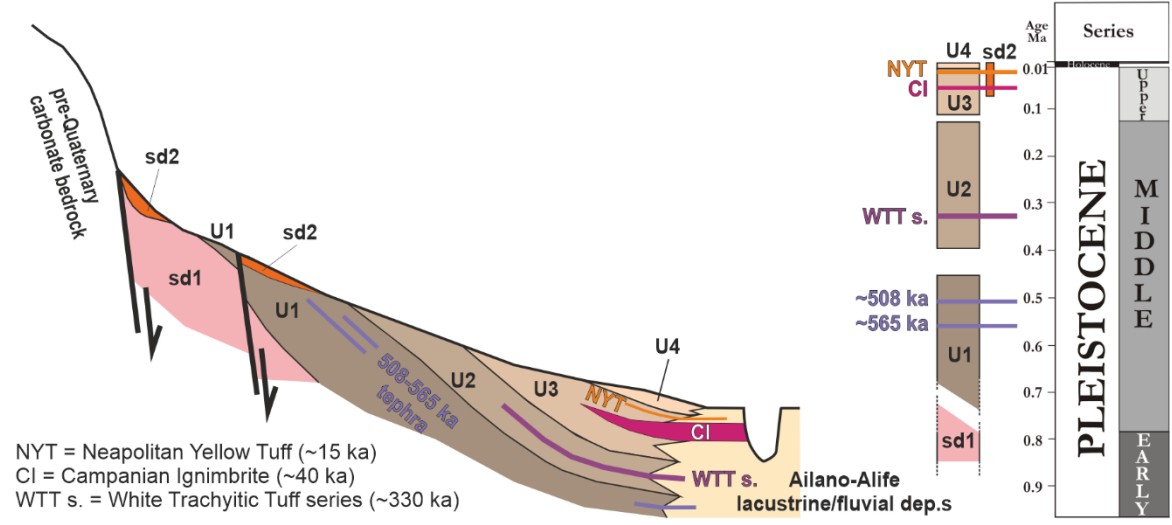

**Figure 3: Schematic morpho-stratigraphic relations and ages of the Quaternary units in the hanging wall of the Gioia Sannitica normal fault (Southern Matese Fault system). The lithology of the units is described in the test and in Plate S1 of the supplementary material.**

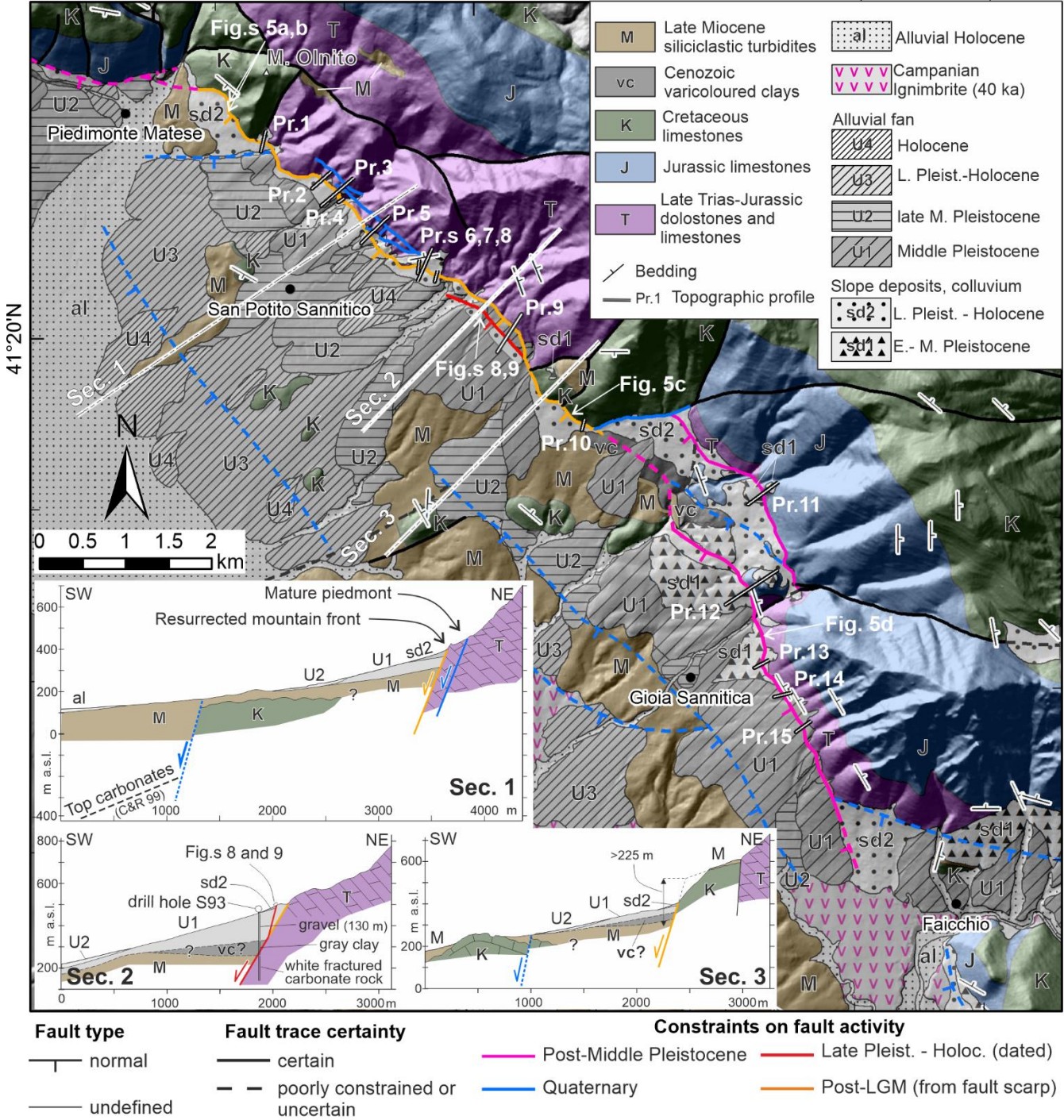

**Figure 4: Geologic map of the Gioia Sannitica normal fault (additional details and cross sections in Plate S1 of the supplementary material). In Section 1, C&R 99 = Corniello and Russo, 1999. Fault traces are represented with different colours indicating different age constraints on fault activity derived from all the data collected in this work (discussed in the dedicated section 5.2).**

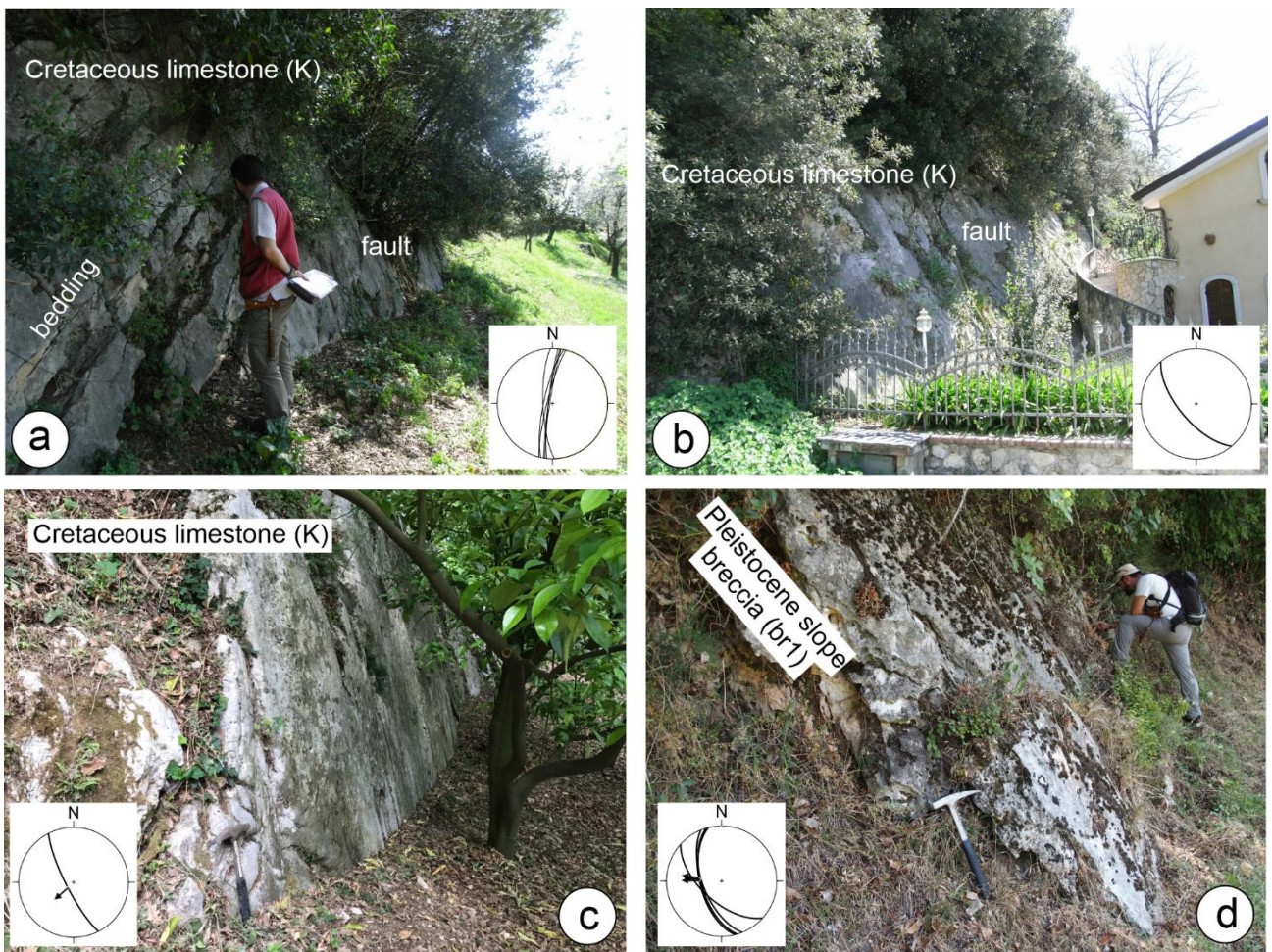

**Figure 5: Field view of the Gioia Sannitica normal fault on carbonate bedrock (a, b, c; San Potito fault section) and on Quaternary breccia (c; Castello di Gioia fault section). Location in Fig. 4.**

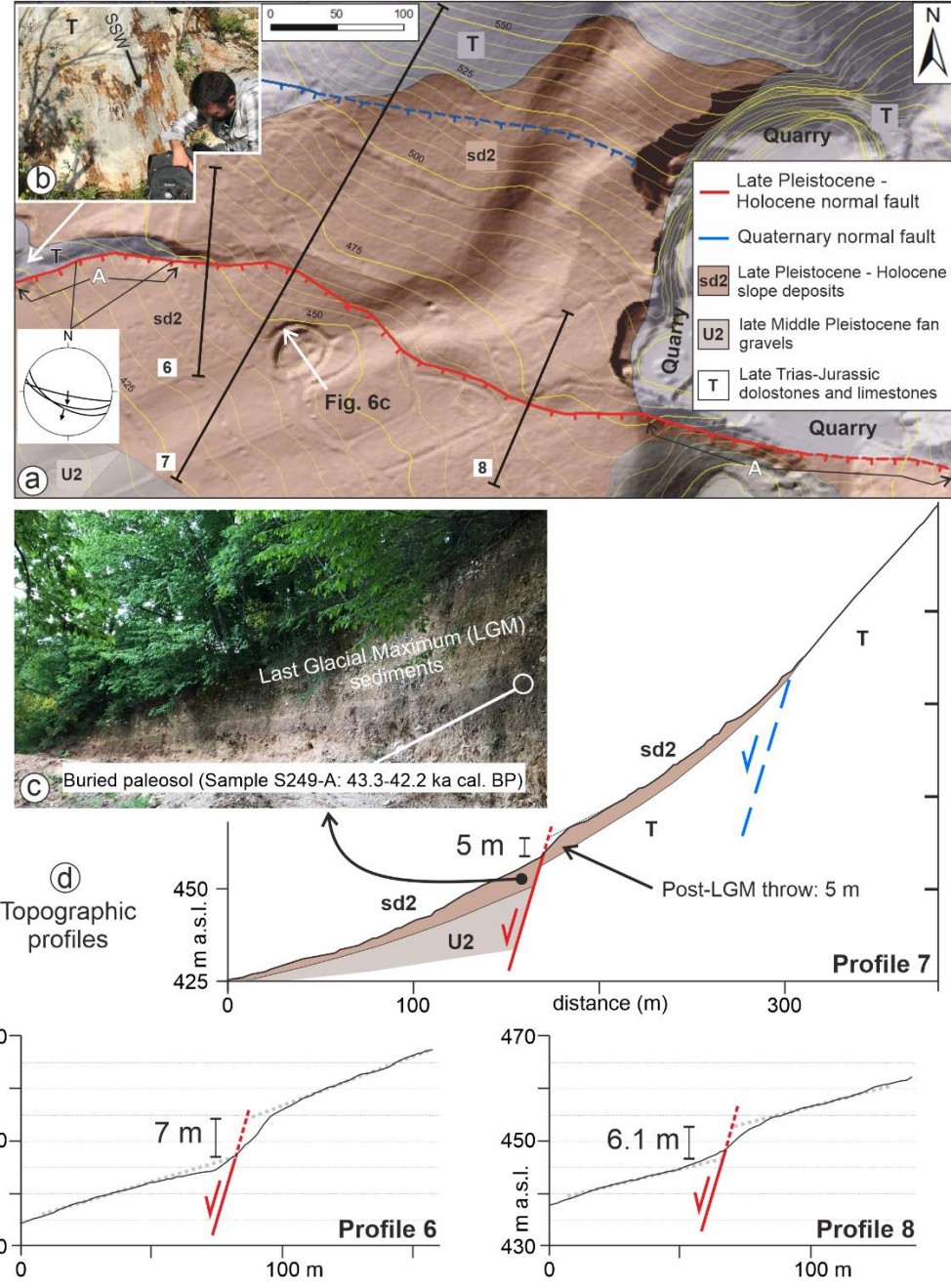

Figure 6: Detail of the geologic map (a), view of the striated fault plane on bedrock (b), field view of the sd2 slope sediments within which a buried paleosol was sampled for radiocarbon dating (c), and topographic profiles across the Gioia Sannitica normal fault trace (San Potito fault section) (d). A: scarp disturbed by anthropic activity (not analysed for scarp measurements). The topography is from the 1-m resolution LiDAR DEM. Location in Fig.s 2b and 4.

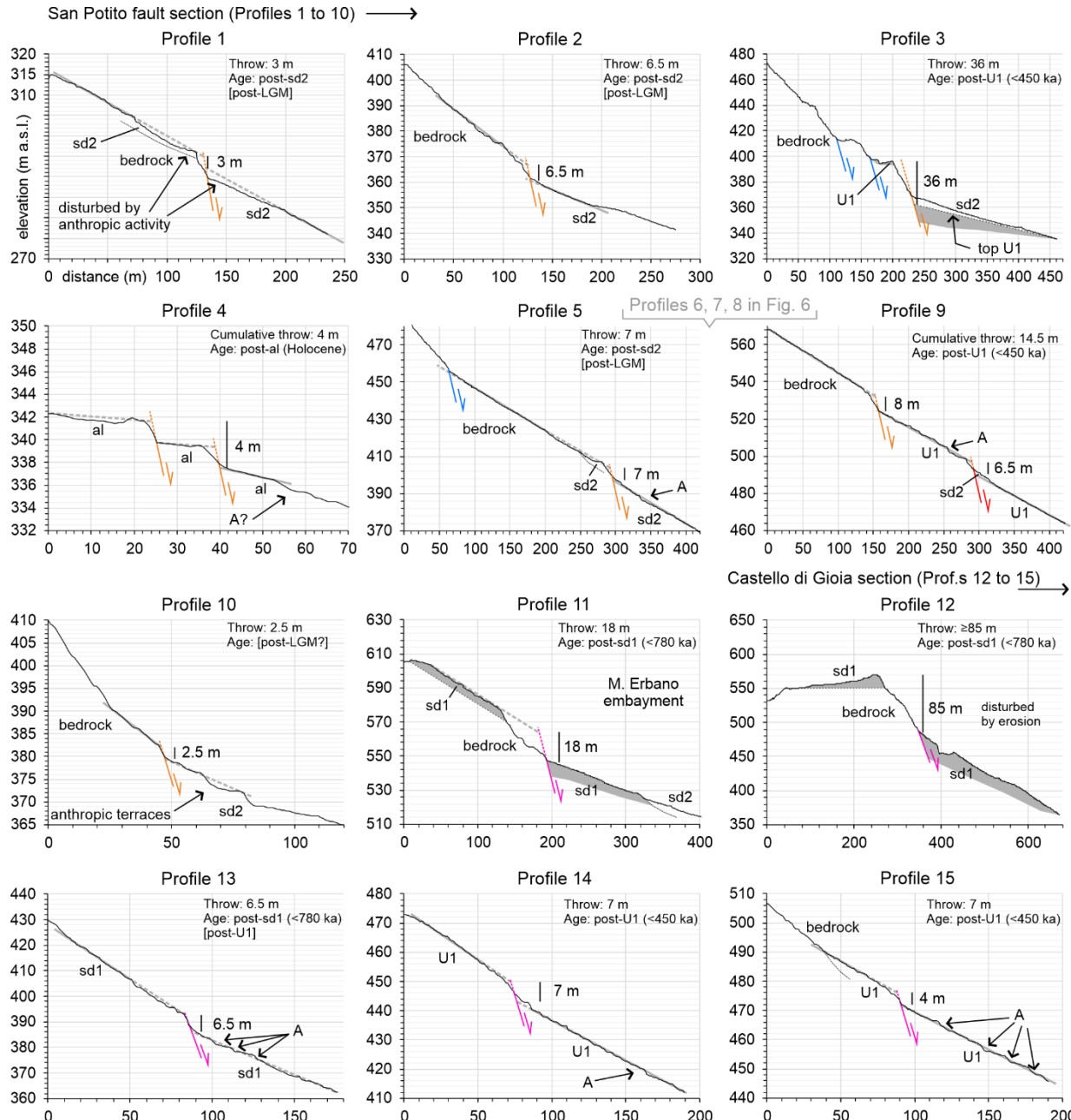

**Figure 7: Topographic profiles across the Gioia Sannitica normal fault trace extracted from the 1-m resolution LiDAR DEM. A: small anthropogenic terraces/excavations (do not modify significantly the mean slope profile). Fault colours are as in Fig. 4 (constraints on fault activity). The preferred age of faulting, reported in square parentheses, for profiles 1, 2, 5 and 10 is based on comparison with fault scarps in Fig. 6; the preferred age for profile 13 is based on similarity with scarps in profiles 14 and 15; the estimated slip rates are reported in Tab. 3. Location of profile traces in Fig. 4.**

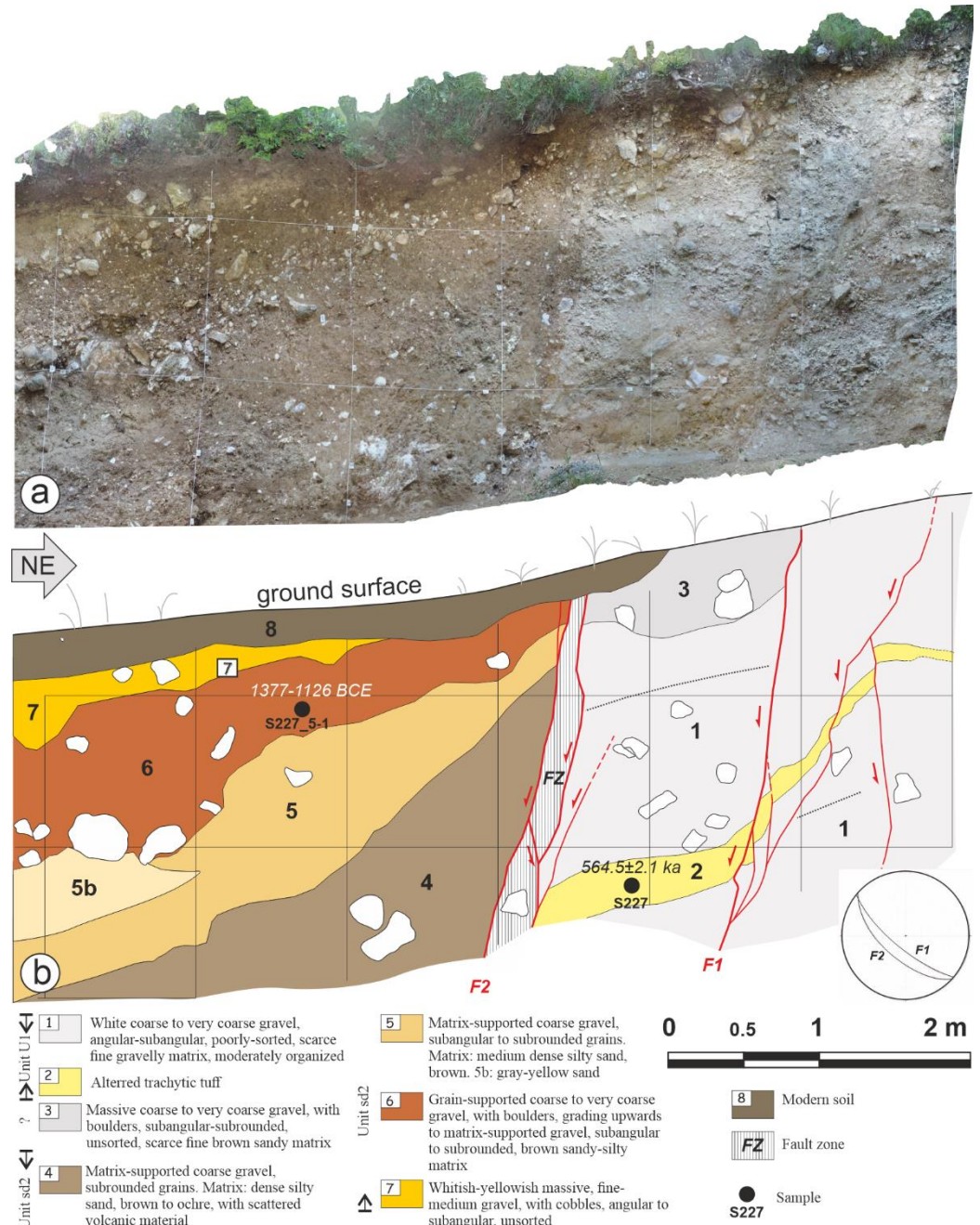

**Figure 8: Photomosaic (a) and interpreted log (b) of fault zone 1 in the western splay of the Gioia Sannitica normal fault (San Potito fault section, central part). Unit numbering (1 to 8) refers only to this outcrop (units do not necessarily correlate with other units with the same number in Fig.s 9 and 10); units U1 and sd2 refer to Fig. 3. Complete details of age determinations in Tab.s 1 and 2 and in Tab. S1 of the supplementary material. Location in Fig.s 2b and 4.**

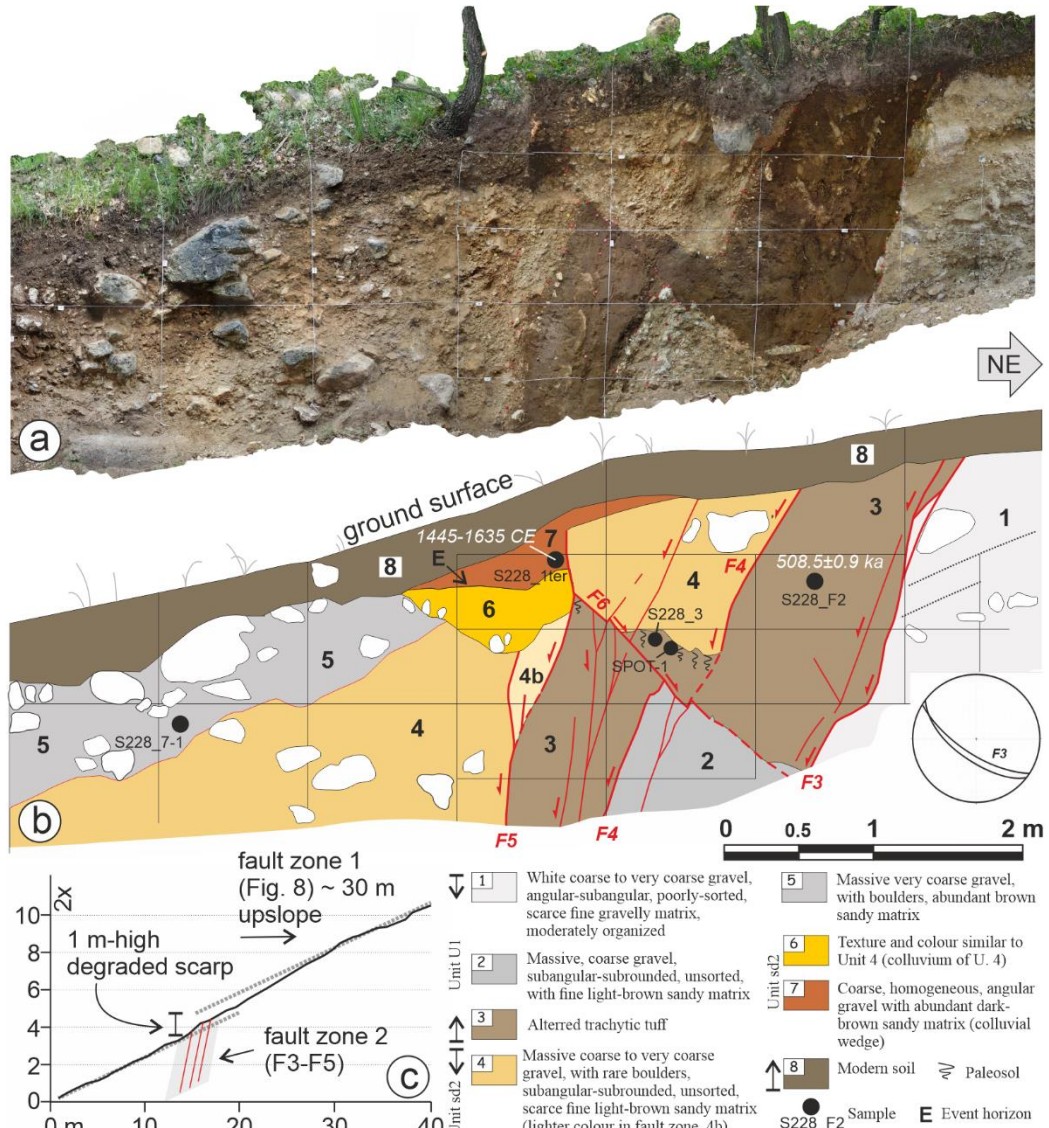

Figure 9: Photomosaic (a) and interpreted log (b) of fault zone 2 and micro-topographic profile (c) across the fault zone in the western splay of the Gioia Sannitica normal fault (San Potito fault section, central part; location in Fig.s 2b and 4). Unit numbering (1 to 8) refers only to this outcrop (units do not necessarily correlate with other units with the same number in Fig.s 8 and 10); units U1 and sd2 refer to Fig. 3. Event horizon (E) = topographic surface at the time of the last surface faulting event. Complete details of age determinations in Tab.s 1 and 2 and in Tab. S1 of the supplementary material. Detailed analysis of the paleosol (sample S228_3 and SPOT-1) in Text S1 of the supplementary material. Photographic details of the fault zone, a restoration of displacements on faults F4, F5 and F6, and a possible restoration of the last two surface faulting events are reported in Fig.s S2, S3 and S4 of the supplementary material, respectively.

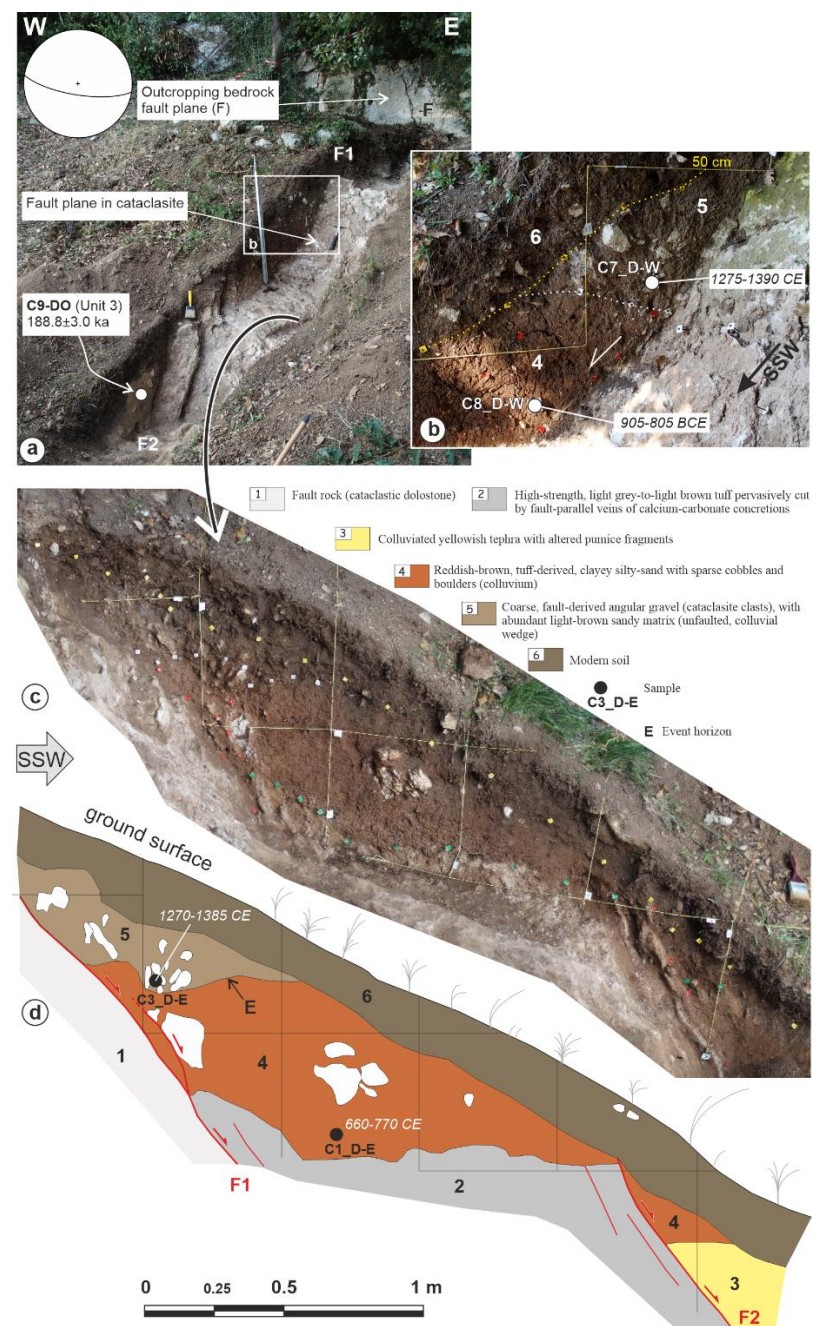

Figure 10: Field view to the North (a), detail (b), and photomosaic and log of the southern wall (c, d) of a shallow hand-dug trench across the southern Ailano-Piedimonte Matese fault (Raviscanina fault section, southern part) near Sant'Angelo d'Alife. Unit numbering (1 to 6) refers only to this outcrop (units do not necessarily correlate with other units with the same number in Fig.s 8 and 9); units 3 to 6 belong to unit sd2 of Fig. 3. Fault numbering (F1, F2) refers only to this outcrop (are not the same of Fig.s 8 and 9). Event horizon (E) = topographic surface at the time of the last surface faulting event. Complete details of age determinations in Tab.s 1 and 2 and in Tab. S1 of the supplementary material. Location in Fig. 2a.

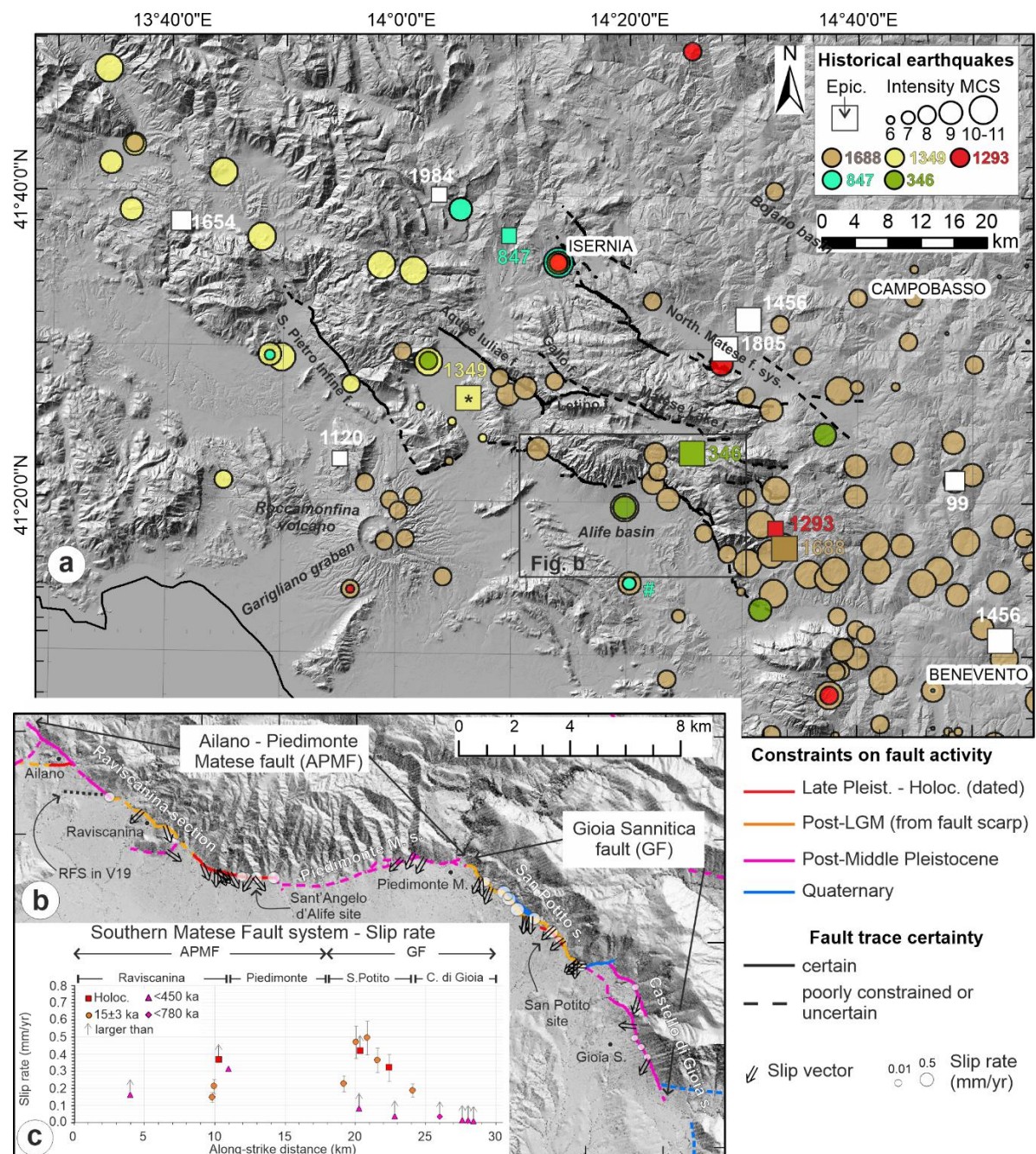

**Figure 11: a)** Distribution of intensity data points (from DBMI15, Locati et al., 2021) of the largest historical earthquakes in the Matese area (epicentres from CPTI15, Rovida et al., 2020; from CFTI5Med for events before 1000, Guidoboni et al., 2019; * = epicentre from Galli and Naso, 2009); # = intensity data point of the 847 earthquake from Bottari et al., 2020. **b)** Fault activity classification and fault slip data along the Southern Matese Fault system are based on data from this work (GF, Sant'Angelo d'Alife site) and from Boncio et al. (2016) (APMF); RFS in V19 = Raviscanina fault scarp mapped in Valente et al. (2019). **c)** Along-strike slip rate distribution.

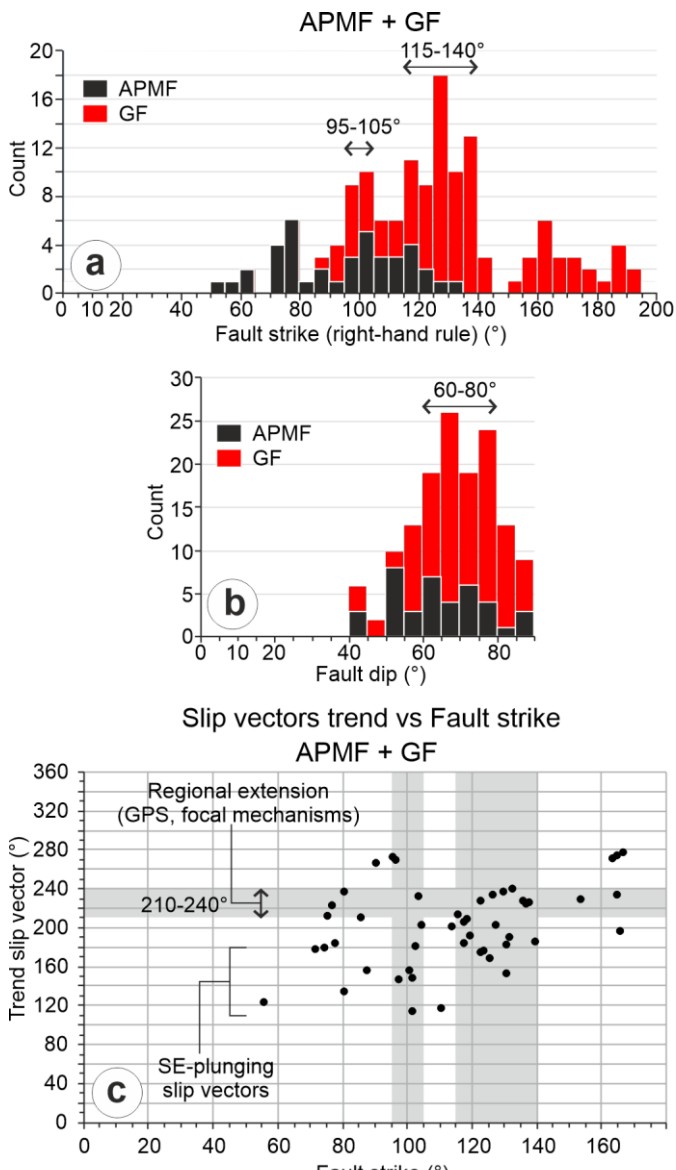

Figure 12: Summary of structural data for fault strike (a), fault dip (b) and slip vector trend (d) collected along the Southern Matese Fault system in this work (GF) and in Boncio et al. 2016 (APMF). Grey vertical bands in (c) correspond to the ranges of strike most represented in (b). Numerical values, location and source of point data in Tab. S3 of supplementary material.

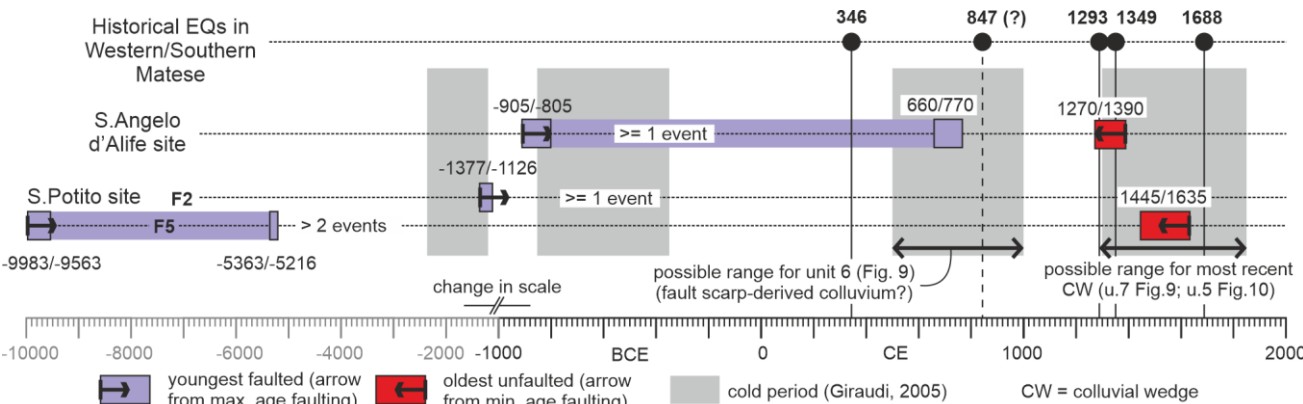

**Figure 13: Age constraints for surface faulting events according to data collected in the San Potito (GF) and Sant'Angelo d'Alife (APMF) sites compared with the ages of historical earthquakes.**