# Peer review of "Late Quaternary faulting in southern Matese (Italy): implications for earthquake potential and slip rate variability in the southern Apennines"

_Solid Earth, 2021_

## Referee Comment (RC2)

**COMMENTS TO THE PAPER SE-2021-73**
**(Late Quaternary faulting in southern Matese (central Italy): implications for earthquake potential in the southern Apennines; by Boncio et al.)**

**GENERAL COMMENTS**

The paper deals with active tectonics along a sector of the Southern Matese mountain front that has been believed to exert low tectonic activity in the late Quaternary.

The theme is of high scientific interest, as very recent activity of this fault strand has not been proved to date. In addition, the authors propose a correlation between the Gioia Sannitica fault activity and some poorly known historical earthquakes (e.g., the 346 and the 1293 events).

English style is fine, I just highlighted very few corrections at the end of this file (see section "Technical correction") but I'm not an English mother tongue, so I may have missed something.

I think the paper may be of high interest, but it needs to be re-thought in the light of general and specific comments. In particular, I encourage the authors to clearly distinguish results and discussion section, which are not easily identifiable in this current version, and to focus the discussion on the comparison between APMF and GF, and not only on the seismogenic potential of GF. So, despite the high interest of the argument, I think the paper needs relevant modifications before acceptance. I highlighted several points that are listed below, and my decision is to reconsider it after major revision.

One of the main criticisms is that authors show evidence of recent tectonic activity along the GF but they also mentioned that this portion of the mountain front is mature by referring to Valente et al. (2019). A mature mountain front would imply either an inactive mountain front or very low fault slip-rate overwhelmed by erosional processes. I never found I clear discussion of these contrasting data, which is just shallowing approached in Sections 6.2 and 7.

Furthermore, proving the tectonic activity of a mature mountain front is a very interesting issue, but a comparison with similar case studies in different tectonic and climate context is missing. I encourage you to address this issue, which would increase interest of the international scientific community towards this paper. This should be discussed in a separate section before the Conclusion.

Evidences of recent tectonic activity of GF include some meter-high scarp detected by Lidar data at the base of the mountain front. By the way, this scarp occurs close to the alluvial fans' topographic apexes, where the alluvial fans are strongly dissected (up to several meters) by channels feeding youngest fans (channels are very clear by Lidar in Fig. 2b). So, you should consider the hypothesis that this scarp may be due to erosion and not to fault activity. Furthermore, post-LGM throw rates are not convincing to the south of GF (see specific comments).

To enforce your hypothesis that this scarp is due to fault activity, you should provide other geomorphic markers of recent tectonic activity of the GF, such as knickpoints. Given that the scarp cross over adjoining drainage basins, I expect that river long profile should have a knickpoint in the surroundings of the scarp. You have Lidar data so river long profile should highlight the presence of some meters high knickpoints. If not, the scarp may be the result of differential erosion, as it seems to be.

Regarding along strike variations in throw rates, you mentioned that the APMF has no evidence of recent tectonic activity. Anyway, I think that throw rates you estimated near Sant'Angelo d'Alife are supported by strong field evidence (Fig. 10), whereas throw rates along GF are not very clear and may lead to possible alternative interpretation (see comments in the entire file about throw rates). In my opinion, your data highlight a more recent activity of the APMF than the GF, which would imply that APMF is more active than GF, according to already published data. I think this is a crucial point, and your discussion should be addressed towards comparison of the two study areas, and not only in addressing the seismogenic potential of GF.
The seismogenic potential of GF should be re-thought according to the previous comment.

Stratigraphical and structural setting in the Mt. Erbano embayment is very tricky. Firstly, the presence of the varicolored clays (intended as a lithological unit, not like a Formation) within the Meso-Cenozoic succession of the Camposauro and northern Matese ridges has been proved (Vitale and Ciarcia, 2018). So, are you sure that Vitale and Ciarcia (2018) setting cannot be exported to your area? If so, you do not need a back-thrust, but the varicolored clays may stratigraphically overlain the Miocene deposits. Secondly, the dashed projection of the GF towards south-east is not convincing as you do not show evidence of recent fault activity in this area. In fact, profile 12, which is the only profile in this sector of GF, show an 85 m high scarp carved in the carbonate bedrock. This would imply that the scarp is due to a lithological contrast between the bedrock and the debris deposits. The apparent offset of unit sd1 may due to erosion. In fact, profile 12 is very close to a deep incision, which may have lead erosion that caused the outcrop of the buried bedrock, with debris on top of the bedrock being simply a remnant of the Quaternary cover.

A point that deserves further discussion is the apparent increasing throw rate from the Middle Pleistocee (<0.1 mm/yr in the last 450 kyr) to the Upper Pleistocene and the Holocene (>0.2-0.4 mm/yr in the last 15 kyr). Several factors can underestimate long-term fault slip rates. One of them refers likely to geological processes such as compaction of accumulated sediments and/or erosion of marker beds. You should also discuss this point.

The age of the paleosol on top of tephra layer in Fig. 9 rise some questions. It would imply that there has been no sedimentation from 508 ka to 6 ka, and that the mountain front has been not active in this very long time interval. Consequently, units 4 to 7 accumulated since the Middle Holocene. This contrast with the age you assigned to Quaternary units as no Holocene deposits have been mapped in this sector of the pediment. Also, the story these data would imply is:
1) tectonic quiescence from 500 ka to 6 ka; 2) accumulation of units 4, 4b and 5; 3) faulting of these units; 4) erosion and accumulation of units 6; 5) faulting of this unit (you mentioned it at line 349, but I disagree with your hypothesis as the colluvial wedge seems accumulated in a topographic low due to erosion. I don't see any thickening of the colluvial wedge towards F5. Again, its geometry may be due to erosion); 6) accumulation of unit 7; 7) formation of the modern soil. Maybe, it is a bit too much to occur in just 6 ka! Is it possible that this very young age is due to contamination from the overlying deposits?

**SPECIFIC COMMENTS**

**MAIN TEXT**

**TITLE:** Central Italy and Southern Apennines sounds in contrast. The Matese ridge is part of the Southern Apennines and study area is geographically constrained to the Southern Italy.

**Line 17:** structural data in Supplementary File "Tab.S3_Data points with structural data" refer to the entire Southern Matese Fault System, and not only to the Gioia Sannitica fault. You should

either add location of structural data in the main map or add a field in the excel tab to specify to what sector of the Southern Matese Fault System the data are referred.

**Line 19:** are you sure "mature geomorphology" is the correct term? It may be used to indicate a tectonically inactive area, so its use may lead to misunderstanding.

**Lines 22:** I guess you reported here the average values listed in Tab. 4. If so, average value along GF is 6.2 and not 6.1.

**Line 25:** the 1349 event is not poorly known such as the 346, the 847 and the 1293 events.

**SECTION 1 - INTRODUCTION:** the authors should introduce papers that discuss very recent activity of the Southern Matese Fault System (e.g., Ascione et al., 2018; Valente et al., 2019). It seems that you addressed the Introduction towards your findings, which is to prove the very recent activity of the Gioia Sannitica Fault, without referring to papers that contrast with your working idea. Furthermore, in Boncio et al. (2016), you already mapped the APMF (Ailano-Piedimonte Matese Fault) as an active structure and the PMGF (Piedimonte Matese – Gioia Sannitica Fault) as a possible active structure, so you should also introduce this point.

**Line 34:** add some reference to support this sentence.

**Lines 63-65:** add references about the morphotectonic setting of the Matese massif.

**Line 68, lines 72-74 and lines 81-82:** refer to scheme showing tectonic evolution from Valente et al. (2019)

**Line 86:** Ailano is not mentioned in Fig. 1, maybe you referred to Alife

**Lines 90-93:** references listed in this section do not talk about the presence of this tephra layers within the Middle Pleistocene deposits, but refer to the volcanic activity of these volcanic areas. You should refer to papers that suggest the presence of the Roccamonfina and the Campi Flegrei products within the Middle Pleistocene deposits.

**SECTION 2.2 – QUATERNARY TECTONICS:** you should mention the paper by Ascione et al. (2018) that show evidences of recent activity of faults bounding the southern slope of the Matese massif.

**Line 95:** Pay attention, you wrote that you mapped these Quaternary faults, and this sentence sounds like you are introducing your results whereas you are still in Geological Setting.

**Lines 97-98:** connection between the San Pietro Infine fault and the SMF is not discussed in this paper, so you should add adequate references. I think your paper Boncio et al., 2016 is one of the papers, or even the first one, where this connection is hypothesized.

**Lines 99-104:** you should also add references to papers that highlight recent tectonic activity of the SMF (e.g., Boncio et al., 2016; Ascione et al., 2018; Valente et al., 2019).

**Lines 108-110:** add reference to justify along strike variation in fault activity along the SMF. I guess it should be the paper by Valente et al. (2019). If so, the sentence at lines 116-118 is a repetition of this sentence.

**Line 131:** add reference to Galadini and Galli, 2004_Annals of Geophysics.

**Line 135:** you mentioned that the 2013 event had a magnitude of 4.9 whereas you indicated a magnitude of 5.2 in Fig. 1. Rovida et al. (2020) indicate magnitude 5.16 for the 2013 Matese event. Please, make the main text and the figure consistent. Furthermore, detail of the 2013 event are also reported in Valente et al. (2018), so you should also refer to this paper.

**Lines 138-141:** why did you refer to the 2016 event? As you correctly stated, it occurs NE of the Matese area. It testifies that earthquakes in this sector of the Apennine chain occur along NW-SE striking normal faults, but this event is not due to activity of the faults bounding the Matese massif. Or, do you think this event is due to activity of the Northern Matese Fault System? Do you have data for supporting this hypothesis? If yes, please add further details to support it. If not, you should not mention the 2016 event in this section.

**Line 146:** what does CTR means? I know, but you should specify it for international readers not used to Italian acronyms.

**SECTION 3.2 – SAMPLE DATING:** you mentioned that you performed tephrostratigraphic analysis, but I did not find any tephrostratigraphic analysis both in the main text and in the supplementary data.

**Lines 182-183:** see comments to Supplementary Data.

**Lines 222-227:** is the correlation between tuff interlayered in the U2 deposits and the WTT supported by tephrostratigraphic analysis you mentioned in the Methods section? Or, is it just a working hypothesis?

**Line 247:** refer to Plate 1 to locate Criscia.

**Lines 298-302:** estimated post-LGM throw rates are not convincing. In fact, profile 13 shows a 6.5 m offset in Sd1 unit, which you constrained to the late Lower – Middle Pleistocene, whereas profiles 14 and 15 show offset U1 unit, dated at the early Middle Pleistocene. Throw rates should be referred to these chronological intervals, which may lead to values like those one obtained by profiles 3, 9, 11 and 12.

**Line 355:** if the scarp is due to erosion, as you mentioned, then it is not a fault scarp and you cannot derive throw rates.

**SECTION 6.1 – OVERALL ARCHITECTURE AND KINEMATICS:** you should also refer to scheme showing tectonic evolution of the SMF proposed by Valente et al. (2019), and not only to papers by the authors.

**SECTION 6.2 – THROW RATES:** regarding mature geomorphology of mountain front, you did not introduce any data supporting this hypothesis, so you should just refer to Valente et al. (2019). You should also consider the possibility that the scarp you used to derive throw rate is only due to erosion (see general comments).

**Line 405:** compare your values with values from Cinque et al. (2000) and Ferranti et al. (2014).

**SECTION 6.3 – SEISMOGENIC POTENTIAL:** this section should be revised according to my comments on how you estimated throw rates.

**Table 1 and 2:** you should add coordinates of sampling site and show their location in some map, at least in Fig. 4 and Plate 1. Regarding samples in the surroundings of Sant'Angelo d'Alife, you may add their location in Fig. 2.

**FIGURES**

**Figure 1:** correct "Northern Maters Fault System". Why did you place the 346 epicentre in the core of the Matese massif? Galli and Naso (2009) and Galli et al. (2017) placed it in the surroundings of Ailano. You also mentioned, in the main text, that the SMF may be the source area of the 346 earthquake, so it looks a bit in contrast the location of this event in the core of the massif. What data do you have to place it to the south of the Matese Lake?
Furthermore, are there marine deposits within the Quaternary basin? Are you sure? If so, cite papers mentioning these marine deposits within the main text. Also, the upper age limit of this deposits is missing.
Add coordinates and north arrow.

**Figure 2:** fault traces should be thicker.

**Figure 3:** you show that units U3 and sd2 contain the NYT and the CI tephra layers. Anyway, you never presented data showing the presence of this tephras both in the main text (see lines 228 to 234) and in the boreholes section of Plate 1. What data do you have? Please, introduce them in the main text.

**Figure 4:** revise accordingly to comments at geological map and cross-sections in Plate 1.

**Figure 6:** the outcrop of Triassic bedrock close to the fault trace and the lateral variation in scarp height, which is maxima where the bedrock outcrops, suggest that the scarp is due to differential erosion and not to fault activity.

**Figure 7:** specify, in the caption, if fault colours are referred to fault colours in Plate 1. If not, add a small legend to specify what different fault colours indicate.

**Figures 8 and 9:** to facilitate readability of these figures, authors should complete details of faulted deposits with reference to units mapped in the geological map (Fig. 4 and Plate 1). Unit U1 is present in this sector of the study area, which should refer to units 1 to 3 in both figures, whereas C14 data indicates that units 4 to 7 are Holocene in age, but there are no Holocene deposits in this sector of the pediment. Furthermore, it seems to me that unit 6 in Figure 8 can be prolonged towards the NE and may cover the fault trace, but maybe this different interpretation is due to colours in the picture.

**Figure 10:** age of unit 4 should be "BCE" and not "CE".

**Figure 11:** I missed in the main text Middle Pleistocene to 15.3 ka throw rates for the Raviscanina and Piedimonte Matese areas. I just found throw rate >0.35 mm/yr during the late Holocene (see line 379 of the main text).

**SUPPLEMENTARY DATA**

**TAB. S3_Data points with structural data**
-   See comment to line 17.

**PLATE S1**
**MAIN MAP**
- The Gioia Sannitica fault passes, laterally, to a generic Quaternary fault to the north of Calvisi. How it is possible?
- The tectonic setting near Curti, Petrella and Caselle is very tricky (see comments in the "General comments" section)
- The Quaternary faults to the west of GF are not supported by your data (see next comments), so I suggest removing them and modifying the cross-sections accordingly

**CROSS-SECTION**
Enlarge text on the X and Y axes. Add "Elevation (m a.s.l.)" to the left of the Y axe and "distance (m)" below the X axes.
**CROSS SECTION A-A'**
- I have some doubts about blue faults you reported in this cross-section. Borehole data do not show any offset, as borehole SS6 drill the Miocene unit whereas boreholes SS2 and PMS4 drill alluvial units. If you project the topographic surface near borehole SS6 south-westwards, you do not need a fault (see red lines in the figure below). So, it is easier to interpret the contact between alluvial fan deposits and Miocene deposits as stratigraphical.

[Figure]

- Furthermore, borehole data do not allow to infer the presence of another fault, near borehole PMS6, because there are no offsets in the Quaternary units. In this area, borehole data do not allow to infer both the thickness of the alluvial fan deposits and the depth of the bedrock, so the lower part of the cross-section is not supported by data. I suggest ending the cross-sections at the base of boreholes (a possible solution is provided below).

[Figure]

- I also found some incongruences about Miocene deposits dip in the hangingwall of red fault (e.g, the Gioia Sannitica fault). In fact, Miocene deposits dip towards SW in cross-section A-A' and D-D', towards NE in cross-section B-B', C-C' and F-F', and are horizontal in cross-section E-E'. Few bedding data on Miocene deposits reported in the main map do not allow to support such variations, so the authors should re-draw this part of the cross-sections.
- Lastly, the Cretaceous units dip 40° towards the SW. This is the only bedding data reported in the main map and it does not support folding of this units as reported in cross-section A-A'. Furthermore, I evaluated the dip of the Mt. Olnito slope which resulted to be of 45°. This would suggest that this slope is a dip-slope. So, if the very recent activity of the Gioia Sannitica fault is estimated by the steeper lower sector of the Mt. Olnito slope, I think this interpretation is not correct.

[Figure]

**CROSS SECTION B-B'**
- What are the evidences to infer the location of a Quaternary fault to the NE of the Gioia Sannitica fault?
- Your data do not allow to infer depth of the Quaternary bedrock, so you should stop your cross-section at the base of the boreholes (the same for all the cross-sections)

- Again, your data do not support the presence of an inferred Quaternary fault to the SW of San Potito Sannitico

**CROSS SECTION C-C'**
- Stratigraphy of the borehole S"S93" is not reported in the borehole stratigraphy figure
- There are no evidences to infer the presence of the dashed blue fault.
- What data do you have to infer the tectonic contact between Mesozoic carbonates and Miocene units along blue fault?

**CROSS SECTION D-D'**
- Again, there are no evidences to infer the presence of the dashed blue faults.
- How can you hypothesize the presence of the Varicoloured Clays? Looking at your map, they seem to be limited to the Petrella area. See also comments in the "General comment" section.

**CROSS SECTION E-E'**
- The map shows, in the surroundings of Auduni, the presence of U2 deposits whereas U1 deposits are reported in the cross-section.
- This cross-section is not very clear as there are no bedding data in its surroundings that allow the infer the complex geological setting, with folded Miocene and Mesozoic units.
- The Gioia Sannitica fault is inferred in the main map whereas it is certain in this cross-section.
- Why you did not project borehole GSA4 in cross-section E-E'?

**CROSS SECTION F-F'**
- There are no evidences to infer the presence of Quaternary faults to the SW of the Gioia Sannitica fault.

**BOREHOLE DATA**
- You distinguished, in the main map, SD1 and SD2. I agree, but most of the boreholes drill another SD unit, that is not listed in the legend. Looking at both the map and the cross-sections, I guess it should be the Miocene unit. Borehole data must be consistent with geological map and cross-sections to avoid any misunderstanding.
- The unit CC drilled in boreholes S14 and GSA5 is not listed in the legend. What is it? Furthermore, authors could project borehole S14 on cross-section A-A'.
- Borehole GSA7 drill SD1 deposits whereas it drills SD2 deposits in cross-section D-D'.
- Boreholes S9, S12 and PMS2 are placed in an area where U2 deposits are present whereas they drill U1 deposits.
- I cannot find boreholes PMS6 and SS1 in the main map. Where are they located?
- Boreholes SS5 and SS6 drill SD unit in the borehole stratigraphy figure whereas they drill Miocene unit in cross-section A-A'. So, is the SD unit composed of debris slope deposits or the SD unit correspond with the Miocene unit?
- Stratigraphy of borehole GSA1 indicates that it drills, in the upper part, U3 deposits whereas it is in an area where U1 deposits are present. Furthermore, you indicated that the WTT tephra is present within the U2 deposits, whereas, in borehole GSA1, the WTT occurs within U3 deposits.
- Regarding the WTT, how can you establish that this tephra layer is the WTT? Did you sample it in all boreholes and performed tephrostratigraphic analysis? Or the classification of this tephra layer as the WTT is just your assumption?
- Again, what is the SD units in boreholes GSA3, GSA4, GSA5, GSA6, GSA7, SPS1, SPS2, SPS3, SPS4, SPS5, SPS6 and SPS7?
- Stratigraphy of borehole GSA6 indicates that it drills, in the upper part, SD2 deposits whereas it is in an area where Miocene deposits are present.
- Boreholes 158202, 163920 and S"S93" and reported in the main map but are not detailed in the borehole stratigraphy figure.

**TECHNICAL CORRECTIONS**

**Line 84:** type NE-SW instead of SW-NE
**Line 168:** delete "which was".
**Line 173:** the letter "o" of Alo, Feo, Sio, should be written in capitals.
**Line 241:** it should be Mt. Acero.

---

## Referee Comment (RC3)

[referee-annotated manuscript omitted]

---

## Author Comment (AC1)

**Response to Reviewer 1 (Francesco Pavano)**

Black: original comments by Reviewer

Red: response by Authors

I concluded to read the manuscript submitted to Solid Earth by Boncio et al., entitled "Late Quaternary faulting in southern Matese (central Italy): implications for earthquake potential in the southern Apennines". I appreciated the opportunity to read this manuscript.

**Overview**

Moved by the tragic events occurred recently in Central Italy, and pointing out the implications in the seismic hazard assessments of the study area, the authors try to address the issue of the definition of the seismic potential of the southern Matese area (central Italy), where slowly-slipping faults, with long return periods of > M 6.5 earthquakes, occur. The authors address these topics by combining geological, paleoseismological and geomorphological approaches and also considering the available data about both the historical seismicity record of the area and the up to 30 m-deep drill holes (Plate 1). The work benefit of a supplementary 1:20,000 scale geological map, equipped with several geological cross-sections, where the geological and some morphological information are presented.

Several age determinations have been performed (e.g., 40Ar/39Ar, 14C) by analysing several samples collected at specific horizons (e.g., tephra, paleosols), in order to reconstruct a history of events (e.g., deposition, weathering, erosion), inferring potential surface faulting episodes suggestive of the occurrence of past earthquakes. These data are used by the authors to characterize the seismic potential of the normal faults of the Southern Matese Fault system (APMF and GF) that control the southern slope of the Matese Mts., trying to associate to them some historical, sometime still poorly constrained, earthquakes (e.g., 847 CE).

**General comments**

I found the manuscript suitable for publication on Solid Earth, and the work has great potential in contributing in the definition of the seismic potential of a still poorly-studied fault system portion of the NE-SW-stretching southern Apennines, with great repercussion on the seismic hazard assessment of the area. In this regard, the provided ages determinations represent useful anchor points to reconstruct the morphotectonic evolution of the study area and to attempt to associate past strong earthquakes to this fault system.

Anyway, as it is, the manuscript needs a general reorganization and a more convincing way to present, interpret and discuss the collected data.

I suggest to the authors to emphasize the main goal of the work in the introduction;

Especially through the sections 4 and 5, it is not clear what are the new and the already available data/information, and sometimes there are some not-univocal interpretations of the presented data;

The discussion section strongly would benefit of a revision, mainly focused i) on the interpretation of data in the light of the initial hypotheses/aims and ii) on arguing any conclusive statement with the strong support of any of the obtained ages data, field evidences and available/previous information.

The data presented in some figures (e.g., Fig. 11b-c) are barely discussed in the main text, and they misleadingly could appear of secondary relevance.

Given the potential of this work, the suitability for publication on SE, and given the amount of work requested for the revision, I suggest that for the final publication, the manuscript should be reconsidered after major revisions.

Below are some general comments for single sections, then line by line comments follow.

We thank the Reviewer for the detailed review and for constructive comments. In the following, the point by point response to the comments:

General comments by sections:

Section 1: in this introduction section I suggest to the authors to give more emphasis to the issues that they want to address in the study, remarking its main goal.

**1) Good suggestion. We will remark the main goal in the Introduction, by adding the following from line 51:**

This paper focuses on the normal faults cropping out along the southern slopes of the Matese Mts., named Southern Matese Fault system (Ailano – Piedimonte Matese and Gioia Sannitica faults; Fig. 2). The main goal is to determine if the fault system must be considered active, possibly seismogenic, and with what seismogenic potential. We seek to achieve the expected result by performing an earthquake geology study aimed at mapping the fault traces in detail, collecting field evidence of recent activity, particularly Late Pleistocene - Holocene, looking for evidence of earthquake-related surface faulting episodes, and combining all the evidence in a consistent seismotectonc frame for estimating the likely earthquake potential. We start studying in detail the geology of the Gioia Sannitica normal fault, which is the less constrained fault of the system. The results are described in Section 4, and a detailed, 1:20,000-scale geologic map of the Gioia Sannitica normal fault is attached as supplementary material (Plate S1). Fault scarp heights are recognized, carefully selected to avoid scarps of non-tectonic origin, measured using high-resolution topography, and used to derive fault throw rates. In sub-section 4.2, we describe evidence of Holocene surface faulting, discovered both on the Gioia Sannitica and Ailano - Piedimonte Matese faults. For the first time, we show clear evidence of late Quaternary and Holocene faulting, thanks to detailed field analyses of fault zones in dated (14C and 40Ar/39Ar) Quaternary sediments. In Section 5 we integrate our new data with data deriving from previous geological works on the Ailano - Piedimonte Matese fault (Boncio et al., 2016). Our results are then discussed in the light of a recent geomorphology and Quaternary stratigraphy study of the Southern Matese mountain front (Valente et al., 2019). Finally, the new and pre-existing data are discussed together in terms of present activity and overall seismogenic potential of the Southern Matese Fault system.

Section 2: except for few comments on both the main text and the related figures (e.g., Fig. 1) this section is well written.

Section 3: Sub-section 3.1 actually could serve as a general introductive paragraph of this section rather than a description of the carried-out field work. Except for additional few comments and suggestions, the rest of this section is well written.

**2**) We prefer to leave sub-section 3.1, which will be integrated with methodological details on fault scarp selection and measurements. In particular, we will specify that sites have been selected in order to avoid erosional exhumation processes.

3.1 Field mapping and fault scarp measurements

Section 4 and 5: Through the text, it seems that the description of data is sometimes mixed with their interpretation. Also, somewhere it is not clear in the description of the different fault sections what data are just descriptive, from previous studies, and what are new. I think that the authors could clearly distinguish previous information (moving them in a more general introductive description) from their new data. Furthermore, I think that the classical structure in "Results" and "Discussion" could help in this case.

3) Suggestion accepted. The revised paper will be organized in Results and Discussion:

Note that all the information provided in former sections 4 and 5 are new and original (i.e., derived from our field survey). There are a few exceptions, with information deriving from the literature and included in sections 4 and 5 because considered relevant for the described subject. When the information is from the literature, the original source is always acknowledged. If there are not any references, it means that this is original work, as usually done in scientific contributions.

Anyway, it is evident that the present organization is not satisfactory, not sufficiently clear. So, we will be careful in removing from the Results section all the unnecessary data deriving from the literature, and better organize the structure of the paper as follow:

4. Results

4.1 Geology of the Gioia Sannitica normal fault from field mapping

4.1.1 Geomorphology and stratigraphy of the southern Matese piedmont along the GF

4.1.2 Geometry, kinematics and fault scarp morphology

San Potito fault section

Castello di Gioia fault section

4.2 Late Pleistocene – Holocene surface faulting

4.2.1 The San Potito site on the Gioia Sannitica Fault

*Tectonic interpretation*

4.2.2 The Sant'Angelo d'Alife trench site on the Ailano - Piedimonte Matese Fault

Tectonic interpretation

**5 Discussion**

5.1 Architecture and kinematics of GF and SMF system

5.2 Activity of SMF system, throw rates and throw rate variability

5.3 Seismogenic potential

**6** Conclusions

Section 6: This section is expected to be the strongest and the main data-supported section of the manuscript, giving space to data interpretation, and their implications, to sustain any hypotheses presented in the introduction of the manuscript. Any hypothesis or statement in this section are expected to be convincingly supported and well constrained by systematically recalling the study's results. Actually, as it is now, this paragraph appears to be weak in this regard and somewhere the discussion sounds as it was randomly or weakly argued in the light of the new data. I think that the produced data, presented through the previous paragraphs, are not appropriately used to strengthen the statements done, too quickly sometimes, in this Section 6. Somewhere through the text (e.g., at the beginning of paragraph 6.1 or paragraph 6.2), the discussion sounds like out of place, fragmented and/or poorly convincing.

Section 7: some revision is recommended in the light of the comments and suggestions provided for the previous sections.

**4**) Suggestion accepted. We are confident that the new organization in Results and Discussion will improve the paper. The Discussion will be better developed, and more convincing, we hope.

Comments line by line:

Line 25-26: CE 1293; CE 1349; CE 847

5) In general, in seismologic or seismotectonic papers, when dealing with historical earthquakes, "CE" (or BCE) is used only for ancient events, in general before 1000 A.D.

Anyway, considering that all the cited events are CE, we will remove "CE" from all the earthquake dates. This is consistent with the standard of the historical earthquake catalogues we used.

Line 52: Refer to Fig. 2 for the APMF and GF.

Line 78: This fault system is not shown in Fig. 1. Is this system partially reactivated as a transfer fault, now forming the Presenzano-Ailano transfer? At least the Garigliano Graben should be labelled somehow in Fig. 1, since it is discussed in Section 6.

7) Accepted. We will add "Garigliano graben" in Fig. 1.

Line 104-110: somewhere cite Fig. 2 for location of the fault sections.

8) Accepted. We will modify the manuscript accordingly.

Line 120-121: cite Fig. 1 for earthquakes' locations.

9) Accepted. We will modify the manuscript accordingly.

Line 144-152: This paragraph sounds like a general, introductive text rather than a description of field geology methods' explanation. A brief description of how locations, lengths and depths of the trenches have been chosen could be added to this section.

10) See reply n. 2. The location and size of the Sant'Angelo d'Alife trench site will be added. Please note that the San Potito site is not a trench, it is a road cut.

Line 157: I think that the reader could benefit of some more detailed information here about the adopted approaches. For example, why the glass fragmentation would be good to analyze? Why it was important to sieve clasts at 1 phi interval?

11) The tephrostratigraphic approach on distal volcaniclastic deposits embedded in Middle-Early Late Pleistocene successions, which are often deeply altered, is generally not so straightforward. As a matter of fact, you generally are able to distinguish in the field the volcaniclastic nature of the deposits, but the following analytical phases are complicated by the deep alteration of the pristine glass fraction characterizing the layer. Obtaining chemical data on the latter, both in terms of major and trace elements, however, would make it possible an attribution to a possible source and, even, to a specific eruptive event. This is the reason why the samples are pretreated with several washing phases and sieved at different grain-sizes, in the hope to find, at least in the finer fraction, a little amount of fresh glass. Unfortunately, this was not the case of our samples, for which only a preliminary characterization of lithological and crystal component could be achieved. Anyway, we are available to better explain the rationale of the procedure, as the reviewer asks.

An eventual, detailed and more technical description of the dating techniques (additional to the paleosol analysis) could be provided as supporting information or supplementary data.

12) We think this is not necessary. Used 14C and 40Ar/39Ar methodologies are rather well-known in the geologic community. The cited references to basic papers and/or to Lab. web sites will address readers interested in technical details.

Line 161: The pre-treatment (e.g., washing, sieving the sediments, sanidine phenocrysts extraction) was performed at the University of Wisconsin-Madison?

13) At the University of Naples "Federico II". This will be specified.

Line 209-210: cite Figure(s) where these locations are reported.

Line 231-233: It could be useful to add any more detailed information on the sedimentary facies, structures and textures of Unit 3 and 4 in this paragraph 4.1.

15) Accepted. We will modify the manuscript accordingly (consistent with the description in Plate S1).

Line 244: Figure 5 show very beautiful and net rejuvenated fault scarp. In addition to the plots, is there any picture showing some kinematic indicators of the striated fault planes?

16) Slickenlines are reported as slip vectors in the stereonets.

A photo with a detail of the fault plane and slickenlines will be added in Fig. 6.

Line 247: "south of Fig. 5c" is clearer than "S of Fig. 5c".

17) Accepted. We will modify the manuscript accordingly.

Line 247: cite the Figure that shows the location of Criscia.

18) Accepted (Fig. 4). The locality Criscia will be removed from the text (this is not necessary).

Line 254-256: any references about this statement?

19) This is an original observation from field evidence. Please, see Plate S1. We will explain in the text more clearly, referring to Plate S1.

Line 259-260: It is difficult to see Early Pleistocene slope breccia deposits in the footwall of the San Potito Fault in Section 2 of Fig. 4. Do you refer to Profile 12 in Figure 7? A figure showing these data needs to be cited.

20) Accepted. This is the sd1 cropping out close to the trace of cross section 2 in Fig. 4, in the footwall of GF. This will be specified in the text. The outcrop of sd1 will be made more visible in Fig. 4.

Line 261: Refer to section C-C' of Plate 1, instead of "section C".

21) Accepted. We will modify the manuscript accordingly.

Line 264: cite Fig. 2a for the location of San Potito Sannitico.

22) Accepted. We will modify the manuscript accordingly.

Line 271 and elsewhere: LiDAR instead of LiDaR

23) Accepted. We will modify the manuscript accordingly.

Line 274: Show M. Olnito in Fig. 4 and cite the Fig. 4 in the text for location.

Line 274-275: is there any picture of the faulted valley? What is the condition of the fault-related knickpoint? What the offset of the valley bottom? Is this 3m, like the cumulated fault scarp? These are additional, important geomorphological elements. As described in the text and as showed in Fig. 4, it seems that the small valley is incised on the colluvial deposits sd2. Since the LGM, the fault-related knickpoint should have been moved upstream of a distance commensurate mainly with the drainage discharge, the channel slope, the uplift rate and the rock (sd2) erodibility; thus, the occurrence of a knickpoint at/close the fault trace could mean that it did not move enough upstream and, thus, potentially it could be an indication of a relatively recent surface faulting event. I do not ask the authors to perform a new drainage system investigation, but just to show, if any, some picture/scheme of this geomorphic marker and to describe/acknowledge in the text this eventual occurrence.

25) We agree that drainage analysis aimed at identifying knickpoints is a very useful tool. In this paper we focused mostly on fault scarps (identified and measured on high-resolution topography) because we think that:

- fault scarps, if measured in proper sites (away from erosional exhumation processes), can provide good estimates of throw rates;

- the fault scarp illustrated in Fig. 6 is a good example of post-LGM fault scarp. We used this area for calibrating our observations. The fault is entirely in Unit sd2 (i.e, no different lithology across the scarp). It is clearly a fault scarp, as it is just along the fault. The bedrock fault planes crop out in the eastern and western (see stereonet) sides of the scarp. A photo of the fault plane will be added in the revised version of the manuscript. Part of the scarp (western and eastern sides) is modified by anthropogenic activity, and we carefully avoided those sites. Because the scarp is entirely in the sd2 deposits, and considering that the dip of the post-LGM deposits (dated) is nearly coincident with the dip of the topographic surface (photo in profile 7), a non-tectonic origin, by e.g. differential erosion, stratigraphic features or other non-tectonic processes, is not reasonable. The fault offsets the topographic surface and the underlying stratigraphy, and this displacement is post-LGM. Therefore, the fault scarp is a good feature to estimate post-LGM throw and throw rate. Please note that the valley in the central part of the figure, entirely within the sd2 unit, is more incised upslope the scarp (i.e., in the footwall of the fault). This is consistent with footwall uplift due to faulting.

Concerning the small scarps in profile 4 (total throw of ca. 4 m), we totally agree with you: "... the occurrence of a knickpoint at/close the fault trace could mean that it did not move enough upstream and, thus, potentially it could be an indication of a relatively recent surface faulting ...".

The scarps are visible in the field, but unfortunately we could not capture a good photo showing the entire scarps, due to logistic difficulties and vegetation. But, the scarps are clearly visible on LIDAR DEM.

Figure A (below) shows: a shaded relief of the LIDAR DEM without geology, the same with geology and a long profile across the Gioia Sannitica fault close to profile 4. In the long profile, the knickpoint (we name this "knick zone") is clear. The knickpoint height (elevation difference between crest and toe) is 3-4 m. By considering the long profile slope and the fault dip, the throw is on the order of 2-3 m. This is less than the cumulated throw measured on profile 4. This is consistent with profile 4, as the knickpoint should have registered younger slip compared to the top of the Holocene alluvium (al unit).

We are still convinced that the throw measured in profile 4 is better for estimating the cumulated throw and throw rate (post Holocene alluvium), also because the knickpoint has no chronologic constraints and because of river erosion and retreat. Moreover, there is some noise in the LIDAR data in the long profile, probably due to dense vegetation (in general the vegetation is denser along rivers compared to the alluvial planes). But, we agree that the data shown in Figure A can help in convincing sceptical readers. Therefore, we decided to add this figure as supplementary material. The graphics will be improved before submitting the revised version.

---

## Author Comment (AC2)

**Response to Reviewer 2 (Ettore Valente)**

Black: original comments by Reviewer

Red: response by Authors

Dear Authors,

you will find attached my comments. Overall, I think the paper is pretty interesting and may be suitable for publication in Solid Earth. It ayway deserves partial re-thinking and re-organisation of the Results and Discussion section, which should be more clearly distinguished.

I hope you will find my comments useful to increase the scientific interest of your paper.

Ettore Valente

We thank the Reviewer for the detailed review. In the following, the point by point response to the comments:

The paper deals with active tectonics along a sector of the Southern Matese mountain front that has been believed to exert low tectonic activity in the late Quaternary.

The theme is of high scientific interest, as very recent activity of this fault strand has not been proved to date. In addition, the authors propose a correlation between the Gioia Sannitica fault activity and some poorly known historical earthquakes (e.g., the 346 and the 1293 events).

English style is fine, I just highlighted very few corrections at the end of this file (see section "Technical correction") but I'm not an English mother tongue, so I may have missed something.

I think the paper may be of high interest, but it needs to be re-thought in the light of general and specific comments. In particular, I encourage the authors to clearly distinguish results and discussion section, which are not easily identifiable in this current version, and to focus the discussion on the comparison between APMF and GF, and not only on the seismogenic potential of GF. So, despite the high interest of the argument, I think the paper needs relevant modifications before acceptance. I highlighted several points that are listed below, and my decision is to reconsider it after major revision.

1) Concerning the re-organization of the paper, we will modify the manuscript separating more clearly Results from Discussion. Interpretations will be moved to the Discussion. The new organization will be:

4. Results

4.1 Geology of the Gioia Sannitica normal fault from field mapping

    4.1.1 Geomorphology and stratigraphy of the southern Matese piedmont along the GF

    4.1.2 Geometry, kinematics and fault scarp morphology

        *San Potito fault section*

        *Castello di Gioia fault section*

4.2 Late Pleistocene – Holocene surface faulting

    4.2.1 The San Potito site on the Gioia Sannitica Fault

        *Tectonic interpretation*

    4.2.2 The Sant'Angelo d'Alife trench site on the Ailano – Piedimonte Matese Fault

        *Tectonic interpretation*

5 Discussion

One of the main criticisms is that authors show evidence of recent tectonic activity along the GF but they also mentioned that this portion of the mountain front is mature by referring to Valente et al. (2019). A mature mountain front would imply either an inactive mountain front or very low fault slip-rate overwhelmed by erosional processes. I never found I clear discussion of these contrasting data, which is just shallowing approached in Sections 6.2 and 7.

2) Criticism accepted.

We will deepen this point in the new discussion section. The main results from Valente et al. (2019), to be honest already acknowledged in the present version of the paper, will be more exhaustively (we hope) summarized in the introductory geologic context (2.2 Quaternary tectonics), and then further discussed in the Discussion. The differences will be discussed in dedicated sub-sections (new sub-section 5.2).

We think that the discrepancies between our results and Valente et al. (2019) results are only apparent. Probably they are due to different scales/detail used in the analysis: at the scale of the mountain front in Valente et al; at a detailed scale from field mapping and high-resolution topography of fault scarps from LIDAR in our paper. I think the two results must be integrated. We tried to give this message in present section 6.2 but evidently our attempt was not efficient. We will get it better in the revised version.

Furthermore, proving the tectonic activity of a mature mountain front is a very interesting issue, but a comparison with similar case studies in different tectonic and climate context is missing. I encourage you to address this issue, which would increase interest of the international scientific community towards this paper. This should be discussed in a separate section before the Conclusion.

3) We agree that this topic can be of great potential interest. But, we think that a discussion of global significance, with comparison with other cases worldwide, is beyond the scope of the present work. Please, consider that the paper is already quite long, and it was submitted to a special issue: "Tools, data and models for 3-D seismotectonics: Italy as a key natural laboratory"

Evidences of recent tectonic activity of GF include some meter-high scarp detected by Lidar data at the base of the mountain front. By the way, this scarp occurs close to the alluvial fans' topographic apexes, where the alluvial fans are strongly dissected (up to several meters) by channels feeding youngest fans (channels are very clear by Lidar in Fig. 2b). So, you should consider the hypothesis that this scarp may be due to erosion and not to fault activity. Furthermore, post-LGM throw rates are not convincing to the south of GF (see specific comments).

4) The first, very important, activity for performing fault scarp analysis is the selection of appropriate sites. The sites must be unaffected by erosional exhumation processes. We agree on that.

We performed our scarp analysis after a careful selection of the sites. We disagree with this very generic, poorly motivated criticism.

The fault scarp illustrated in Fig. 6 is a nice example of post-LGM fault scarp. We used this area for calibrating our observations. The fault is entirely in Unit sd2 (i.e, no different lithology across the scarp). It is clearly a fault scarp, as it is just along the fault. The bedrock fault planes crop out in the eastern and western (see stereonet) sides of the scarp. A photo of the fault plane will be added in the revised version of the manuscript. Part of the scarp (western and eastern sides) is modified by anthropogenic activity. We carefully avoided those sites.

Because the scarp is entirely in the sd2 deposits, and considering that the dip of the post-LGM deposits (dated) is nearly coincident with the dip of the topographic surface (photo in profile 7), a non-tectonic origin, by e.g. differential erosion, stratigraphic features or other non-tectonic processes, is not reasonable. The fault offsets the topographic surface and

the underlying stratigraphy, and this displacement is post-LGM. Therefore, the fault scarp is a good feature to estimate post-LGM throw and throw rate. Please note that the valley in the central part of the figure, entirely within the sd2 unit, is more incised upslope the scarp (i.e., in the footwall of the fault). This is consistent with footwall uplift due to faulting.

To enforce your hypothesis that this scarp is due to fault activity, you should provide other geomorphic markers of recent tectonic activity of the GF, such as knickpoints. Given that the scarp cross over adjoining drainage basins, I expect that river long profile should have a knickpoint in the surroundings of the scarp. You have Lidar data so river long profile should highlight the presence of some meters high knickpoints. If not, the scarp may be the result of differential erosion, as it seems to be.

5) We agree that analyses aimed at identifying other geomorphic markers of recent tectonic activity is a useful tool. In this paper we focused mostly on fault scarps (identified and measured on high-resolution topography) because we think that fault scarps, if measured in proper sites (away from erosional exhumation processes; see reply to comment n. 4), can provide good estimates of throw rates.

Anyway, please have a look at Figure A below. The figure shows the area where the small scarps in Holocene alluvium, shown in profile 4 of Fig. 7 (total throw of ca. 4 m), are located. The scarps are clearly visible on LIDAR DEM.

Figure A shows: a shaded relief of the LIDAR DEM without geology, the same with geology and a long profile across the Gioia Sannitica fault close to profile 4. In the long profile, the knickpoint (we name this "knick zone") is clear. The knickpoint height (elevation difference between crest and toe) is 3-4 m. By considering the long profile slope and the fault dip, the throw is on the order of 2-3 m. This is less than the cumulated throw measured on profile 4. This is consistent with profile 4, as the knickpoint should have registered younger slip compared to the top of the Holocene alluvium (al unit).

We are convinced that the throw measured in profile 4 is better for estimating the cumulated throw and throw rate (post Holocene alluvium), also because the knickpoint has no chronologic constraints and because of river erosion and retreat. Moreover, there is some noise in the LIDAR data in the long profile, probably due to dense vegetation (in general the vegetation is denser along rivers compared to the alluvial planes). But, we agree that the data shown in Figure A can help in convincing sceptical readers. Therefore, we decided to add the figure as supplementary material. The graphics will be improved before submitting the revised version.

[Figure]

Figure A (the dashed grey line in the long profile has been drawn for highlighting the scarp in the knick zone).

Regarding along strike variations in throw rates, you mentioned that the APMF has no evidence of recent tectonic activity. Anyway, I think that throw rates you estimated near Sant'Angelo d'Alife are supported by strong field evidence (Fig. 10), whereas throw rates along GF are not very clear and may lead to possible alternative interpretation (see comments in the entire file about throw rates). In my opinion, your data highlight a more recent activity of the APMF than the GF, which would imply that APMF is more active than GF, according to already published data. I think this is a crucial point, and your discussion should be addressed towards comparison of the two study areas, and not only in addressing the seismogenic potential of GF.

The seismogenic potential of GF should be re-thought according to the previous comment.

6) This criticism cannot be accepted.

Probably the Referee is confused. We did not write " ... *that the APMF has no evidence of recent tectonic activity*". Where did you read this?

Please, note that at lines 425-427, we state that …. **the Piedimonte Matese section** of the APMF has **poor** geologic and geomorphic evidence of recent activity. We confirm this statement, because we did a careful field mapping.

In the paper, we discuss the activity of the entire Southern Matese Fault system, including APMF and GF. This is clear from the text (section 6.3), from Fig. 11 and from Tab. 4, we think. Please, note that the central part of the APMF is from red (Late Pleistocene – Holocene activity, dated) to orange (post-LGM from fault scarp).

Therefore, your criticism cannot be accepted, because unmotivated.

Please, also read carefully at lines 104 – 110:

The SMF can be divided into the Ailano – Piedimonte Matese fault to the NW, and the Gioia Sannitica fault to the SE (Boncio et al., 2016). The 18 km-long Ailano – Piedimonte Matese fault is in turn divided into the Raviscanina and Piedimonte Matese fault sections. The Raviscanina section is 11.5 km long, strikes NW-SE and progressively bends to ~W-E in the southern part (~1 km SE of the Sant'Angelo d'Alife village). The Piedimonte Matese fault section is 7 km long and strikes from W-E to WSW-ENE. The eastern part of the Piedimonte Matese fault section, striking ~ W-E, has strong geomorphic evidence of Quaternary activity, while the western part, striking WSW-ENE, is less evident due to cover deposits and larger mountain front sinuosity.

Stratigraphical and structural setting in the Mt. Erbano embayment is very tricky. Firstly, the presence of the varicolored clays (intended as a lithological unit, not like a Formation) within the Meso-Cenozoic succession of the Camposauro and northern Matese ridges has been proved (Vitale and Ciarcia, 2018). So, are you sure that Vitale and Ciarcia (2018) setting cannot be exported to your area? If so, you do not need a back-thrust, but the varicolored clays may stratigraphically overlain the Miocene deposits.

7) Thank you for the suggestion. In the Mt. Erbano embayment, bedding and structural data are a few due to poor exposures, especially for arenaceous and clayey deposits. According to the literature and geological considerations at smaller scale (Sannio-Matese), we have interpreted the contact between varicoloured clays (obviously intended as a lithological unit) and the Miocene turbidites as a folded thrust. However, taking into consideration the suggested reference (Vitale and Ciarcia, 2018), interpreting the varicoloured clays as an olistostrome within the Miocene arenaceous deposits (Caiazzo formation) is plausible. We will take into consideration this suggestion during the revision.

Secondly, the dashed projection of the GF towards south-east is not convincing as you do not show evidence of recent fault activity in this area. In fact, profile 12, which is the only profile in this sector of GF, show an 85 m high scarp carved in the carbonate bedrock. This would imply that the scarp is due to a lithological contrast between the bedrock and the debris deposits. The apparent offset of unit sd1 may due to erosion. In fact, profile 12 is very close to a deep incision, which may have lead erosion that caused the outcrop of the buried bedrock, with debris on top of the bedrock being simply a remnant of the Quaternary cover.

8) We agree that this is not the best place. We selected the location of profile 12 in order to avoid erosion from nearby channels as much as possible. But, we are aware that erosion might have had a contribution. This is the reason why in the text we say only that unit sd1 is hanging of 80-100 m in the footwall of the fault. Without further speculations.

In the revised version of the manuscript we will stress that this site can be "contaminated" by un unquantified amount of exhumation due to erosion, preventing reliable throw rate estimates.

A point that deserves further discussion is the apparent increasing throw rate from the Middle Pleistocee (<0.1 mm/yr in the last 450 kyr) to the Upper Pleistocene and the Holocene (>0.2-0.4 mm/yr in the last 15 kyr). Several factors can underestimate long-term fault slip rates. One of them refers likely to geological processes such as compaction of accumulated sediments and/or erosion of marker beds. You should also discuss this point.

9) We agree that the paper can benefit from a deeper discussion of this apparent increase in slip rate. We will discuss this in the revised version.

The age of the paleosol on top of tephra layer in Fig. 9 rise some questions. It would imply that there has been no sedimentation from 508 ka to 6 ka, and that the mountain front has been not active in this very long time interval. Consequently, units 4 to 7 accumulated since the Middle Holocene. This contrast with the age you assigned to Quaternary units as no Holocene deposits have been mapped in this sector of the pediment. Also, the story these data would imply is:

1) tectonic quiescence from 500 ka to 6 ka; 2) accumulation of units 4, 4b and 5; 3) faulting of these units; 4) erosion and accumulation of units 6; 5) faulting of this unit (you mentioned it at line 349, but I disagree with your hypothesis as the colluvial wedge seems accumulated in a topographic low due to erosion. I don't see any thickening of the colluvial wedge towards F5. Again, its geometry may be due to erosion); 6) accumulation of unit 7; 7) formation of the modern soil. Maybe, it is a bit too much to occur in just 6 ka! Is it possible that this very young age is due to contamination from the overlying deposits?

10) The paleosol is 8,460-8,200 years old (6,510 – 6,250 yrs BCE).

Yes! We are aware that this is a young age. This is the reason why we did not stop at the 14C dating, but we performed a paleosol analysis (please, read Text S1), in order to verify if the characteristics and maturity of the paleosol are consistent with the obtained ages. The paleosol shows weak andic properties and poor pedogenetic evolution, suggesting a young age (please, read lines 334-338).

Therefore, two data from independent analyses show consistent results. The obtained ages are a matter of fact; this is not an opinion. We cannot omit this fact. We must take this into consideration, and provide a reasonable explanation.

About the " ... *it is a bit too much to occur in just 6 ka!*":

What do you mean for "a bit too much"? Please provide specific, quantitative arguments.

Moreover, we have estimated a minimum vertical displacement of the paleosol of 2.4 m in 8,145 years. Considering displacements per event at the site on the order of 50 cm, as suggested by the colluvial units (see also below), we can obtain the total displacement with 5 surface faulting events, with average recurrence interval between consecutive events of >1600 years. This is not "too much".

About the "*This contrast with the age you assigned to Quaternary units as no Holocene deposits have been mapped in this sector of the pediment*": you are right. The Holocene deposits are missing due to a mistake. They will be added in the map.

About Unit 6 and the story of the outcrop of Fig. 9, please consider that:

- the bottom of U 6 truncates the underlying units. I think we agree on that. So erosion (truncation), must be considered;

- the sedimentary facies of U 6 is very similar to that of U 4, but U 4 lies on a paleosol and U 6 does not. Moreover, in the hanging wall of F5, U 4 crops out below U 6. A plausible explanation is that U 4 sourced U 6. Moreover, in U 6 there are elongated clasts with the long axis plunging sub-vertical, which suggests colluvial sedimentation from a nearby source (as for scarp-derived colluvium, within which clasts plunging downslope at the foot of the scarp are typical);

- U 6 pinches out (i.e., zero thickness) at a distance of ~1 m from the fault; it is 30 cm-thick at the contact with fault F5. This indicates thickening towards F5. This suggests a strict relation with the fault;

- please, note that U 6 is faulted. So, you have to consider its original position before the last faulting event;

- The possible explanation is (please, see figure B below):

1) faulting along F5: a fault scarp forms, with a free face exposing U 4 in the footwall; in the hanging wall, there is a sequence formed by U 4 (already previously faulted by N events) and U 5 on top;

2a) erosion started after the faulting event; possibly by water running parallel to the fault scarp and cutting down the underlying stratigraphy for a maximum of about 50 cm; the formation of a coseismic scarp, a depression at the foot of the scarp immediately after faulting, and/or open fissures that are typical during surface faulting, would have favoured linear erosion, aided by the presence of a deep valley to the SE (see Plate S1) that would have driven surficial running water. In this stage, it is likely that part of U 4 in the footwall was removed by erosion.

2b) River incision ceased for some reason, but the degradation of the coseismic scarp continued (U 4 exposed in the free face), sourcing U 6 with colluvium (scarp-derived colluvium) (stage 3);

4) after some time, a second surface faulting event faulted U 6; a new coseismic scarp formed which sourced U 7 (a scarp-derived colluvium; i.e., colluvial wedge sensu Mc Calpin, 2009; see reference below).

We are aware that the interpretation of U 6 is not easy as it must take into account erosion and then sedimentation. The uncertainties in the interpretation are always considered in the present version of the paper (e.g., question mark in Fig. 11c). But we judge our reconstruction (figure B) plausible.

We agree that the overall tectonic interpretation of the trench wall can be improved, and the uncertainties in the interpretation of unit 6 emphasized. For the sake of clarity, we decided to add a sub-section entitled: Tectonic interpretation.

We will add the figure B below as supplementary material.

[Figure]

Schematic restoration of surface faulting events

E = event horizon (topographic surface at the time of surface faulting)

Figure B - Schematic restoration of surface faulting events

SPECIFIC COMMENTS

MAIN TEXT

TITLE: Central Italy and Southern Apennines sounds in contrast. The Matese ridge is part of the Southern Apennines and study area is geographically constrained to the Southern Italy.

11) Accepted. We will change in southern Italy.

Line 17: structural data in Supplementary File "Tab.S3_Data points with structural data" refer to the entire Southern Matese Fault System, and not only to the Gioia Sannitica fault. You should either add location of structural data in the main map or add a field in the excel tab to specify to what sector of the Southern Matese Fault System the data are referred.

12) Good suggestion. We will add a field in the excel file.

Line 19: are you sure "mature geomorphology" is the correct term? It may be used to indicate a tectonically inactive area, so its use may lead to misunderstanding.

13) Accepted. We will check and use the most appropriate term.

Lines 22: I guess you reported here the average values listed in Tab. 4. If so, average value along GF is 6.2 and not 6.1.

14) Yes, it is a mistake. We will change in 6.2.

Line 25: the 1349 event is not poorly known such as the 346, the 847 and the 1293 events.

15) Accepted. We will modify the manuscript accordingly.

SECTION 1 - INTRODUCTION: the authors should introduce papers that discuss very recent activity of the Southern Matese Fault System (e.g., Ascione et al., 2018; Valente et al., 2019). It seems that you addressed the Introduction towards your findings, which is to prove the very recent activity of the Gioia Sannitica Fault, without referring to papers that contrast with your working idea. Furthermore, in Boncio et al. (2016), you already mapped the APMF (Ailano-Piedimonte Matese Fault) as an active structure and the PMGF (Piedimonte Matese – Gioia Sannitica Fault) as a possible active structure, so you should also introduce this point.

16) Accepted. We will better summarize and acknowledge the work by Ascione et al. (2018) and Valente et al. (2019).

See also replies to comments N. 2 and 6.

Line 34: add some reference to support this sentence.

Lines 63-65: add references about the morphotectonic setting of the Matese massif.

17) Accepted. See reply to comment N. 16.

Line 68, lines 72-74 and lines 81-82: refer to scheme showing tectonic evolution from Valente et al. (2019)

18) Accepted. See reply to comment N. 16.

Line 86: Ailano is not mentioned in Fig. 1, maybe you referred to Alife

19) Yes, the correct locality is Alife. We will correct the manuscript accordingly.

Lines 90-93: references listed in this section do not talk about the presence of this tephra layers within the Middle Pleistocene deposits, but refer to the volcanic activity of these volcanic areas. You should refer to papers that suggest the presence of the Roccamonfina and the Campi Flegrei products within the Middle Pleistocene deposits.

20) We had listed those papers because they were referred to the available age constraints for the whole of the Roccamonfina activity and for the main markers from Campi Flegrei (CI and NYT). We agree with the reviewer that in the present form the sentence could be confusing, so we will rephrase the statement and separate the references for previous tephra findings in the area (quoting the right papers) and age of activity at the hypothesized sources.

SECTION 2.2 – QUATERNARY TECTONICS: you should mention the paper by Ascione et al.(2018) that show evidences of recent activity of faults bounding the southern slope of the Matese massif.

21) Accepted.

Line 95: Pay attention, you wrote that you mapped these Quaternary faults, and this sentence sounds like you are introducing your results whereas you are still in Geological Setting.

22) Accepted. We will pay attention.

Lines 97-98: connection between the San Pietro Infine fault and the SMF is not discussed in this paper, so you should add adequate references. I think your paper Boncio et al., 2016 is one of the papers, or even the first one, where this connection is hypothesized.

Lines 99-104: you should also add references to papers that highlight recent tectonic activity of the SMF (e.g., Boncio et al., 2016; Ascione et al., 2018; Valente et al., 2019).

Lines 108-110: add reference to justify along strike variation in fault activity along the SMF. I guess it should be the paper by Valente et al. (2019). If so, the sentence at lines 116-118 is a repetition of this sentence.

Line 131: add reference to Galadini and Galli, 2004_Annals of Geophysics.

23) Accepted. Additional references will be added.

Line 135: you mentioned that the 2013 event had a magnitude of 4.9 whereas you indicated a magnitude of 5.2 in Fig. 1. Rovida et al. (2020) indicate magnitude 5.16 for the 2013 Matese event. Please, make the main text and the figure consistent. Furthermore, detail of the 2013 event are also reported in Valente et al. (2018), so you should also refer to this paper.

24) The correct magnitude is Mw 5.2, according to RCMT catalogue (http://rcmt2.bo.ingv.it/). We will correct the manuscript accordingly.

Lines 138-141: why did you refer to the 2016 event? As you correctly stated, it occurs NE of the Matese area. It testifies that earthquakes in this sector of the Apennine chain occur along NW-SE striking normal faults, but this event is not due to activity of the faults bounding the Matese massif.

Or, do you think this event is due to activity of the Northern Matese Fault System? Do you have data for supporting this hypothesis? If yes, please add further details to support it. If not, you should not mention the 2016 event in this section.

25) We mentioned the 2016 earthquake simply because it is in figure 1. We do not draw speculations, inferences or similar. Why we should not mention that event? Instead, it provides insights on the area currently undergoing SW-NE extension.

Line 146: what does CTR means? I know, but you should specify it for international readers not used to Italian acronyms.

26) Accepted. We will specify this acronym (Carta Tecnica Regionale, CTR: Italian name and acronym of the 1:5,000-scale topographic map of the Campania Regional authority).

SECTION 3.2 – SAMPLE DATING: you mentioned that you performed tephrostratigraphic analysis, but I did not find any tephrostratigraphic analysis both in the main text and in the supplementary data.

27) As a matter of fact, we wrote "Three tephra layers interbedded within faulted alluvial fan and colluvial sediments were sampled for tephrostratigraphic analysis and $^{40}$Ar/ $^{39}$Ar dating" and "Unfortunately, no well preserved glass fragment survived washing pre-treatment and hence chemical composition of glass could not be achieved", which is the reason why no chemical data has been reported. For the sake of shortness, as required by the journal, we did not either add the description of the results of lithological and mineral component analysis achieved on the dated and other sampled tephra, which we used to make at least some inferences on their possible attribution. In the revised version of the manuscript, we plan to add a new supplementary section, containing further information on tephra useful for the time constraining of the identified units.

Lines 182-183: see comments to Supplementary Data.

Lines 222-227: is the correlation between tuff interlayered in the U2 deposits and the WTT supported by tephrostratigraphic analysis you mentioned in the Methods section? Or, is it just a working hypothesis?

28) The age is assigned on the basis of field observations, field lithological characterization and correlations with previously published papers. We agree with the reviewer that a precise attribution to WTT Formation could be quite speculative in the absence of precise age determination and chemical data on fresh glasses (see also replay to comment n. 26), so we decided to change the possible correlation referring to Roccamonfina pre-caldera activity (the dated layers pertaining to U1) and Roccamonfina post-caldera activity (the leucite free tephra pertaining to U2). This only slightly predates the inferred age of U2 unit to ca. 430 ka, but does not change the stratigraphic setting. It will be specified more clearly and the references cited.

Line 247: refer to Plate 1 to locate Criscia.

29) Accepted. The locality Criscia will be removed from the text (this is not necessary).

Lines 298-302: estimated post-LGM throw rates are not convincing. In fact, profile 13 shows a 6.5 m offset in Sd1 unit, which you constrained to the late Lower – Middle Pleistocene, whereas profiles 14 and 15 show offset U1 unit, dated at the early Middle Pleistocene. Throw rates should be referred to these chronological intervals, which may lead to values like those one obtained by profiles 3, 9, 11 and 12.

30) We agree with the reviewer that the age of fault scarps is not constrained there. We infer a post-LGM age because of:

- the scarp offsets the mountain slope in a similar way, independently from the underlying carved unit (U1 or sd1), suggesting the scarp formed after the shaping of the mountain slope;

- similarity with the area of Fig. 6, in terms of both the scarp size (height) and geometrical relations between the scarp and the mountain slope.

But we agree that the minim value (post-U1) is a more conservative value, and in any case a correct one. We will change the present value in the minimum throw rate value, as suggested by the reviewer.

Line 355: if the scarp is due to erosion, as you mentioned, then it is not a fault scarp and you cannot derive throw rates.

31) This criticism is unacceptable.

At line 355 we do not say " … *the scarp is due to erosion* …".

At line 355 we write exactly this: "A ~1 m-high eroded scarp across the F3-F5 fault zone is visible on a ….".

Once a coseismic surface faulting occurs, a coseismic scarp forms, if there is vertical displacement (e.g., normal fault). After surface faulting, the fault scarp starts to be degraded and eroded, and the scarp retreats. The size and shape of the eroded scarp depend on several factors, including: size of the coseismic displacement, single-event or compound scarp, lithology, climate, erosional processes, time after surface faulting, etc.

This process has been described in papers that are milestones of the earthquake geology literature. Please, read:

- Wallace, R. E. (1977). Profiles and ages of young fault scarps, north-central Nevada. Geol. Soc. Am. Bull. 88, 1267–1281.

and/or

- Mc Calpin, 2009 - Paleoseismology. Academic Press. Chapter 3

- Yeats, R.S., Sieh, K., Allen, C.R., 1997 - THE GEOLOGY OF EARTHQUAKES –Oxford Univ. Press. Chapter 7.

SECTION 6.1 – OVERALL ARCHITECTURE AND KINEMATICS: you should also refer to scheme showing tectonic evolution of the SMF proposed by Valente et al. (2019), and not only to papers by the authors.

SECTION 6.2 – THROW RATES: regarding mature geomorphology of mountain front, you did not introduce any data supporting this hypothesis, so you should just refer to Valente et al. (2019).

32) See replies to comment N. 2.

You should also consider the possibility that the scarp you used to derive throw rate is only due to erosion (see general comments).

33) See replies to comments N. 4, 5 and 31.

Line 405: compare your values with values from Cinque et al. (2000) and Ferranti et al. (2014).

34) Accepted. We will compare with values from Cinque et al. (2000) and Ferranti et al. (2014).

SECTION 6.3 – SEISMOGENIC POTENTIAL: this section should be revised according to my comments on how you estimated throw rates.

Table 1 and 2: you should add coordinates of sampling site and show their location in some map, at least in Fig. 4 and Plate 1.

Regarding samples in the surroundings of Sant'Angelo d'Alife, you may add their location in Fig. 2.

35) Accepted. We will add coordinates in Table 2.

The Sant'Angelo d'Alife site is already located in Fig. 2. Please, see Fig. 2.

FIGURES

Figure 1: correct "Northern Maters Fault System".

36) Thank you, we will correct the name.

Why did you place the 346 epicentre in the core of the Matese massif? Galli and Naso (2009) and Galli et al. (2017) placed it in the surroundings of Ailano. You also mentioned, in the main text, that the SMF may be the source area of the 346 earthquake, so it looks a bit in contrast the location of this event in the core of the massif. What data do you have to place it to the south of the Matese Lake?

37) This is the epicentre from the CFTI5Med catalogue (Guidoboni et al., 2019). We will explain in the text and in caption.

Furthermore, are there marine deposits within the Quaternary basin? Are you sure? If so, cite papers mentioning these marine deposits within the main text. Also, the upper age limit of this deposits is missing.

Add coordinates and north arrow.

Figure 2: fault traces should be thicker.

All the above suggestions will be adequately considered during the revision.

Figure 3: you show that units U3 and sd2 contain the NYT and the CI tephra layers. Anyway, you never presented data showing the presence of this tephras both in the main text (see lines 228 to 234) and in the boreholes section of Plate 1. What data do you have? Please, introduce them in the main text.

The occurrence of these tephra in the outcropping and borehole successions has been reported in a PhD thesis (Leone, 2018), with full chemical data supporting this attribution. We will make full reference on the occurrence of these tephra in the new supplementary section. See reply to comment N. 28.

The data are available in the appendix sections of the PhD thesis "Leone, 2018. Studio dell'evoluzione quaternaria di alcune conche intermontane dell'Appennino campano-molisano, a supporto della pianificazione e gestione del territorio e della prevenzione del rischio sismico. https://iris.unimol.it/handle/11695/75943#.YSxoUY4zZPY

Figure 4: revise accordingly to comments at geological map and cross-sections in Plate 1.

Figure 6: the outcrop of Triassic bedrock close to the fault trace and the lateral variation in scarp height, which is maxima where the bedrock outcrops, suggest that the scarp is due to differential erosion and not to fault activity.

See replies to comments N. 4 and 5.

Figure 7: specify, in the caption, if fault colours are referred to fault colours in Plate 1. If not, add a small legend to specify what different fault colours indicate.

Accepted.

Figures 8 and 9: to facilitate readability of these figures, authors should complete details of faulted deposits with reference to units mapped in the geological map (Fig. 4 and Plate 1). Unit U1 is present in this sector of the study area, which should refer to units 1 to 3 in both figures, whereas C14 data indicates that units 4 to 7 are Holocene in age, but there are no Holocene deposits in this sector of the pediment. Furthermore, it seems to me that unit 6 in Figure 8 can be prolonged towards the NE and may cover the fault trace, but maybe this different interpretation is due to colours in the picture.

Accepted. We will show the relations between the units in the logs and the units in the main map.

Unit 6 is clearly faulted. No questions about that. Perhaps the Reviewer used a low-resolution image. We are planning to add a photographic documentation of fault F5 in the supplementary material (see Fig. C below).

[Figure]

Photographic documentation of fault zone 2 (Fig. 9 of the main text) during different stages of wall cleaning

[Figure]

very steep, no shear (slightly degraded fault scarp; unit 7 seals the scarp)

U8

U7

apparent bends due to change in dip of trench wall

U4

change in dip of trench wall

U6

p

U3

p = paleosol

antithetic fault

U7

U4

Unit 6 faulted

U5

U6

U4

U4b

Figure 10: age of unit 4 should be "BCE" and not "CE".

Figure 11: I missed in the main text Middle Pleistocene to 15.3 ka throw rates for the Raviscanina and Piedimonte Matese areas. I just found throw rate >0.35 mm/yr during the late Holocene (see line 379 of the main text).

All the above suggestions will be adequately considered during the revision.

SUPPLEMENTARY DATA

TAB. S3_Data points with structural data

- See comment to line 17.

38) General reply to specific comments to Plate S1 and geologic sections (below).

Geologic maps and sections are always a combination of: 1) firm constraints (outcrops); 2) geometric reconstructions according to stratimetric rules and structural geology; and 3) reasonable interpretations. Al the geologists doing field mapping know about that.

In the map and sections proposed as supplementary material, we did our best on the basis of months of field work and reasoning. Certainly they can be improved. But we are sure to have done the best with a scientific approach.

We are happy to modify and improve map and sections if they conflict with documented field constraints, or if there are errors, geometrical problems, inconsistencies.

But, if the asked modifications are merely aimed at proposing alternative interpretations, without constraints, we need to be free of rejecting them.

PLATE S1

MAIN MAP

- The Gioia Sannitica fault passes, laterally, to a generic Quaternary fault to the north of Calvisi. How it is possible?

Accepted. We will correct the map.

- The tectonic setting near Curti, Petrella and Caselle is very tricky (see comments in the "General comments" section)

- The Quaternary faults to the west of GF are not supported by your data (see next comments), so I suggest removing them and modifying the cross-sections accordingly

See specific reply.

CROSS-SECTION

Enlarge text on the X and Y axes. Add "Elevation (m a.s.l.)" to the left of the Y axe and "distance (m)" below the X axes.

Accepted. We will modify the sections.

CROSS SECTION A-A'

- I have some doubts about blue faults you reported in this cross-section. Borehole data do not show any offset, as borehole SS6 drill the Miocene unit whereas boreholes SS2 and PMS4 drill alluvial units. If you project the topographic surface near borehole SS6 south-westwards, you do not need a fault (see red lines in the figure below). So, it is easier to interpret the contact between alluvial fan deposits and Miocene deposits as stratigraphical.

[Figure]

- Furthermore, borehole data do not allow to infer the presence of another fault, near borehole PMS6, because there are no offsets in the Quaternary units. In this area, borehole data do not allow to infer both the thickness of the alluvial fan deposits and the depth of the bedrock, so the lower part of the cross-section is not supported by data. I suggest ending the cross-sections at the base of boreholes (a possible solution is provided below).

[Figure]

39) Partially accepted.

The debated faults are dashed faults. This means that they are not well constrained (dashed if inferred or buried). But there are reasons for their presence.

The fault close to PM6 is drawn in order to justify the difference in elevation of the top of the carbonate bedrock in the hanging wall (deeper than -200 m a.s.l. from geophysical data in Corniello and Russo, 1999; see section 1 in Fig. 4) compared to the footwall (outcropping Cretaceous limestone at elevations higher than + 175 m a.s.l.; see outcrops in the geologic map).

The fault between SS2 and SS6 is drawn (again dashed) because of the higher elevation of Miocene sediments in the footwall compared to Quaternary sediments in the hanging wall.

But, we agree that we have no constraints for inferring the displacement of the Quaternary units. We will modify the section adding a dashed line along the bottom of Quaternary units, as suggested, and a question mark at the intersection with the inferred fault.

- I also found some incongruences about Miocene deposits dip in the hangingwall of red fault (e.g, the Gioia Sannitica fault). In fact, Miocene deposits dip towards SW in cross-section A-A' and D-D', towards NE in cross-section B-B', C-C' and F-F', and are horizontal in cross-section E-E'. Few bedding data on Miocene deposits reported in the main map do not allow to support such variations, so the authors should re-draw this part of the cross-sections.

- Lastly, the Cretaceous units dip 40° towards the SW. This is the only bedding data reported in the main map and it does not support folding of this units as reported in cross-section A-A'.

40) Accepted.

Unfortunately bedding data are only a few, due to poor exposures. We agree that sections are poorly constrained from this point of view.

The attitude of bedding has been drawn and laterally extrapolated/interpreted in order to preserve as much as possible the cut-off angles across faults. This is a basic structural geology rule.

But we understand this criticism. In the revised version, we will draw attitudes only where reasonably constrained.

Furthermore, I evaluated the dip of the Mt. Olnito slope which resulted to be of 45°. This would suggest that this slope is a dip-slope. So, if the very recent activity of the Gioia Sannitica fault is estimated by the steeper lower sector of the Mt. Olnito slope, I think this interpretation is not correct.

This criticism is not clear to us. The activity of the GF has been evaluated on the basis of all the data collected in the field and described in the paper.

[Figure]

CROSS SECTION B-B'

- What are the evidences to infer the location of a Quaternary fault to the NE of the Gioia Sannitica fault?

41) We have structural evidence from filed mapping (fault plane and fault zone) plus geomorphic lineaments in the LIDAR topography and U1 deposits in the hanging wall.

- Your data do not allow to infer depth of the Quaternary bedrock, so you should stop your cross section at the base of the boreholes (the same for all the cross-sections)

See reply N. 39

- Again, your data do not support the presence of an inferred Quaternary fault to the SW of San Potito Sannitico

See reply N. 39

CROSS SECTION C-C'

- Stratigraphy of the borehole S"S93" is not reported in the borehole stratigraphy figure

42) You are right. It is missing by mistake. It is a water well. It will be added in the figure.

- There are no evidences to infer the presence of the dashed blue fault.

See reply N. 39

- What data do you have to infer the tectonic contact between Mesozoic carbonates and Miocene units along blue fault?

See reply N. 39

CROSS SECTION D-D'

- Again, there are no evidences to infer the presence of the dashed blue faults.

43) The blue fault (again dashed because poorly constrained) is added in order to explain the difference in elevation between the NE-dipping Cretaceous rocks in the footwall at ~275 m a.s.l. (Auduni) and the NE-dipping Cretaceous rocks in the hanging wall ~250 m a.s.l. (east of Carattano), and considering the strike-lines from bedding attitude.

- How can you hypothesize the presence of the Varicoloured Clays? Looking at your map, they seem to be limited to the Petrella area. See also comments in the "General comment" section.

44) In correspondence of the geologic section the Varicoloured clays are covered by Quaternary sediments. So we don't know from surface geology, but according to the elevation of the boundary, the section should intercept this boundary.

In the stratigraphy of the S"S93" well, blue/gay clays were drilled. We interpreted those as VC. But, we will add a question mark in the geologic section.

CROSS SECTION E-E'

- The map shows, in the surroundings of Auduni, the presence of U2 deposits whereas U1 deposits are reported in the cross-section.

45) This is a mistake. Thank you. We will correct this mistake.

- This cross-section is not very clear as there are no bedding data in its surroundings that allow the infer the complex geological setting, with folded Miocene and Mesozoic units.

46) This is drawn in order to be consistent with the structural setting of the adjacent section D-D'. Please note that the fold in section D is drawn in order to account for the available bedding attitudes (e.g., NE-dipping K unit) in the

hanging wall of GF, and in order to preserve cut-off angles across the GF. Anyway, we will check carefully the consistency of the cross section.

See also reply N. 39.

- The Gioia Sannitica fault is inferred in the main map whereas it is certain in this cross-section.

47) Accepted. This is a mistake. We will modify the cross section.

- Why you did not project borehole GSA4 in cross-section E-E'?

48) Accepted. We will project the well on the section.

CROSS SECTION F-F'

- There are no evidences to infer the presence of Quaternary faults to the SW of the Gioia Sannitica fault.

49) The blue fault (again dashed because poorly constrained) is added in order to explain the difference in topographic elevation between the footwall, where Miocene siliciclastic deposits crop out, and the hanging wall, where Quaternary deposits accumulated.

BOREHOLE DATA

- You distinguished, in the main map, SD1 and SD2. I agree, but most of the boreholes drill another SD unit, that is not listed in the legend. Looking at both the map and the cross-sections, I guess it should be the Miocene unit. Borehole data must be consistent with geological map and cross-sections to avoid any misunderstanding.

- The unit CC drilled in boreholes S14 and GSA5 is not listed in the legend. What is it? Furthermore, authors could project borehole S14 on cross-section A-A'.

- Borehole GSA7 drill SD1 deposits whereas it drills SD2 deposits in cross-section D-D'.

- Boreholes S9, S12 and PMS2 are placed in an area where U2 deposits are present whereas they drill U1 deposits.

- I cannot find boreholes PMS6 and SS1 in the main map. Where are they located?

- Boreholes SS5 and SS6 drill SD unit in the borehole stratigraphy figure whereas they drill Miocene unit in cross-section A-A'. So, is the SD unit composed of debris slope deposits or the SD unit correspond with the Miocene unit?

- Stratigraphy of borehole GSA1 indicates that it drills, in the upper part, U3 deposits whereas it is in an area where U1 deposits are present. Furthermore, you indicated that the WTT tephra is present within the U2 deposits, whereas, in borehole GSA1, the WTT occurs within U3 deposits.

- Regarding the WTT, how can you establish that this tephra layer is the WTT? Did you sample it in all boreholes and performed tephrostratigraphic analysis? Or the classification of this tephra layer as the WTT is just your assumption?

Please, see our comment n. 27 and 28. The legend of the Figure will be changed accordingly.

- Again, what is the SD units in boreholes GSA3, GSA4, GSA5, GSA6, GSA7, SPS1, SPS2, SPS3, SPS4, SPS5, SPS6 and SPS7?

- Stratigraphy of borehole GSA6 indicates that it drills, in the upper part, SD2 deposits whereas it is in an area where Miocene deposits are present.

- Boreholes 158202, 163920 and S"S93" and reported in the main map but are not detailed in the borehole stratigraphy figure.

50) You are right. There are several inconsistencies between borehole stratigraphy in the inset and map and sections. This is due to a change in the acronym of the geological units, for making them more understandable by English readers, but that was not uniform by mistake. For example: CC (Calcari cristallini) in borehole stratigraphy is K in the map and sections; SD (Siliciclastic deposits) in borehole stratigraphy is M in the map etc.

Thank you for pointing this out.

We will check carefully all these observations during the revision of the manuscript.

TECHNICAL CORRECTIONS

Line 84: type NE-SW instead of SW-NE

?

Line 168: delete "which was".

Corrected.

Line 173: the letter "o" of Alo, Feo, Sio, should be written in capitals.

No. Lower case is correct. They are acid ammonium oxalate extractable.

Line 241: it should be Mt. Acero

Corrected.

---

## Author Comment (AC3)

**Response to Reviewer3 (A.M. Michetti)**

Black: original comments by Reviewer

Red: response by Authors

The manuscript is an interesting contribution for the characterization of seismic hazard from a major earthquake fault in Southern Italy, along the Southern Border of the Matese Massif. The collected data are excellent, and very detailed.

The general geologic and seismotectonic interpretation is convincing. My main comment is on the San Potito fault exposure along the Gioia Sannitica segment. I suggest to provide a general geological cross section of this exposure, to help the reader to understand the details presented in Figure 8 and 9.

1) Accepted. Certainly the geologic section will help the reader.

In fact, we have already drawn a geologic section across the San Potito site. This is Section C-C' in Plate S1 of the Supplementary material.

But, we understand that a geologic section within the main text is more efficient. We will add the geologic section in the main text, with location of the fault zones illustrated in Fig.s 8 and 9.

[Figure]

In my opinion, the most relevant result here is that the fault must have been activated at this aite during late Holocene. The implication is that atthis site capable faulting shifted basinward, moving from the range front to the piedmont belt.

2) Interesting suggestion.

In section C-C' (above), we interpreted faulting at the sites of Figs. 8 and 9 as due to synthetic splaying from the main fault. The site is very close to the range front. Moreover, from the topographic profile n. 9 in Fig. 7, it seems that the displacement (the topographic scarps) is partitioned between the two faults (main fault and synthetic splay). But, we cannot exclude a migration basinward, as suggested. Perhaps, with the new splay, the fault is trying to regularize its trace, becoming more and more strait.

 The explanation of this shift is not clear, and must be discussed in the revised version of the manuscript. In any case, this is a relevant new result, and very interesting one. One may argue that the Matese Southern Front migth have been characterized by large slip-rate fluctuations during the mid to late Quaternary.

3) We agree. A deeper discussion on the activity of the SMF system, and the possible slip rate fluctuations will be discussed in the new Discussion section, after a general re-organization of the paper, as suggested by Reviewers 1 and 2.

Some specific comment is included in the attched annotated manuscript.

All the comments and suggestions in the attached pdf will be adequately considered during the revision.